# In-Context Benign Overfitting: A Feature-Selection Model in In-Context Linear Regression

## Abstract

In in-context learning (ICL), a frozen pre-trained model solves tasks by conditioning on a prompt of a few input–output examples, without gradient updates. If the task was present in pretraining but the particular prompt sequence was not, the resulting in-distribution generalization is retrieval-based ICL. Learning-based ICL instead reflects out-of-distribution generalization: the model succeeds on prompts generated by a novel task. Empirically, both forms improve with scale. By analogy to benign overfitting in supervised learning, we call this in-context benign overfitting: larger models more faithfully memorize the pretraining tasks (improving retrieval ICL) while also generalizing better to novel tasks (improving learning ICL). We prove that this phenomenon already arises in a minimal in-context linear-regression feature-selection model. In contrast, standard in-context linear-regression models exhibit a retrieval–learning tradeoff, where the emergence of learning-based ICL coincides with degraded retrieval-based performance.

## 1 Introduction

In-context learning (ICL) has established itself as a cornerstone capability of large language models, where the model solves tasks at inference time by conditioning on a prompt of input–output demonstrations, without any updates to model weights (Brown et al., 2020). While the prompt contains only a handful of examples, the model also draws on information acquired during pretraining. As a result, there are two qualitatively different settings in which a model can succeed: it can perform well because the underlying task was present during pretraining, or it can perform well on prompts generated by a task that is novel relative to pretraining. Pan et al. (2023) introduced a useful distinction between these two modes of ICL, which they call *retrieval* and *learning*. Concretely, let $\mathcal{D}_{\text{task}}^{\text{train}} = \{\boldsymbol{\theta}_1, \boldsymbol{\theta}_2, \ldots, \boldsymbol{\theta}_M\}$ denote the set of tasks present during pretraining. For a task $\boldsymbol{\theta}$, a prompt consists of $T$ demonstrations and a query, $X_{\boldsymbol{\theta}} = [(\boldsymbol{x}_1, y_1), (\boldsymbol{x}_2, y_2), \ldots, (\boldsymbol{x}_T, y_T), \boldsymbol{x}_q]$, where labels $y_t = f_{\boldsymbol{\theta}}(\boldsymbol{x}_t)$ are generated through a fixed mapping $f$ that is shared across demonstrations inside the prompt. This mapping is parameterized by the task $\boldsymbol{\theta}$ and applies to all $t \in [T]$ as well as the query $\boldsymbol{x}_q$. The two modes are then defined as follows:

- **Retrieval mode (ID generalization).** We sample $\boldsymbol{\theta} \sim \mathcal{D}_{\text{task}}^{\text{train}}$, so while the prompt $X_{\boldsymbol{\theta}}$ is unseen during pretraining, the underlying task is drawn from the pretraining task set. We evaluate whether the model correctly predicts the query label $y_q$. Success in this mode is consistent with the model having *memorized* the pretraining tasks and *retrieving* the correct task mechanism from its parameters. This mode, therefore, probes task-level in-distribution (ID) generalization.

- **Learning mode (OOD generalization).** We sample $\boldsymbol{\theta} \sim \mathcal{D}_{\text{task}}$ from a distribution that differs from $\mathcal{D}_{\text{task}}^{\text{train}}$. We again evaluate prediction on $y_q$. Since the task is novel relative to pretraining, the model cannot rely on task memorization alone and must *learn* from demonstrations. This mode probes task-level out-of-distribution (OOD) generalization.

Hereon, we will refer to these as ID and OOD generalization, respectively.

Understanding these two modes has motivated a surge of work in synthetic, fully controlled settings. These studies train transformers from scratch on sequences generated by a finite task set, for example in-context linear regression (Garg et al., 2022; Raventos et al., 2023; Carroll et al., 2025) or Markov-chains (Park et al., 2025), and then evaluate ID and OOD generalization as discussed above. This line

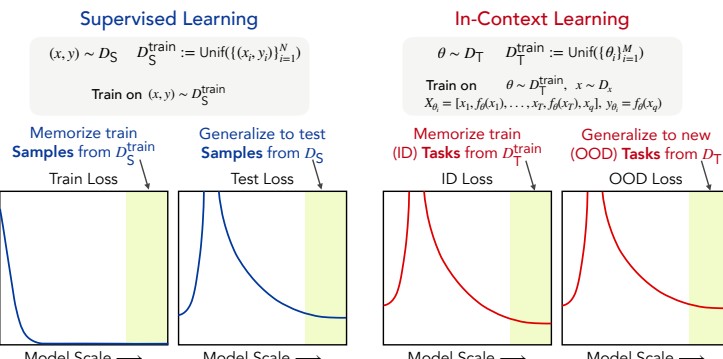

Figure 1: **Benign Overfitting in Supervised vs. In-Context Learning.** Left illustrates benign overfitting commonly observed in standard supervised learning setups, where large models far beyond the interpolation threshold (green band) memorize the train *samples* while simultaneously generalizing to new ones. We introduce and formalize an in-context analogue of this phenomenon (right): large models (in green) memorize *tasks* seen during (pre)training while simultaneously generalizing to novel out-of-distribution tasks.

of work has uncovered several qualitative phenomena, including task-diversity thresholds (Raventos et al., 2023; Park et al., 2025) for the emergence of *good* OOD generalization and transient dynamics (Singh et al., 2023; Carroll et al., 2025) in how the two modes evolve over the course of training. We defer a detailed discussion and pointers to the relevant papers to Appendix A.

One question, however, remains conceptually unsettled: *how should ID and OOD generalization evolve with model scale?* In the era of scaling laws, the prevailing intuition is that larger models should excel in both modes. Indeed, this has been observed experimentally: Wei et al. (2023) (Figs. 2 and 3) show that in large language models, both modes tend to improve with scale: larger models more accurately solve tasks encountered during pretraining and also generalize better to novel tasks at inference time.

**Contributions.** Drawing an analogy to the literature on benign overfitting in supervised learning (Bartlett et al., 2020; Hastie et al., 2020; Belkin et al., 2020), we frame the scaling behavior of ID and OOD generalization in ICL as what we call *in-context benign overfitting*.

> **In-Context Benign Overfitting**
>
> **Definition 1.** *In-context benign overfitting is the property whereby large (overparameterized) models memorize ID ICL tasks, achieving low risk on prompts from $\mathcal{D}_{task}^{train}$, while simultaneously generalizing to novel OOD ICL tasks (from $\mathcal{D}_{task}$).*

This lifts the classical notion of benign overfitting from the sample level to the task level: whereas standard benign overfitting concerns memorization of noisy training *samples*, in-context benign overfitting concerns memorization of pretraining *tasks*. See Fig. 1 for an illustration.

We ground our framework in the minimal setup of in-context linear regression (Garg et al., 2022), employing a simplified linear attention model consistent with recent theoretical works (Zhang et al., 2023; Lu et al., 2025; Wu et al., 2023; Zhang et al., 2025). To explicitly model the effect of scale, we incorporate a feature subsampling mechanism inspired by the double descent literature (Hastie et al., 2020; Belkin et al., 2020). Using data generated from this model, we study the two canonical predictors extensively studied in classical statistical learning: the minimum-norm interpolator (in the overparameterized regime) and the least-squares solution (in the underparameterized regime). We analyze these predictors in the proportional high-dimensional asymptotic limit, where the number of (pre)training tasks, number of (pre)training prompts, number of in-context demonstrations, number of learnable parameters, and the ambient feature dimension grow strictly in proportion (Lu et al., 2025). In this regime, we derive sharp analytical characterizations of the ID and OOD risks, demonstrating that our theoretical predictions accurately track the empirical risk curves and faithfully capture the phenomenon of in-context benign overfitting.

## 2 PROBLEM SETUP

**Data Model.** We study in-context linear regression first introduced by Garg et al. (2022). Input sequences are interleaved pairs $X = [\boldsymbol{x}_1, y_1, \ldots, \boldsymbol{x}_T, y_T, \boldsymbol{x}_{T+1}]$, where features $\boldsymbol{x}_t \sim \mathcal{N}(\boldsymbol{0}, \boldsymbol{\Sigma_x})$ and labels $y_t = \boldsymbol{w}^\top \boldsymbol{x}_t + \epsilon_t$ for a task vector $\boldsymbol{w} \sim \mathcal{D}_{\text{task}} = \mathcal{N}(\boldsymbol{0}, \boldsymbol{\Sigma_w})$ and noise $\epsilon_t \sim \mathcal{N}(0, \sigma_n^2)$. We refer to $\boldsymbol{x}_q := \boldsymbol{x}_{T+1}$ as the query vector, $y_q := y_{T+1}$ as the query label, and the number of demonstrations $T$ as the context-length. Given the $T$ demonstrations $(\boldsymbol{x}_t, y_t)_{t=1}^T$, the goal is predicting $y_q$.

**Training Data.** We construct a training set by sampling a finite pool of $M$ task vectors $\{\boldsymbol{w}_i\}_{i=1}^M \sim \mathcal{D}_{\text{task}}$ independently. The training task distribution is uniform over this pool, i.e. $\mathcal{D}_{\text{task}}^{\text{train}} = \text{unif}\{\boldsymbol{w}_1, \ldots, \boldsymbol{w}_M\}$. We generate $N$ training sequences $X_i$ by sampling $\boldsymbol{w}_i \sim \mathcal{D}_{\text{task}}^{\text{train}}$ and constructing query labels $y_q^i = \boldsymbol{w}_i^\top \boldsymbol{x}_q^i + \epsilon_q^i$, for all $i \in [N]$.

**Learning Model.** To solve the ICL linear regression problem, we employ a simplified model inspired by linear attention. In its vanilla form, this model has been used extensively in prior work as a tractable framework for theoretically analyzing ICL (Zhang et al., 2025; Lu et al., 2025; Wu et al., 2023; Zhang et al., 2024; Frei & Vardi, 2024).

Our key modeling contribution is a feature-selection formulation inspired by the double descent literature in supervised learning (Hastie et al., 2020; Belkin et al., 2020). We vary model capacity by restricting the learner to a subset of original features $\mathcal{S} \subseteq [d]$ of size $p = |\mathcal{S}|$, sampled uniformly without replacement. The model first constructs a feature embedding $\mathbf{h}_{\mathcal{S}_p}(X) = \texttt{vec}\big(\boldsymbol{x}_q[\mathcal{S}], \hat{\boldsymbol{w}}_{\text{avg}}[\mathcal{S}]^\top\big) \in \mathbb{R}^{p^2}$ where $\hat{\boldsymbol{w}}_{\text{avg}}[\mathcal{S}] = \frac{1}{T}\sum_{t=1}^T y_t \boldsymbol{x}_t[\mathcal{S}]$ is the *averaging estimator* restricted to $\mathcal{S}$. The prediction is then linear in these features $f_{\boldsymbol{\theta}, \mathcal{S}}(X) = \boldsymbol{\theta}^\top \mathbf{h}_{\mathcal{S}_p}(X)$, with trainable parameters $\boldsymbol{\theta} \in \mathbb{R}^{p^2}$.

Given training sequences $\{X_i\}_{i=1}^N$, we obtain $\boldsymbol{\theta}$ by minimizing the empirical risk $\mathcal{L}_{\text{emp}}(\boldsymbol{\theta}) := \frac{1}{N}\|\boldsymbol{H}_{\mathcal{S}_p}\boldsymbol{\theta} - \boldsymbol{y}\|^2$. The solution $\hat{\boldsymbol{\theta}}_{\mathcal{S}_p}$ follows the problem geometry:

- **Underparameterized regime**: For full column rank $\boldsymbol{H}_{\mathcal{S}_p}$, $\hat{\boldsymbol{\theta}}_{\mathcal{S}_p}$ is the unique least-squares solution $\hat{\boldsymbol{\theta}}_{\mathcal{S}_p} := \arg\min_{\boldsymbol{\theta}} \|\boldsymbol{y} - \boldsymbol{H}_{\mathcal{S}_p}\boldsymbol{\theta}\|^2$.
- **Overparameterized regime**: For full row rank $\boldsymbol{H}_{\mathcal{S}_p}$, gradient descent initialized at zero converges to the minimum-norm interpolator (Hastie et al., 2020; Bartlett et al., 2020), i.e. $\hat{\boldsymbol{\theta}}_{\mathcal{S}_p} := \arg\min_{\boldsymbol{\theta}} \|\boldsymbol{\theta}\|$ s.t. $\boldsymbol{H}_{\mathcal{S}_p}\boldsymbol{\theta} = \boldsymbol{y}$.

This characterization allows us to study generalization of $\hat{\boldsymbol{\theta}}_{\mathcal{S}_p}$ across both regimes as the model size $p$ varies. We evaluate the trained predictor $\hat{\boldsymbol{\theta}}_{\mathcal{S}_p}$ on two types of generalization:

(i) **In-distribution (ID)**: performance on training tasks, i.e., $\boldsymbol{w} \sim \mathcal{D}_{\text{task}}^{\text{train}}$;

(ii) **Out-of-distribution (OOD)**: performance on novel tasks, i.e., $\boldsymbol{w} \sim \mathcal{D}_{\text{task}}$.

To define these metrics, we first introduce the *task-specific mean-square-error loss*, which measures the prediction error of $\hat{\boldsymbol{\theta}}_{\mathcal{S}_p}$ on sequences generated from a fixed task vector $\boldsymbol{w}$, $\mathcal{L}(\hat{\boldsymbol{\theta}}_{\mathcal{S}_p}; \boldsymbol{w}) := \mathbb{E}_{X,\epsilon}\big[(\hat{\boldsymbol{\theta}}_{\mathcal{S}_p}^\top \mathbf{h}_{\mathcal{S}_p}(X) - y_q)^2\big]$, where the expectation is over the feature vectors and noise in the sequence $X$, i.e., $\boldsymbol{x}_1, \ldots, \boldsymbol{x}_T, \boldsymbol{x}_q \sim \mathcal{N}(\boldsymbol{0}, \boldsymbol{\Sigma_x})$ and $\epsilon_1, \ldots, \epsilon_q \sim \mathcal{N}(0, \sigma_n^2)$. The ID and OOD risks are then defined by averaging over the respective task distributions: $\mathcal{L}_{\text{ID}}(\hat{\boldsymbol{\theta}}_{\mathcal{S}_p}) := \mathbb{E}_{\mathcal{S}_p}\mathbb{E}_{\boldsymbol{w} \sim \mathcal{D}_{\text{task}}^{\text{train}}}\mathcal{L}(\hat{\boldsymbol{\theta}}_{\mathcal{S}_p}; \boldsymbol{w})$, $\mathcal{L}_{\text{OOD}}(\hat{\boldsymbol{\theta}}_{\mathcal{S}_p}) := \mathbb{E}_{\mathcal{S}_p}\mathbb{E}_{\boldsymbol{w} \sim \mathcal{D}_{\text{task}}}\mathcal{L}(\hat{\boldsymbol{\theta}}_{\mathcal{S}_p}; \boldsymbol{w})$. In both cases, the outer expectation $\mathbb{E}_{\mathcal{S}_p}$ averages over the random feature subset $\mathcal{S}$, ensuring that we evaluate performance independently of any particular realization of it.

## 3 SHARP ASYMPTOTICS OF ID AND OOD RISKS

We characterize the ID and OOD risks in the joint high-dimensional limit where the ambient dimension $d$, context length $T$, model size $p$, number of training tasks $M$, and number of training sequences $N$ all grow to infinity as follows: $N, T, M, d, p \to \infty$ with

$$\frac{N}{d^2} \to \nu, \quad \frac{M}{d} \to \mu, \quad \frac{T}{d} \to \tau, \quad \frac{p}{d} \to \rho. \tag{1}$$

Throughout this section, we assume isotropic features and task vectors, i.e., $\boldsymbol{\Sigma_x} = \sigma_x^2 \mathbb{I}_d$ and $\boldsymbol{\Sigma_w} = \sigma_w^2 \mathbb{I}_d$. As we will show this setting already suffices to establish the in-context benign overfitting phenomenon. Further, following Lu et al. (2025), we fix $\sigma_x = 1/\sqrt{d}$ and $\sigma_w = 1$.

**Analysis Framework.** Our derivations leverage a Gaussian equivalence principle (Mei & Montanari, 2019; Goldt et al., 2020). To find the asymptotic risks, we analyze a Linear Gaussian Equivalent Problem (LGP) which matches the first and second-order statistics of our original model (see App. B). The predictive power of this equivalence is rooted in the broader phenomenon of Gaussian universality (Goldt et al., 2020). While a formal proof of equivalence for our problem is left for future work, we analyze the LGP using the Convex Gaussian Min-Max Theorem (CGMT) and verify experimentally (Sec. 4) that its theoretical risk curves accurately track the original model's performance.

Next, we present our main results. See Appendix B for a detailed overview of the derivations.

**Theorem 1** (Overparameterized Regime, $\nu < \rho^2$). *Under the asymptotic scaling (Eq. (1)), as $d \to \infty$, the risks of minimum-norm estimator $\hat{\boldsymbol{\theta}}_{\mathcal{S}_p}$ for the LGP (Def. 3) converge in probability to:*

$$\mathcal{L}_{OOD}(\hat{\boldsymbol{\theta}}_{\mathcal{S}_p}) \xrightarrow{P} \left(1 - \frac{\nu}{\rho^2}\right)\rho(1+ac)(1-c(1+a+ac)) + (1+a+ac)(\sigma_n^2 + \rho c + 1 - \rho), \quad \mathcal{L}_{ID}(\hat{\boldsymbol{\theta}}_{\mathcal{S}_p}) \xrightarrow{P} \frac{\nu}{a^2}\bar{\beta}_*^2,$$

*where $c := \frac{\sigma_n^2 + 1}{\tau}$ and the scalars $a$ and $\bar{\beta}_*$ are the unique positive solutions to Eq. (6) (see App. B).*

**Theorem 2** (Underparameterized Regime, $\nu > \rho^2$). *Under the asymptotic scaling (Eq. (1)), as $d \to \infty$, the risks for the least-squares estimator $\hat{\boldsymbol{\theta}}_{\mathcal{S}_p}$ for the LGP (Def. 3) converge in probability to:*

$$\mathcal{L}_{OOD}(\hat{\boldsymbol{\theta}}_{\mathcal{S}_p}) \xrightarrow{P} m_c\big(\kappa_\infty^2(1+c) - 2c^2\rho\big) + (1+c)\rho c^2 m_c' + \sigma_n^2 + 1 - \rho + \rho c, \quad \mathcal{L}_{ID}(\hat{\boldsymbol{\theta}}_{\mathcal{S}_p}) \xrightarrow{P} \frac{\nu}{\rho^2}\kappa_\infty^2,$$

*where $\kappa_\infty$ is a scalar defined in Eq. (9) (see App. B), $m_c := m_\gamma(z_c)$ is the Stieltjes transform of the Marchenko-Pastur law with parameter $\gamma = \rho/\mu$ evaluated at $z_c = -c$, and $m_c'$ denotes the derivative of $m_\gamma(\cdot)$ at $z_c$.*

## 4 EXPERIMENTAL RESULTS

We present experimental results demonstrating in-context benign overfitting and validating our theoretical predictions. We follow the setup in Sec. 2 unless otherwise noted; see App. I for details.

Fig. 2 compares our derived asymptotic risk curves (Theorems 1 and 2) against empirical loss values. We plot both ID and OOD risk as a function of the relative model scale $\rho^2/\nu$ (specifically, we fix $N$ and vary $p$). We observe that the theoretical predictions closely match the empirical observations. The vertical asymptote at $\rho^2/\nu = 1$ marks the phase transition between the underparameterized ($\rho^2/\nu < 1$) and overparameterized ($\rho^2/\nu > 1$) regimes. As model scale increases, both ID and OOD loss exhibit double descent. Importantly, increasing the model scale causes the ID loss to converge to a significantly lower value compared to the underparameterized regime, while the OOD loss converges to a value comparable to the underparameterized minimum. This demonstrates that the model benignly overfits the

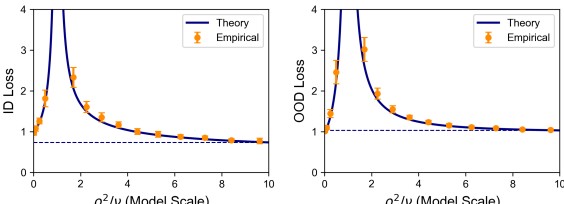

Figure 2: **In-Context Benign Overfitting**. Comparison of our theoretical risk curves (solid lines) vs. empirical observations (points) for ID Loss (left) and OOD Loss (right) as a function of model scale $\rho^2/\nu$ (with fixed $\nu$). The horizontal dashed lines denote the asymptotic value in the overparameterized regime. The vertical asymptote at $\rho^2/\nu = 1$ marks the interpolation threshold. In the overparameterized regime ($\rho^2/\nu > 1$), the model exhibits in-context benign overfitting: it effectively memorizes pre-training tasks (low ID risk) while simultaneously yielding a small risk on OOD tasks.

training tasks—memorizing ID tasks while simultaneously generalizing to OOD tasks, capturing the essence of in-context benign overfitting. See App. I for experiments illustrating effect of task diversity ($M$, Fig. 4 in App.), and experiments with non-isotropic $\boldsymbol{\Sigma}_x, \boldsymbol{\Sigma}_w$ (Fig. 5, 6 in App.).

## 5 CONCLUSION

We introduced *in-context benign overfitting*—a task-level analogue of classical benign overfitting, and provided sharp asymptotic characterizations showing that it arises in a minimal feature-selection model of in-context linear regression. Moving forward, rigorously establishing the Gaussian equivalence principle in our setting and extending the analysis to richer architectures (e.g., softmax attention) and other task families (e.g., Markov chains) remain natural next steps. More broadly, our results suggest a reassuring message: scaling up need not force a tradeoff between solving known tasks and generalizing to new ones, even though prior theoretical models of ICL might suggest otherwise (Lu et al., 2025; Raventos et al., 2023; Park et al., 2025) (See Fig. 3 in App. for discussion).

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

CONTENTS

# A   RELATED WORK

To deconstruct the mechanics of in-context learning, a growing body of literature has turned to controlled synthetic environments where Transformers are pre-trained from scratch on canonical function classes. Linear regression has served as a primary testbed for these investigations (Garg et al., 2022; Raventos et al., 2023; Akyürek et al., 2023; von Oswald et al., 2022; Ahn et al., 2023; Zhang et al., 2025), alongside sequence modeling tasks defined by Markov chains (Park et al., 2025; Edelman et al., 2024; Rajaraman et al., 2024; Deora et al., 2025). A central theme in this line of inquiry is algorithmic discovery: determining whether trained Transformers implement known estimation procedures, such as gradient descent variants for regression (von Oswald et al., 2022; Ahn et al., 2023; Fu et al., 2024) or induction-head mechanisms for Markov processes (Edelman et al., 2024; Rajaraman et al., 2024; Chen et al., 2024b). Additionally, a growing body of work also investigates the training dynamics of in-context learning for linear regression, specifically examining the optimization dynamics for both one-layer linear attention (Zhang et al., 2025; 2023; 2024) and softmax attention (Chen et al., 2024a).

**Retreival and Learning Modes of ICL**. Most relevant to our work are studies investigating the dual nature of ICL: the retrieval and learning modes (Pan et al., 2023). Empirical works involving transformers trained on finite task sets have identified task diversity thresholds—critical points where the model transitions from retrieval (good ID generalization) to learning (good OOD tasks) (Raventos et al., 2023; Park et al., 2025). While Wu et al. (2023) provided an initial theoretical quantification of this transition in linear attention for in-context linear regression, our analysis is most closely related to and partly inspired by Lu et al. (2025). They derive precise risk asymptotics in a proportional limit similar to ours (Eq. (1)), but with a crucial distinction: their setup is restricted to $\rho = 1$ (utilizing all features). This difference is fundamental; as we demonstrate in Section 4 (Fig. 3), fixing $\rho = 1$ couples model scale with the ambient dimension. Under this constraint, the in-context benign overfitting phenomenon disappears, highlighting the necessity of the feature-selection mechanism to decouple model capacity from data dimensionality.

Finally, we note that Frei & Vardi (2024) also coin the term *in-context benign overfitting* though in a fundamentally different context. Their work focuses on in-context binary classification and interprets *overfitting* through the lens of label noise: they show that models can fit prompts with flipped labels (noise) while still generalizing to the query. This is strictly analogous to classical benign overfitting over samples. In contrast, our framework applies the concept to the task level, describing models that overfit (memorize) the pre-training *task* distribution while generalizing to novel distributions.

# B   OVERVIEW OF THEORY

## B.1   LINEAR GAUSSIAN EQUIVALENT MODEL

Our asymptotic analysis leverages powerful machinery from high-dimensional statistics that provides sharp characterizations for solutions of *Linear Gaussian Problems* (LGPs).

**Definition 2** (Linear Gaussian MSE Problem). *An* $\mathrm{LGP}(\boldsymbol{\theta}_*, \boldsymbol{\Sigma}_g, \boldsymbol{\Sigma}_\epsilon)$ *with* $\boldsymbol{\Sigma}_g, \boldsymbol{\Sigma}_\epsilon$ *positive definite, is a linear regression problem with squared loss on* $N$ *samples* $\{(\boldsymbol{g}_i, y_i)\}_{i=1}^N$ *generated as follows: features* $\boldsymbol{g}_i \sim \mathcal{N}(\boldsymbol{0}, \mathbb{I}_{\tilde{p}})$ *and labels* $y_i = \boldsymbol{g}_i^\top \boldsymbol{\Sigma}_g^{1/2} \boldsymbol{\theta}_* + \epsilon_i$, *where* $\boldsymbol{\epsilon} = (\epsilon_1, \dots, \epsilon_N)^\top \sim \mathcal{N}(\boldsymbol{0}, \boldsymbol{\Sigma}_\epsilon)$. *Let* $\tilde{p}$ *denote the dimension of the regressor* $\boldsymbol{\theta}_* \in \mathbb{R}^{\tilde{p}}$, *and let* $\boldsymbol{G} = [\boldsymbol{g}_1, \dots, \boldsymbol{g}_N]^\top \in \mathbb{R}^{N \times \tilde{p}}$ *be the feature matrix. The estimator is defined as*

$$\hat{\boldsymbol{\theta}} = \boldsymbol{\Sigma}_g^{-1/2} \hat{\boldsymbol{a}} + \boldsymbol{\theta}_* ,$$

*with the error vector* $\hat{\boldsymbol{a}}$ *given as follows.*

- *If* $\tilde{p} < N$: *the least-squares solution*

$$\hat{\boldsymbol{a}} = \boldsymbol{a}_{\mathrm{LS}} := \arg \min_{\boldsymbol{a}} \|\boldsymbol{\epsilon} - \boldsymbol{G}\boldsymbol{a}\|^2 .$$

- *If* $\tilde{p} > N$: *the minimum-norm interpolator*

$$\hat{\boldsymbol{a}} = \boldsymbol{a}_{\mathrm{MN}} := \arg \min_{\boldsymbol{a}} \|\boldsymbol{a}\| \quad s.t. \quad \boldsymbol{G}\boldsymbol{a} = \boldsymbol{\epsilon}.$$

LGPs are central to our analysis for two reasons: (i) Well-established tools yield sharp asymptotic characterizations for the generalization error of squared-loss minimizers in LGPs; (ii) A *Gaussian equivalence principle* allows us to transfer these LGP results to our original (non-Gaussian) problem setup in Section 2. We will empirically verify the accuracy of these predictions.

The LGP solves least-squares or minimum-norm optimization as in our original setup. However, there are two key advantages in the LGP setting. First, the transition threshold between regimes is precisely $\tilde{p} = N$: as $\tilde{p}$ and $N$ grow proportionally, the Gaussian feature matrix $\boldsymbol{G}$ is almost surely full column rank if and only if $\tilde{p} < N$. Second, and most importantly, the optimization problems (LS or min-norm) for LGP involve only an isotropic Gaussian matrix and independent Gaussian noise, which admits sharp asymptotic characterizations. Instead, the original problems involve non-Gaussian features $\mathbf{h}_{\mathcal{S}_p}(X_i)$ and correlated labels $y_q^i = \boldsymbol{w}_i^\top \boldsymbol{x}_q + \epsilon$.

We now explicitly relate LGPs to our original setup of interest detailed in Section 2.

**Definition 3** (Linear Gaussian Equivalent Problem). *Consider $N$ features $\mathbf{h}_{\mathcal{S}_p}(X_i) \in \mathbb{R}^{p^2}$ as in original model with labels $y_q^i = \boldsymbol{w}_i^\top \boldsymbol{x}_q + \epsilon$ for $\boldsymbol{w}_i \sim \mathcal{D}_{task}^{train}$ and the corresponding least-squares estimates. The equivalent $\mathrm{LGP}(\boldsymbol{\theta}_*, \boldsymbol{\Sigma}_g, \boldsymbol{\Sigma}_\epsilon)$ is given by setting $\boldsymbol{\Sigma}_g \leftarrow \boldsymbol{\Sigma}_{\boldsymbol{H}_{\mathcal{S}_p}}$, $\boldsymbol{\theta}_* \leftarrow \frac{1}{N} \boldsymbol{\Sigma}_{\boldsymbol{H}_{\mathcal{S}_p}}^{-1} \boldsymbol{\sigma}_{\boldsymbol{H}_{\mathcal{S}_p} \boldsymbol{y}}$ and $\boldsymbol{\Sigma}_\epsilon \leftarrow \mathrm{diag}(\sigma_{n_1}, \ldots, \sigma_{n_N})$ with*

$$\boldsymbol{\Sigma}_{\boldsymbol{H}_{\mathcal{S}_p}} := \frac{1}{N} \mathbb{E}_{X,\epsilon}\left[ \boldsymbol{H}_{\mathcal{S}_p}^\top \boldsymbol{H}_{\mathcal{S}_p} \right] \in \mathbb{R}^{p^2 \times p^2}, \tag{2a}$$

$$\boldsymbol{\sigma}_{\boldsymbol{H}_{\mathcal{S}_p} \boldsymbol{y}} = \mathbb{E}_{X,\epsilon}[\boldsymbol{H}_{\mathcal{S}_p}^\top \boldsymbol{y}] \tag{2b}$$

$$\sigma_{n_i}^2 := \sigma_x^2 \|\boldsymbol{w}_i\|^2 + \sigma_n^2 - \boldsymbol{\theta}_\star^\top \boldsymbol{\Sigma}_{\boldsymbol{H}_{\mathcal{S}_p}} \boldsymbol{\theta}_\star \; i \in [N]. \tag{2c}$$

*See App. C for explicit formulas for the expectations above.*

Equivalence is in the strong sense that the asymptotic *empirical distribution* of the LGP estimator $\hat{\boldsymbol{\theta}}$ matches that of the original estimator $\hat{\boldsymbol{\theta}}_{\mathcal{S}}$ (Goldt et al., 2020; Hu & Lu, 2020). Thus, it suffices to characterize the former, since knowing the latter suffices to compute ID and OOD limiting risks.

The LGP parameters are chosen so that the first and second-order statistics of the LGP samples $\{(\boldsymbol{g}_i, y_i)\}_{i \in [N]}$ match the first and second-order statistics (with respect to $X, \epsilon$) of the original data $\{(\mathbf{h}_{\mathcal{S}}(X_i), y_q^i)\}_{i \in [N]}$. The principle that Gaussian systems matching a non-Gaussian system to first and second order are predictive of properties of the latter is broadly known as *universality*. Its roots lie in classical random matrix theory results on eigendistributions of random ensembles (Tao, 2023). The specific form relevant here, where the "system" refers to a random optimization problem and "properties" to quantities such as its optimal cost or empirical distribution of its solution, has been developed within the high-dimensional statistics literature, from early applications in compressed sensing (e.g. (Montanari & Nguyen, 2017; Oymak & Tropp, 2018; Panahi & Hassibi, 2017; Abbasi et al., 2019)) to more recent machine learning settings (e.g. (Hastie et al., 2019; Mei & Montanari, 2019; Goldt et al., 2020; Bosch et al., 2023; Montanari & Saeed, 2022; Hu & Lu, 2022; Ghane et al., 2024; Akhtiamov et al., 2025)). The terminology *Gaussian equivalence principle* was introduced by Mei & Montanari (2019); Goldt et al. (2020), who applied it to study linear regression with random features. Since then, numerous extensions have applied and confirmed the gaussian universality principle in various contexts, e.g., (Ghane et al., 2024; Dandi et al., 2023; Gerace et al., 2024; Pesce et al., 2023; Misiakiewicz & Saeed, 2024; Ghane et al., 2025). It is also understood that universality does not hold always and very recent work has extended the methodology to handle such cases (Wen et al., 2025). Here, we identify another instance where the Gaussian equivalence principle applies. Figure 2 empirically demonstrates this. Formalizing this equivalence rigorously is left for future work, as our focus here is on deriving predictions and establishing in-context benign overfitting; a formal proof could likely build on recent formal universality equivalence proofs such as (Hu & Lu, 2022; Montanari et al., 2023).

### B.1.1 OVERPARAMETERIZED REGIME

We begin with the overparameterized regime $\rho^2 > \nu$ (i.e., $p^2 > N$), analyzing the following minimum-norm linear Gaussian equivalent optimization:

$$\min_{\boldsymbol{a}} \frac{1}{2} \|\boldsymbol{\Sigma}_{\boldsymbol{H}_{\mathcal{S}_p}}^{-1/2} \boldsymbol{a} + \boldsymbol{\theta}_*\|^2 \quad \text{s.t.} \quad \boldsymbol{G}\boldsymbol{a} = \boldsymbol{D_w} \boldsymbol{g}', \tag{Primal}$$

where $\boldsymbol{\Sigma}_{\boldsymbol{H}_{\mathcal{S}_p}}$ and $\boldsymbol{\theta}_*$ are as in Definition 3; $\boldsymbol{D}_{\boldsymbol{w}} \in \mathbb{R}^{N \times N}$ is a diagonal matrix with entries $(\boldsymbol{D}_{\boldsymbol{w}})_{ii} = \sigma_{n_i}$ with $\sigma_{n_i}$ as given also in Definition 3; $\boldsymbol{G} \in \mathbb{R}^{N \times p^2}$ has i.i.d. standard Gaussian entries; $\boldsymbol{g}' \sim \mathcal{N}(\mathbf{0}, \mathbb{I}_N)$ is independent of $\boldsymbol{G}$. Note that $\boldsymbol{\Sigma}_{\boldsymbol{H}_{\mathcal{S}_p}}$, $\boldsymbol{\theta}_*$, and $\boldsymbol{D}_{\boldsymbol{w}}$ depend on the pretraining task vectors $\{\boldsymbol{w}_1, \ldots, \boldsymbol{w}_M\}$ sampled from $\mathcal{D}_{\text{task}}^{\text{train}}$, while $\boldsymbol{G}$ and $\boldsymbol{g}'$ are independent of these.

We will derive an asymptotic characterization for the empirical distribution of the solution to (Primal). To this end, we apply the convex Gaussian min-max theorem (CGMT) (Stojnic, 2013; Thrampoulidis et al., 2015). The application of the CGMT to analyze minimum-norm linear Gaussian problems (Definition 2) appears in various recent works, e.g., (Deng et al., 2019; Montanari et al., 2019; Loureiro et al., 2021; Chang et al., 2021; Liang & Sur, 2020). While the CGMT framework provides a general recipe, its application still requires problem-specific analysis, which we carry out here since, to the best of our knowledge, no plug-and-play result exists for our specific setting. We outline the key calculations below and defer remaining derivations to the appendix.

Our starting point is what is the following Auxiliary Optimization (AO):

$$\min_{\boldsymbol{a}} \max_{\boldsymbol{\lambda}} \ \|\boldsymbol{a}\| \boldsymbol{\lambda}^\top \mathbf{h} + \|\boldsymbol{\lambda}\| \boldsymbol{a}^\top \boldsymbol{g} - \boldsymbol{\lambda}^\top \boldsymbol{D}_{\boldsymbol{w}} \boldsymbol{g}'$$
$$+ \tfrac{1}{2} \|\boldsymbol{\Sigma}_{\boldsymbol{H}_{\mathcal{S}_p}}^{-1/2} \boldsymbol{a} + \boldsymbol{\theta}_\star\|^2 \tag{3}$$

where $\boldsymbol{g} \sim \mathcal{N}(\mathbf{0}, \mathbb{I}_{p^2})$ and $\mathbf{h} \sim \mathcal{N}(\mathbf{0}, \mathbb{I}_N)$ are independent of each other and of all other quantities. The CGMT guarantees that the optimal cost of Eq. (3) converges to the same asymptotic limit as that of (Primal), which sets the stage for establishing that the empirical distribution of the AO solution converges to the same limit as that of the primal (Montanari et al., 2019; Chang et al., 2021).

**Solving the AO.** Following standard CGMT machinery, we reduce the high-dimensional AO in Eq. (3) to a two-dimensional min-max optimization:

$$\min_{u>0} \max_{\beta \geq 0} \ \frac{\beta u}{2} + \beta \frac{\sum_{i=1}^n \sigma_{n_i}^2}{2u} + \frac{\|\boldsymbol{\theta}_\star\|^2}{2}$$
$$- \frac{1}{2}(\beta \boldsymbol{g} - \boldsymbol{\Sigma}_{\boldsymbol{H}_{\mathcal{S}_p}}^{-1/2} \boldsymbol{\theta}_\star)^\top \boldsymbol{M}^{-1} (\beta \boldsymbol{g} - \boldsymbol{\Sigma}_{\boldsymbol{H}_{\mathcal{S}_p}}^{-1/2} \boldsymbol{\theta}_\star), \tag{AO}$$

where we define the shorthand

$$\boldsymbol{M} := \boldsymbol{M}(\beta, u) := \frac{\beta N}{u} \mathbb{I}_{p^2} + \boldsymbol{\Sigma}_{\boldsymbol{H}_{\mathcal{S}_p}}^{-1}.$$

Direct differentiation yields that for any fixed $u > 0$, the objective in (AO) is *strictly* concave on $\beta > 0$ (except on a measure-zero set). Also, for any fixed $\beta > 0$, it is strictly convex in $u > 0$. This implies that the saddle point $(u_*, \beta_*)$ is unique whenever it satisfies $\beta_* > 0$.

Moreover, the same reduction that yields (AO) also shows the following closed form for $\boldsymbol{a}_*$, the minimizer of Eq. (3):

$$\boldsymbol{a}_* = \boldsymbol{a}_*(\beta_*, u_*) = \boldsymbol{M}(\beta_*, u_*)^{-1} (\beta_* \boldsymbol{g} - \boldsymbol{\Sigma}_{\boldsymbol{H}_{\mathcal{S}_p}}^{-1/2} \boldsymbol{\theta}_\star), \tag{4}$$

where $(\beta_*, u_*)$ is the unique saddle point of (AO) (when $\beta_* > 0$). In particular, conditional on this saddle point, $\boldsymbol{a}_*$ is Gaussian, with mean and covariance determined by $(\beta_*, \tau_*)$. As we show next, in the high-dimensional limit, $(\beta_*, \tau_*)$ converge to deterministic values, and thus $\boldsymbol{a}_*$ is asymptotically Gaussian with mean and covariance determined by the limiting saddle point.

**Deterministic reduction.** In the regime specified in Eq. (1), the AO objective converges to a deterministic function of the scalar parameters $(u, \beta)$. To state this limit compactly, we introduce deterministic quantities defined as the in-probability limits of the following random terms:

$$c_\infty := \text{plim}_{d \to \infty} \frac{1}{d^2} \sum_{i=1}^N \sigma_{n_i}^2, \tag{5a}$$

$$s_\infty(\bar{u}, \bar{\beta}) := \text{plim}_{d \to \infty} \frac{1}{d^2} \text{tr}(\overline{\boldsymbol{M}}^{-1}), \tag{5b}$$

$$v_\infty(\bar{u}, \bar{\beta}) := \text{plim}_{d \to \infty} \frac{1}{d} \overline{\boldsymbol{\theta}}_\star^\top \overline{\boldsymbol{\Sigma}}_{\boldsymbol{H}_{\mathcal{S}_p}}^{-1/2} \overline{\boldsymbol{M}}^{-1} \overline{\boldsymbol{\Sigma}}_{\boldsymbol{H}_{\mathcal{S}_p}}^{-1/2} \overline{\boldsymbol{\theta}}_\star, \tag{5c}$$

where $\bar{\beta}, \bar{u}, \overline{\boldsymbol{\Sigma}}_{\boldsymbol{H}_{\mathcal{S}_p}}, \overline{\boldsymbol{\theta}}_\star, \overline{\boldsymbol{M}}$ are the normalized versions of $\beta, u, \boldsymbol{\Sigma}_{\boldsymbol{H}_{\mathcal{S}_p}}, \boldsymbol{\theta}_\star, \boldsymbol{M}$, chosen so that these quantities remain $O(1)$ (see App. D.1). We prove these convergences (with respect to the randomness in $\boldsymbol{g}$ and the task vectors $\{\boldsymbol{w}_i\}_{i \in [N]}$) in App. E. With these definitions, the AO objective admits the following deterministic limit

$$D(\bar{u}, \bar{\beta}) := \frac{\bar{\beta}\bar{u}}{2} + \frac{\bar{\beta}}{2\bar{u}}c_\infty - \frac{\bar{\beta}^2}{2}s_\infty(\bar{u}, \bar{\beta}) - \frac{1}{2}v_\infty(\bar{u}, \bar{\beta}),$$

Moreover, since $D$ is strictly convex-concave on $\{\bar{u} > 0, \bar{\beta} > 0\}$, the limiting saddle point $(\bar{\beta}_*, \bar{u}_*)$ is unique and characterized by first-order optimality conditions, which, after algebraic simplification, yield:

$$\bar{u}_* = \frac{\bar{\beta}_*\nu}{a}, \quad \text{where} \quad \left(1 - \frac{\nu}{\rho^2}\right)a = m_a, \tag{6}$$

$$\bar{\beta}_*^2 = \frac{a^2(\sigma_n^2 + 1 - \rho + \rho c) + \rho(1 - a^2 c^2)m_a - \rho a z_a^2 m_a'}{\rho^2\left(1 - \frac{\nu}{\rho^2} - \frac{m_a'}{a^2}\right)}.$$

Here, $m_a := m_\gamma(z_a)$ is the Stieltjes transform of the Marchenko-Pastur law with $\gamma := p/M = \rho/\mu$ (using the scaling defined in Eq. (1)) evaluated at point $z_a := -1/a - c$. See Definition 4 in the Appendix. In addition, $m_a' := m_\gamma'(z_a)$ denotes the derivative of $m_\gamma(z)$ evaluated at $z_a$, and we have used the shorthand $c := \frac{\sigma_n^2+1}{\tau}$.

The appearance of the Marchenko–Pastur Stieltjes transform follows from the spectral structure of $\boldsymbol{\Sigma}_{\boldsymbol{H}_{\mathcal{S}_p}}$: Its explicit form (Eq. (10) in the App.) shows that $\boldsymbol{\Sigma}_{\boldsymbol{H}_{\mathcal{S}_p}}$ can be expressed in terms of blocks of the form $\mathbb{I}_p + \zeta \frac{1}{M}\sum_{i=1}^{M}\boldsymbol{w}_i[\mathcal{S}]\boldsymbol{w}_i[\mathcal{S}]^\top$ (for a scalar $\zeta$ independent of $\{\boldsymbol{w}_i\}$). Because $\boldsymbol{w}_i[\mathcal{S}]$ are Gaussian, the matrix $\frac{1}{M}\sum_{i=1}^{M}\boldsymbol{w}_i[\mathcal{S}]\boldsymbol{w}_i[\mathcal{S}]^\top$ is Wishart, whose empirical spectral distribution converges to the Marchenko–Pastur law in the proportional limit. Consequently, trace terms such as $\frac{1}{p}\text{tr}\big(\mathbb{I}_p + \zeta\frac{1}{M}\sum_{i=1}^{M}\boldsymbol{w}_i[\mathcal{S}]\boldsymbol{w}_i[\mathcal{S}]^\top\big)^{-1}$ converge to quantities given in terms of the Stieltjes transform.

**Evaluation of risk.** We are now ready to evaluate the asymptotic limit of the ID and OOD risks. Fix a task vector $\boldsymbol{w}$. We first express the per-task loss (Eq. (7)) in terms of the error vector $\boldsymbol{a}$; see Lemma 1 for the proof:

$$\mathcal{L}(\boldsymbol{a}; \boldsymbol{w}) = \underbrace{\boldsymbol{a}^\top \overline{\boldsymbol{\Sigma}}_{\boldsymbol{H}_{\mathcal{S}_p}}^{-1/2}\overline{\boldsymbol{\Sigma}}(\boldsymbol{w})\overline{\boldsymbol{\Sigma}}_{\boldsymbol{H}_{\mathcal{S}_p}}^{-1/2}\boldsymbol{a}}_{\text{Term-1(a)}} + \underbrace{\frac{1}{d}\overline{\boldsymbol{\theta}}_\star^\top \overline{\boldsymbol{\Sigma}}(\boldsymbol{w})\overline{\boldsymbol{\theta}}_\star}_{\text{Term-1(b)}}$$

$$+ 2\underbrace{\frac{1}{\sqrt{d}}\boldsymbol{a}^\top \overline{\boldsymbol{\Sigma}}_{\boldsymbol{H}_{\mathcal{S}_p}}^{-1/2}\overline{\boldsymbol{\Sigma}}(\boldsymbol{w})\overline{\boldsymbol{\theta}}_\star}_{\text{Term-1(c)}} + \frac{\|\boldsymbol{w}\|^2}{d} + \sigma_n^2$$

$$- 2\underbrace{\left(\frac{1}{\sqrt{d}}\boldsymbol{a}^\top \overline{\boldsymbol{\Sigma}}_{\boldsymbol{H}_{\mathcal{S}_p}}^{-1/2}(\boldsymbol{w} \otimes \boldsymbol{w}) + \frac{1}{d}\overline{\boldsymbol{\theta}}_\star^\top(\boldsymbol{w} \otimes \boldsymbol{w})\right)}_{\text{Term-2}}. \tag{7}$$

Here $\boldsymbol{\Sigma}(\boldsymbol{w}) := \mathbb{E}_{X, \epsilon}\big[\boldsymbol{h}_{\mathcal{S}_p}(X)\boldsymbol{h}_{\mathcal{S}_p}(X)^\top\big]$ and $\overline{\boldsymbol{\Sigma}}(\boldsymbol{w})$ denotes its normalized version; likewise, $\overline{\boldsymbol{\Sigma}}_{\boldsymbol{H}_{\mathcal{S}_p}}$ is the normalized version of $\boldsymbol{\Sigma}_{\boldsymbol{H}_{\mathcal{S}_p}}$. All normalizations are chosen so that each term in Eq. (7) is $O(1)$ under the scaling in Eq. (1).

Then, to compute the limits for the ID and OOD losses, we take the expectation of Eq. (7) over $\mathcal{D}_{\text{task}}^{\text{train}}$ and $\mathcal{D}_{\text{task}}$, respectively. We then substitute the AO's prediction for the error vector $\boldsymbol{a}$ (derived in Eq. (4)), normalized and evaluated at the deterministic optimal scalars $(\bar{\beta}_*, \bar{u}_*)$:

$$\boldsymbol{a} = \frac{1}{d}\overline{\boldsymbol{M}}(\bar{\beta}_*, \bar{u}_*)^{-1}\left(\bar{\beta}_*\boldsymbol{g} - d^{1/2}\overline{\boldsymbol{\Sigma}}_{\boldsymbol{H}_{\mathcal{S}_p}}^{-1/2}\overline{\boldsymbol{\theta}}_\star\right), \tag{8}$$

where $\boldsymbol{g} \sim \mathcal{N}(\boldsymbol{0}, \mathbb{I}_{p^2})$ is independent of the environment variables $\overline{\boldsymbol{\Sigma}}_{\boldsymbol{H}_{\mathcal{S}_p}}, \overline{\boldsymbol{\theta}}_\star, \{\sigma_{n_i}\}_{i=1}^N$. Finally, we derive closed-form limits for each component of Eq. (7) using Eq. (8). Detailed computations for the OOD loss limits (Terms 1(a)-(c) and Term 2) are provided in Lemmas 4 and 5, and the corresponding ID loss limits are derived in Lemmas 6 and 7.

With these we arrive at our first main result.

**Theorem 3** (Asymptotic Risk of Minimum-Norm LGP). *Consider the minimum norm estimator $\hat{\boldsymbol{\theta}}_{\mathcal{S}_p}$ for the linear Gaussian equivalent problem in Definition 3 under the overparameterized regime ($\nu < \rho^2$). Under the asymptotic scaling defined in Eq. (1), as $d \to \infty$, the ID and OOD risks converge in probability to the following deterministic limits:*

$$\mathcal{L}_{OOD}(\hat{\boldsymbol{\theta}}_{\mathcal{S}_p}) \xrightarrow{P} \left(1 - \tfrac{\nu}{\rho^2}\right) \rho(1+ac)(1-c(1+a+ac))$$
$$+ (1 + a + ac)(\sigma_n^2 + \rho c + 1 - \rho),$$
$$\mathcal{L}_{ID}(\hat{\boldsymbol{\theta}}_{\mathcal{S}_p}) \xrightarrow{P} \frac{\nu}{a^2}\bar{\beta}_*^2,$$

*where $c := \frac{\sigma_n^2+1}{\tau}$ and the scalars $a$ and $\bar{\beta}_*$ are the unique positive solutions to Eq. (6).*

### B.1.2 UNDERPARAMETERIZED REGIME

In the underparameterized regime (where $\nu > \rho^2$), the estimator $\hat{\boldsymbol{\theta}}_{\mathcal{S}_p}$ is the unique least-squares solution. We analyze this setting using the same framework applied in Appendix B.1.1 for the overparameterized case. Since the derivation is analogous (in fact, considerably simpler due to the strong convexity of the objective and simpler analysis of the AO) we defer the intermediate steps to App. F and present the final limits directly. The asymptotic behavior in this regime is governed by a scalar $\kappa_\infty$, which arises as the high-dimensional limit of the dual variable in the auxiliary optimization (recall Eq. (5)):

$$\kappa_\infty := \sqrt{\frac{c_\infty}{\nu\left(\frac{\nu}{\rho^2} - 1\right)}}. \tag{9}$$

**Theorem 4** (Asymptotic Risk of LS LGP). *Consider the least-squares estimator $\hat{\boldsymbol{\theta}}_{\mathcal{S}_p}$ for the linear Gaussian equivalent problem in the underparameterized regime ($\nu > \rho^2$). Under the asymptotic scaling in Eq. (1), as $d \to \infty$, the ID and OOD risks converge in probability to:*

$$\mathcal{L}_{OOD}(\hat{\boldsymbol{\theta}}_{\mathcal{S}_p}) \xrightarrow{P} m_c\big(\kappa_\infty^2(1+c) - 2c^2\rho\big) + (1+c)\rho c^2 m_c'$$
$$+ \sigma_n^2 + 1 - \rho + \rho c,$$
$$\mathcal{L}_{ID}(\hat{\boldsymbol{\theta}}_{\mathcal{S}_p}) \xrightarrow{P} \frac{\nu}{\rho^2}\kappa_\infty^2,$$

*where $\kappa_\infty$ is defined in Eq. (9), $c := \frac{\sigma_n^2+1}{\tau}$, and $m_c := m_\gamma(z_c)$ is the Stieltjes transform of the Marchenko-Pastur law with parameter $\gamma = \rho/\mu$ evaluated at $z_c = -c$. In addition, $m_c'$ denotes the derivative of $m_\gamma(\cdot)$ at $z_c$.*

## C GAUSSIAN EQUIVALENT DATA MODEL

Here, we discuss the moment-matching process for the first and second moments involving features $\boldsymbol{g}$ and labels $y$ in Definition 3 with those from the original data model discussed in Section 2. Recall that for the Gaussian equivalent data model, the features are i.i.d $\boldsymbol{g}_i \sim \mathcal{N}(\boldsymbol{0}, \boldsymbol{\Sigma}_{\boldsymbol{H}_{\mathcal{S}_p}})$ for all $i \in [N]$, where $\boldsymbol{\Sigma}_{\boldsymbol{H}_{\mathcal{S}_p}}$ matches the second moment of the original feature matrix $\boldsymbol{H}_{\mathcal{S}_p}$ by definition as $\boldsymbol{\Sigma}_{\boldsymbol{H}_{\mathcal{S}_p}} := \frac{1}{N}\mathbb{E}\big[\boldsymbol{H}_{\mathcal{S}_p}^\top \boldsymbol{H}_{\mathcal{S}_p}\big]$. Below, we show that the first moments also match, *i.e.*, $\mathbb{E}[\boldsymbol{g}] = \mathbb{E}[\mathbf{h}_{\mathcal{S}_p}(X)]$.

We compute the first and second moments of feature matrix $\boldsymbol{H}_{\mathcal{S}_p}$ by finding the expected value of each feature row. By definition, we have

$$\mathbb{E}[\mathbf{h}_{\mathcal{S}_p}(X)] = \mathbb{E}[\mathtt{vec}(\boldsymbol{x}_q[\mathcal{S}]\hat{\boldsymbol{w}}_{\mathrm{avg}}[\mathcal{S}])^\top] = \mathbb{E}[\hat{\boldsymbol{w}}_{\mathrm{avg}}[\mathcal{S}]] \otimes \mathbb{E}[\boldsymbol{x}_q[\mathcal{S}]] = 0,$$

where the second equality uses the independence of the query vector from the the rest of the context, and the third equality uses $\boldsymbol{x}_q \sim \mathcal{N}(\boldsymbol{0}, \sigma_x^2\mathbb{I}_d)$.

Next, we compute the second moment of $\boldsymbol{H}_{\mathcal{S}_p}$. We have

$$\boldsymbol{\Sigma}_{\boldsymbol{H}_{\mathcal{S}_p}} := \frac{1}{n}\mathbb{E}\left[\boldsymbol{H}_{\mathcal{S}_p}^\top \boldsymbol{H}_{\mathcal{S}_p}\right] = \frac{1}{n}\sum_{i=1}^n \mathbb{E}[\mathbf{h}_i \mathbf{h}_i^\top]$$

$$= \frac{\sigma_x^4}{TM}\sum_{i=1}^M \left[(\sigma_x^2\|\boldsymbol{w}_i\|^2 + \sigma_n^2)\mathbb{I}_{p^2} + \sigma_x^2(T+1)(\boldsymbol{w}_i[\mathcal{S}]\boldsymbol{w}_i[\mathcal{S}]^\top) \otimes \mathbb{I}_p\right]. \tag{10}$$

Here, the first equality uses the independence of the rows of $\boldsymbol{H}_{\mathcal{S}_p}$, and the second equality follows by using Lemma 16.

Recall that the labels in the Gaussian equivalent problem are generated as $y_i = \boldsymbol{g}_i^\top \boldsymbol{\theta}_\star + \epsilon_i$, where we select $\boldsymbol{\theta}_\star$ and noise $\epsilon_i$ such that the moments, $\mathbb{E}[y]$, $\boldsymbol{\sigma}_{\boldsymbol{H}_{\mathcal{S}_p}y} := \mathbb{E}[\boldsymbol{H}_{\mathcal{S}_p}^\top \boldsymbol{y}]$ and $\mathbb{E}[\boldsymbol{y}\boldsymbol{y}^\top]$ also match with that of the original problem.

First, in the original model, $\mathbb{E}[y] = 0$. In the Gaussian equivalent data model, $\mathbb{E}[y] = \mathbb{E}[\boldsymbol{g}]^\top \boldsymbol{\theta}_\star + \mathbb{E}[\epsilon] = \mathbb{E}[\epsilon]$. To match, we require $\mathbb{E}[\epsilon] = 0$.

Next, as the labels $y$ are independent, $\mathbb{E}[\boldsymbol{y}\boldsymbol{y}^\top]$ is a diagonal matrix, and we only need to look at the entries $\mathbb{E}[y^2]$.

In the original model, we have

$$\mathbb{E}[y^2] = \mathbb{E}[(\boldsymbol{w}^\top \boldsymbol{x} + \epsilon)^2] = \sigma_x^2\|\boldsymbol{w}\|^2 + \sigma_n^2.$$

For the Gaussian equivalent data model,

$$\mathbb{E}[y^2] = \boldsymbol{\theta}_\star^\top \boldsymbol{\Sigma}_{\boldsymbol{H}_{\mathcal{S}_p}} \boldsymbol{\theta}_\star + \mathbb{E}[\epsilon^2].$$

Therefore, for the Gaussian equivalent data model $\epsilon_i \sim \mathcal{N}(0, \sigma_{n_i}^2)$, where

$$\sigma_{n_i}^2 = \sigma_x^2\|\boldsymbol{w}_i\|^2 + \sigma_n^2 - \boldsymbol{\theta}_\star^\top \boldsymbol{\Sigma}_{\boldsymbol{H}_{\mathcal{S}_p}} \boldsymbol{\theta}_\star =: d_i^2. \tag{11}$$

Here, $\boldsymbol{w}_i$ corresponds to the task vector for sample $i \in [n]$.

Recall that $\boldsymbol{G} \in \mathbb{R}^{N \times p^2}$ has i.i.d. entries $G_{ij} \sim \mathcal{N}(0, 1)$, so that $\boldsymbol{G}\boldsymbol{\Sigma}_{\boldsymbol{H}_{\mathcal{S}_p}}^{1/2}$ has i.i.d. rows distributed as $\mathcal{N}(0, \boldsymbol{\Sigma}_{\boldsymbol{H}_{\mathcal{S}_p}})$. Therefore, the remaining second moment-matching condition gives

$$\boldsymbol{\sigma}_{\boldsymbol{H}_{\mathcal{S}_p}y} = \mathbb{E}[\boldsymbol{\Sigma}_{\boldsymbol{H}_{\mathcal{S}_p}}^{1/2}\boldsymbol{G}^\top \boldsymbol{G}\boldsymbol{\Sigma}_{\boldsymbol{H}_{\mathcal{S}_p}}^{1/2}\boldsymbol{\theta}_\star] = \boldsymbol{\Sigma}_{\boldsymbol{H}_{\mathcal{S}_p}}^{1/2}\mathbb{E}[\boldsymbol{G}^\top \boldsymbol{G}]\boldsymbol{\Sigma}_{\boldsymbol{H}_{\mathcal{S}_p}}^{1/2}\boldsymbol{\theta}_\star = N\boldsymbol{\Sigma}_{\boldsymbol{H}_{\mathcal{S}_p}}\boldsymbol{\theta}_\star,$$

which implies

$$\boldsymbol{\theta}_\star = \frac{1}{N}\boldsymbol{\Sigma}_{\boldsymbol{H}_{\mathcal{S}_p}}^{-1}\boldsymbol{\sigma}_{\boldsymbol{H}_{\mathcal{S}_p}y}. \tag{12}$$

Here, using Lemma 16, we have

$$\boldsymbol{\sigma}_{\boldsymbol{H}_{\mathcal{S}_p}y} = \mathbb{E}\left[\sum_{i=1}^n \mathbf{h}_i y_i\right] = \sum_{i=1}^M \frac{n}{M}\mathbb{E}[\mathbf{h}_i y_i]$$

$$= \frac{n}{M}\sum_{i=1}^M \sigma_x^4 \boldsymbol{w}_i[\mathcal{S}] \otimes \boldsymbol{w}_i[\mathcal{S}]. \tag{13}$$

## D  FORMING THE AUXILIARY OPTIMIZATION PROBLEM

**Min norm using Gaussian equivalent.**  First, recall the minimum norm defined using the Gaussian equivalent data. We have

$$\min_{\boldsymbol{\theta}} \ \tfrac{1}{2}\|\boldsymbol{\theta}\|^2 \ \text{subject to} \ \ \boldsymbol{G}(\boldsymbol{\theta} - \boldsymbol{\theta}_\star) = \boldsymbol{D_w}\boldsymbol{g}', \tag{14}$$

where $\boldsymbol{D_w}$ is a diagonal matrix with the $i^{\text{th}}$ entry equal to $\sigma_{n_i}$ for sample $i \in [n]$. Lastly, we use $\boldsymbol{g}' \sim \mathcal{N}(\boldsymbol{0}, \mathbb{I})$ to denote noise that is independent of entries in $\boldsymbol{G} \in \mathbb{R}^{N \times p^2}$. Here, the rows of $\boldsymbol{G}$, $\boldsymbol{g}_i \sim \mathcal{N}(\boldsymbol{0}, \boldsymbol{\Sigma}_{\boldsymbol{H}_{\mathcal{S}_p}})$ are i.i.d sampled for all $i \in [N]$.

Changing variable

$$\Sigma_{H_{\mathcal{S}_p}}^{1/2}(\boldsymbol{\theta} - \boldsymbol{\theta}_\star) =: \boldsymbol{a}, \tag{15}$$

and substituting $\boldsymbol{G} \leftarrow \boldsymbol{G}\Sigma_{H_{\mathcal{S}_p}}^{1/2}$, where $\boldsymbol{G}$ now refers to entries from unit variance isotropic Gaussian, Eq. (14) is now

$$\min_{\boldsymbol{a}} \ \tfrac{1}{2}\|\Sigma_{H_{\mathcal{S}_p}}^{-1/2}\boldsymbol{a} + \boldsymbol{\theta}_\star\|^2 \quad \text{subject to} \quad \boldsymbol{G}\boldsymbol{a} = \boldsymbol{D_w}\boldsymbol{g}'.$$

Using Lagrangian formulation, the solution to the above problem is the same as the one to the following unconstrained min-max problem.

$$\min_{\boldsymbol{a}} \max_{\boldsymbol{\lambda}} \boldsymbol{\lambda}^\top \boldsymbol{G}\boldsymbol{a} - \boldsymbol{\lambda}^\top \boldsymbol{D_w}\boldsymbol{g}' + \tfrac{1}{2}\|\Sigma_{H_{\mathcal{S}_p}}^{-1/2}\boldsymbol{a} + \boldsymbol{\theta}_\star\|^2.$$

Applying the CGMT Thrampoulidis et al. (2015; 2018), the corresponding Auxiliary Optimization (AO) is

$$\min_{\boldsymbol{a}} \max_{\boldsymbol{\lambda}} \ \|\boldsymbol{a}\|\boldsymbol{\lambda}^\top \mathbf{h} + \|\boldsymbol{\lambda}\|\boldsymbol{a}^\top \boldsymbol{g} - \boldsymbol{\lambda}^\top \boldsymbol{D_w}\boldsymbol{g}' + \tfrac{1}{2}\|\Sigma_{H_{\mathcal{S}_p}}^{-1/2}\boldsymbol{a} + \boldsymbol{\theta}_\star\|^2$$

Let $\beta = \|\boldsymbol{\lambda}\|$, and $\hat{\boldsymbol{\lambda}}$, be the corresponding unit norm direction, then the above can be written as

$$\min_{\boldsymbol{a}} \max_{\hat{\boldsymbol{\lambda}},\beta \geq 0} \beta\hat{\boldsymbol{\lambda}}^\top(\|\boldsymbol{a}\|\mathbf{h} - \boldsymbol{D_w}\boldsymbol{g}') + \beta\boldsymbol{a}^\top \boldsymbol{g} + \tfrac{1}{2}\|\Sigma_{H_{\mathcal{S}_p}}^{-1/2}\boldsymbol{a} + \boldsymbol{\theta}_\star\|^2$$

$$= \min_{\boldsymbol{a}} \max_{\beta \geq 0} \beta\|(\|\boldsymbol{a}\|\mathbf{h} - \boldsymbol{D_w}\boldsymbol{g}')\| + \beta\boldsymbol{a}^\top \boldsymbol{g} + \tfrac{1}{2}\|\Sigma_{H_{\mathcal{S}_p}}^{-1/2}\boldsymbol{a} + \boldsymbol{\theta}_\star\|^2,$$

where the final step uses the fact that the objective maximizes over $\hat{\boldsymbol{\lambda}}$ when $\hat{\boldsymbol{\lambda}} = \frac{\|\boldsymbol{a}\|\mathbf{h} - \boldsymbol{D_w}\boldsymbol{g}'}{\|\|\boldsymbol{a}\|\mathbf{h} - \boldsymbol{D_w}\boldsymbol{g}'\|}$.

The objective is convex in $\boldsymbol{a}$ and linear, hence concave, in $\beta$. As shown in previous works, e.g. (Chang et al., 2021), the constraint sets can be restricted to compact subsets, hence using Sion's minimax theorem we can swap the order of the optimization to min-max to $\max_\beta \min_{\boldsymbol{a}}$:

$$\max_{\beta \geq 0} \min_{\boldsymbol{a}} \beta\|(\|\boldsymbol{a}\|\mathbf{h} - \boldsymbol{D_w}\boldsymbol{g}')\| + \beta\boldsymbol{a}^\top \boldsymbol{g} + \tfrac{1}{2}\|\Sigma_{H_{\mathcal{S}_p}}^{-1/2}\boldsymbol{a} + \boldsymbol{\theta}_\star\|^2$$

Next, using the variational characterization of the sqrt function (again, see for example (Chang et al., 2021, Sec. B.4)), the above optimization is

$$\max_{\beta \geq 0} \min_{\boldsymbol{a},u>0} \frac{\beta u}{2} + \beta\frac{(\|\boldsymbol{a}\|\mathbf{h} - \boldsymbol{D_w}\boldsymbol{g}')^2}{2u} + \beta\boldsymbol{a}^\top \boldsymbol{g} + \tfrac{1}{2}\|\Sigma_{H_{\mathcal{S}_p}}^{-1/2}\boldsymbol{a} + \boldsymbol{\theta}_\star\|^2 \tag{16}$$

Next, as $n \to \infty$, we have $\frac{\|\mathbf{h}\|^2}{n} \to 1$ and $\frac{\mathbf{h}^\top \boldsymbol{D_w}\boldsymbol{g}'}{n} \to 0$, and $\frac{\|\boldsymbol{D_w}\boldsymbol{g}'\|^2}{n} - \frac{1}{n}\sum_{i=1}^n d_i^2 \to 0$ in probability. Here and onwards, we denote $d_i = \sigma_{n_i}$. Replacing these random quantities in Eq. (16) yields the following asymptotically equivalent optimization wpa1:

$$\max_{\beta \geq 0} \min_{\boldsymbol{a},u>0} \frac{\beta u}{2} + \beta\frac{n\|\boldsymbol{a}\|^2 + \sum_{i=1}^n d_i^2}{2u} + \beta\boldsymbol{a}^\top \boldsymbol{g} + \tfrac{1}{2}\|\Sigma_{H_{\mathcal{S}_p}}^{-1/2}\boldsymbol{a} + \boldsymbol{\theta}_\star\|^2 \tag{17}$$

Notice, we have a quadratic in $\boldsymbol{a}$. Minimizing the above w.r.t. $\boldsymbol{a}$, the optimal is at

$$\boldsymbol{a}_* := \boldsymbol{a}_*(\beta,u) = \boldsymbol{M}^{-1}(\beta\boldsymbol{g} - \Sigma_{H_{\mathcal{S}_p}}^{-1/2}\boldsymbol{\theta}_\star), \quad \boldsymbol{M} := \frac{\beta n}{u}\mathbb{I} + \Sigma_{H_{\mathcal{S}_p}}^{-1}. \tag{18}$$

Plugging this back in Eq. (17), it simplifies to

$$\max_{\beta \geq 0} \min_{u>0} \underbrace{\frac{\beta u}{2} + \beta\frac{\sum_{i=1}^n d_i^2}{2u} - \tfrac{1}{2}(\beta\boldsymbol{g} - \Sigma_{H_{\mathcal{S}_p}}^{-1/2}\boldsymbol{\theta}_\star)^\top \boldsymbol{M}^{-1}(\beta\boldsymbol{g} - \Sigma_{H_{\mathcal{S}_p}}^{-1/2}\boldsymbol{\theta}_\star) + \frac{\|\boldsymbol{\theta}_\star\|^2}{2}}_{R(u,\beta)} \tag{AO}$$

### D.1 SOLVING THE AUXILIARY OPTIMIZATION.

The subsequent analysis will use the saddle point $(\bar{u}_\star, \bar{\beta}_\star)$ of the deterministic limit objective defined below. Formally relating $(\bar{u}_\star, \bar{\beta}_\star)$ to the finite-dimensional saddle point requires a uniform convergence argument for the AO objective for which we refer the reader to Chang et al. (2021, Sec. B.4). We treat this as a standard technical condition and proceed.

**Choice of scaling.** We next identify the natural normalization under the asymptotic regime. For our setup, note that $\|\boldsymbol{\theta}_\star\| = \Theta(d^{3/2})$, $\lambda_i = \Theta(1/d^3)$ for all $i \in [p^2]$, where $\lambda_i$ denote the Eigen values of $\boldsymbol{\Sigma}_{\boldsymbol{H}_{\mathcal{S}_p}}$. Moreover, $\tilde{\lambda}_i = \Theta(d^3)$, where $\tilde{\lambda}_i$ denote the Eigen values of $\mathbf{M}$. Using this, and Eqs. (1), (10), (12), (13) and (18), we define the following rescaled quantities:

$$\bar{u} := u/d, \quad \bar{\beta} := \beta/d^2, \tag{19}$$

$$\overline{\boldsymbol{\Sigma}}_{\boldsymbol{H}_{\mathcal{S}_p}} := d^3 \boldsymbol{\Sigma}_{\boldsymbol{H}_{\mathcal{S}_p}} = \frac{1}{\tau}\left(\sigma_n^2 + \frac{1}{M}\sum_{i=1}^{M}\frac{\|\boldsymbol{w}_i\|^2}{d}\right)\mathbb{I}_{p^2} + \frac{1}{M}\sum_{i=1}^{M}(\boldsymbol{w}_i[\mathcal{S}]\boldsymbol{w}_i[\mathcal{S}]^\top)\otimes\mathbb{I}_p, \tag{20}$$

$$\overline{\boldsymbol{M}} := \frac{\boldsymbol{M}}{d^3} = \frac{\bar{\beta}\nu}{\bar{u}}\mathbb{I} + \overline{\boldsymbol{\Sigma}}_{\boldsymbol{H}_{\mathcal{S}_p}}^{-1} \tag{21}$$

$$\overline{\boldsymbol{\theta}}_\star := \frac{\boldsymbol{\theta}_\star}{d} = \frac{1}{dn}\boldsymbol{\Sigma}_{\boldsymbol{H}_{\mathcal{S}_p}}^{-1}\boldsymbol{\sigma}_{\boldsymbol{H}_{\mathcal{S}_p}\boldsymbol{y}} = \overline{\boldsymbol{\Sigma}}_{\boldsymbol{H}_{\mathcal{S}_p}}^{-1}\left(\frac{1}{M}\sum_{i=1}^{M}\boldsymbol{w}_i[\mathcal{S}]\otimes\boldsymbol{w}_i[\mathcal{S}]\right). \tag{22}$$

Next, we normalize the auxiliary objective (AO) by $d^3$, and get the following objective

$$D_d(u,\beta) := \frac{1}{d^3}R(u,\beta)$$

$$= \bar{\beta}\bar{u} + \frac{\bar{\beta}}{2\bar{u}}\frac{1}{d^2}\sum_{i=1}^{n}d_i^2 - \frac{1}{2d^2}\bar{\beta}^2\boldsymbol{g}^\top\overline{\boldsymbol{M}}^{-1}\boldsymbol{g} - \frac{1}{2d}\overline{\boldsymbol{\theta}}_\star^\top\overline{\boldsymbol{\Sigma}}_{\boldsymbol{H}_{\mathcal{S}_p}}^{-1/2}\overline{\boldsymbol{M}}^{-1}\overline{\boldsymbol{\Sigma}}_{\boldsymbol{H}_{\mathcal{S}_p}}^{-1/2}\overline{\boldsymbol{\theta}}_\star + \frac{1}{d^{7/2}}\boldsymbol{g}^\top\overline{\boldsymbol{M}}^{-1}\overline{\boldsymbol{\Sigma}}_{\boldsymbol{H}_{\mathcal{S}_p}}^{-1/2}\overline{\boldsymbol{\theta}}_\star + \frac{\|\overline{\boldsymbol{\theta}}_\star\|^2}{2d}.$$

Using Lemma 14, it follows that

$$\lim_{d\to\infty}\frac{1}{2d^2}\boldsymbol{g}^\top\overline{\boldsymbol{M}}^{-1}\boldsymbol{g} = \lim_{d\to\infty}\frac{1}{d^2}\operatorname{tr}(\overline{\boldsymbol{M}}^{-1}) =: s_\infty(\bar{u},\bar{\beta}), \qquad \lim_{d\to\infty}\frac{1}{d^{7/2}}\boldsymbol{g}^\top\overline{\boldsymbol{M}}^{-1}\overline{\boldsymbol{\Sigma}}_{\boldsymbol{H}_{\mathcal{S}_p}}^{-1/2}\overline{\boldsymbol{\theta}}_\star = 0. \tag{23}$$

Further, define the deterministic limits

$$c_\infty := \lim_{d\to\infty}\frac{1}{d^2}\sum_{i=1}^{n}d_i^2, \quad v_\infty(\bar{u},\bar{\beta}) := \frac{1}{d}\overline{\boldsymbol{\theta}}_\star^\top\overline{\boldsymbol{\Sigma}}_{\boldsymbol{H}_{\mathcal{S}_p}}^{-1/2}\overline{\boldsymbol{M}}^{-1}\overline{\boldsymbol{\Sigma}}_{\boldsymbol{H}_{\mathcal{S}_p}}^{-1/2}\overline{\boldsymbol{\theta}}_\star. \tag{24}$$

Using Eq. (23), Eq. (24) we have the following deterministic limit of $D_d(\tau,\beta)$

$$D(\bar{u},\bar{\beta}) = \frac{\bar{\beta}\bar{u}}{2} + \frac{\bar{\beta}}{2\bar{u}}c_\infty - \frac{\bar{\beta}^2}{2}s_\infty(\bar{u},\bar{\beta}) - \frac{1}{2}v_\infty(\bar{u},\bar{\beta}). \tag{25}$$

Here, we first compute $\lim_{d\to\infty}D_d(u,\beta)$, and then, since $\lim_{d\to\infty}\frac{\|\overline{\boldsymbol{\theta}}_\star\|^2}{2d} = 0.5$, we omit it as it is independent of $\bar{\beta}, \bar{u}$.

### D.2 SIMPLIFYING PER-TASK LOSS

**Lemma 1.** *The per-task loss evaluated at fixed $\boldsymbol{w}$ with respect to the error vector $\boldsymbol{a}$ is given by.*

$$\mathcal{L}(\boldsymbol{a};\boldsymbol{w}) = \underbrace{\boldsymbol{a}^\top\overline{\boldsymbol{\Sigma}}_{\boldsymbol{H}_{\mathcal{S}_p}}^{-1/2}\overline{\boldsymbol{\Sigma}}(\boldsymbol{w})\overline{\boldsymbol{\Sigma}}_{\boldsymbol{H}_{\mathcal{S}_p}}^{-1/2}\boldsymbol{a}}_{\textit{Term-1(a)}} + \underbrace{\frac{1}{d}\overline{\boldsymbol{\theta}}_\star^\top\overline{\boldsymbol{\Sigma}}(\boldsymbol{w})\overline{\boldsymbol{\theta}}_\star}_{\textit{Term-1(b)}} + 2\underbrace{\frac{1}{\sqrt{d}}\boldsymbol{a}^\top\overline{\boldsymbol{\Sigma}}_{\boldsymbol{H}_{\mathcal{S}_p}}^{-1/2}\overline{\boldsymbol{\Sigma}}(\boldsymbol{w})\overline{\boldsymbol{\theta}}_\star}_{\textit{Term-1(c)}}$$

$$- 2\underbrace{\left(\frac{1}{d^{1/2}}\boldsymbol{a}^\top\overline{\boldsymbol{\Sigma}}_{\boldsymbol{H}_{\mathcal{S}_p}}^{-1/2}(\boldsymbol{w}\otimes\boldsymbol{w}) + \frac{1}{d}\overline{\boldsymbol{\theta}}_\star^\top(\boldsymbol{w}\otimes\boldsymbol{w})\right)}_{\textit{Term-2}} + \frac{\|\boldsymbol{w}\|^2}{d} + \sigma_n^2.$$

*Proof.*

$$\mathcal{L}(\hat{\boldsymbol{\theta}}_{\mathcal{S}_p};\boldsymbol{w}) = \mathbb{E}_{X,\epsilon}[\hat{\boldsymbol{\theta}}_{\mathcal{S}_p}^\top\mathbf{h}_{\mathcal{S}_p}(X) - y_q]^2.$$

$$= \hat{\boldsymbol{\theta}}_{\mathcal{S}_p}^\top\mathbb{E}_{X,\epsilon}[\mathbf{h}_{\mathcal{S}_p}(X)\mathbf{h}_{\mathcal{S}_p}(X)^\top]\hat{\boldsymbol{\theta}}_{\mathcal{S}_p} + \mathbb{E}_{X,\epsilon}[y_q^2] - 2\hat{\boldsymbol{\theta}}_{\mathcal{S}_p}^\top\mathbb{E}_{X,y}[\mathbf{h}_{\mathcal{S}_p}(X)y_q]$$

$$= \underbrace{\hat{\boldsymbol{\theta}}_{\mathcal{S}_p}^\top\mathbb{E}_{X,\epsilon}[\mathbf{h}_{\mathcal{S}_p}(X)\mathbf{h}_{\mathcal{S}_p}(X)^\top]\hat{\boldsymbol{\theta}}_{\mathcal{S}_p}}_{\text{Term-1}} + \sigma_x^2\|\boldsymbol{w}\|^2 + \sigma_n^2 - 2\underbrace{\hat{\boldsymbol{\theta}}_{\mathcal{S}_p}^\top\mathbb{E}_{X,y}[\mathbf{h}_{\mathcal{S}_p}(X)y_q]}_{\text{Term-2}}. \tag{26}$$

Let $\mathbf{\Sigma}(\boldsymbol{w}) := \mathbb{E}_{X,\epsilon}[\mathbf{h}_{\mathcal{S}_p}(X)\mathbf{h}_{\mathcal{S}_p}(X)^\top]$. Then, using Lemma 16 and Eq. (1), define

$$\overline{\mathbf{\Sigma}}(\boldsymbol{w}) := d^3\mathbf{\Sigma}(\boldsymbol{w}) = \frac{1}{\tau}\left(\frac{\|\boldsymbol{w}\|^2}{d} + \sigma_n^2\right)\mathbb{I}_{p^2} + \left(1 + \frac{1}{\tau d}\right)(\boldsymbol{w}[\mathcal{S}]\boldsymbol{w}[\mathcal{S}]^\top)\otimes\mathbb{I}_p.$$

To simplify Term-1 and Term-2, we use $\hat{\boldsymbol{\theta}}_{\mathcal{S}_p} = \mathbf{\Sigma}_{\boldsymbol{H}_{\mathcal{S}_p}}^{-1/2}\boldsymbol{a} + \boldsymbol{\theta}_\star$ from Eq. (15), and Lemma 16.

First, we have Term-1:

$$(\mathbf{\Sigma}_{\boldsymbol{H}_{\mathcal{S}_p}}^{-1/2}\boldsymbol{a} + \boldsymbol{\theta}_\star)^\top\mathbf{\Sigma}(\boldsymbol{w})(\mathbf{\Sigma}_{\boldsymbol{H}_{\mathcal{S}_p}}^{-1/2}\boldsymbol{a} + \boldsymbol{\theta}_\star) = \boldsymbol{a}^\top\mathbf{\Sigma}_{\boldsymbol{H}_{\mathcal{S}_p}}^{-1/2}\mathbf{\Sigma}(\boldsymbol{w})\mathbf{\Sigma}_{\boldsymbol{H}_{\mathcal{S}_p}}^{-1/2}\boldsymbol{a} + \boldsymbol{\theta}_\star^\top\mathbf{\Sigma}(\boldsymbol{w})\boldsymbol{\theta}_\star + 2\boldsymbol{a}^\top\mathbf{\Sigma}_{\boldsymbol{H}_{\mathcal{S}_p}}^{-1/2}\mathbf{\Sigma}(\boldsymbol{w})\boldsymbol{\theta}_\star$$

$$= \underbrace{\boldsymbol{a}^\top\overline{\mathbf{\Sigma}}_{\boldsymbol{H}_{\mathcal{S}_p}}^{-1/2}\overline{\mathbf{\Sigma}}(\boldsymbol{w})\overline{\mathbf{\Sigma}}_{\boldsymbol{H}_{\mathcal{S}_p}}^{-1/2}\boldsymbol{a}}_{\text{Term-1(a)}} + \underbrace{\frac{1}{d}\overline{\boldsymbol{\theta}}_\star^\top\overline{\mathbf{\Sigma}}(\boldsymbol{w})\overline{\boldsymbol{\theta}}_\star}_{\text{Term-1(b)}} + 2\underbrace{\frac{1}{\sqrt{d}}\boldsymbol{a}^\top\overline{\mathbf{\Sigma}}_{\boldsymbol{H}_{\mathcal{S}_p}}^{-1/2}\overline{\mathbf{\Sigma}}(\boldsymbol{w})\overline{\boldsymbol{\theta}}_\star}_{\text{Term-1(c)}}. \quad (27)$$

Next, we have Term-2:

$$\hat{\boldsymbol{\theta}}_{\mathcal{S}_p}^\top\mathbb{E}_{X,y}[\mathbf{h}_{\mathcal{S}_p}(X)y_q] = (\mathbf{\Sigma}_{\boldsymbol{H}_{\mathcal{S}_p}}^{-1/2}\boldsymbol{a} + \boldsymbol{\theta}_\star)^\top(\sigma_x^4\boldsymbol{w}[\mathcal{S}]\otimes\boldsymbol{w}[\mathcal{S}])$$

$$= \sigma_x^4\boldsymbol{a}^\top\mathbf{\Sigma}_{\boldsymbol{H}_{\mathcal{S}_p}}^{-1/2}\boldsymbol{w}\otimes\boldsymbol{w} + \sigma_x^4\boldsymbol{\theta}_\star^\top\boldsymbol{w}\otimes\boldsymbol{w}$$

$$= \frac{1}{d^2}\boldsymbol{a}^\top\mathbf{\Sigma}_{\boldsymbol{H}_{\mathcal{S}_p}}^{-1/2}\boldsymbol{w}\otimes\boldsymbol{w} + \frac{1}{d^2}\boldsymbol{\theta}_\star^\top\boldsymbol{w}\otimes\boldsymbol{w}$$

$$= \frac{1}{d^{1/2}}\boldsymbol{a}^\top\overline{\mathbf{\Sigma}}_{\boldsymbol{H}_{\mathcal{S}_p}}^{-1/2}(\boldsymbol{w}\otimes\boldsymbol{w}) + \frac{1}{d}\overline{\boldsymbol{\theta}}_\star^\top(\boldsymbol{w}\otimes\boldsymbol{w}). \quad (28)$$

Substituting Eqs. (27) and (28) in Eq. (26) finishes the proof. $\qquad\square$

# E  PROOFS FOR MIN NORM AO

## E.1  WISHART MATRIX ASYMPTOTICS

**Definition 4.** *Let $\mu$ be a probability measure on $\mathbb{R}$. The Stieltjes transform of $\mu$, denoted by $m_\mu(z)$, is a function defined for all complex numbers $z \in \mathbb{C} \setminus \text{supp}(\mu)$ by:*

$$m_\mu(z) = \int_\mathbb{R} \frac{1}{x-z}\,d\mu(x).$$

**Remark 1** (Stieltjes transform of the Marchenko-Pastur law). *In the asymptotic regime where $p, M \to \infty$ with $p/M \to \gamma$, the limiting spectral distribution of the normalized Wishart matrix $\boldsymbol{W}_p$ is the Marchenko-Pastur law $\mu_\gamma$. Its Stieltjes transform $m_\gamma(z)$ is determined by the quadratic equation:*

$$z\gamma m_\gamma(z)^2 + (z + \gamma - 1)m_\gamma(z) + 1 = 0.$$

*For $z \in \mathbb{C} \setminus \text{supp}(\mu_\gamma)$:*

$$m_\gamma(z) = \frac{-(z+\gamma-1) - \sqrt{(z+\gamma-1)^2 - 4z\gamma}}{2z\gamma}.$$

## E.2  GENERAL NOTATION

We use the following notation throughout the appendix. Let $\boldsymbol{G}$ be a $M \times p$ random matrix with independent and identically distributed entries $G_{ij} \sim \mathcal{N}(0,1)$. Let $\boldsymbol{g}_i$ denote row $i$ of $\boldsymbol{G}$. Let $\boldsymbol{W}_p = \frac{1}{M}\boldsymbol{G}^\top\boldsymbol{G}$ denote the normalized Wishart matrix. Let $\lambda_1, \ldots, \lambda_p$ denote its eigenvalues. Let $\boldsymbol{g} \sim \mathcal{N}(0, \mathbb{I}_{p^2})$ be independent of $\boldsymbol{W}_p$, and let $\boldsymbol{w} \sim \mathcal{N}(0, \mathbb{I}_d)$ be independent of $(\boldsymbol{g}, \boldsymbol{W}_p)$.

Let $\otimes$ denote the Kronecker product, and $\text{tr}(\cdot)$ denote the trace. We state some of its properties below. For conformable matrices $\boldsymbol{A}, \boldsymbol{B}, \boldsymbol{C}$, vectors $\boldsymbol{a}, \boldsymbol{b}$,

$$(\boldsymbol{A} \otimes \boldsymbol{B})^{-1} = \boldsymbol{A}^{-1} \otimes \boldsymbol{B}^{-1}, \quad (29)$$

$$\text{tr}(\boldsymbol{A} \otimes \boldsymbol{B}) = \text{tr}(\boldsymbol{A})\text{tr}(\boldsymbol{B}), \quad (30)$$

$$(\boldsymbol{A} \otimes \mathbb{I})\text{vec}(\boldsymbol{C}) = \text{vec}(\boldsymbol{AC}), \quad (31)$$

$$(\boldsymbol{A} \otimes \mathbb{I})(\boldsymbol{B} \otimes \mathbb{I}) = (\boldsymbol{AB}) \otimes \mathbb{I}, \quad (32)$$

$$\text{vec}(\boldsymbol{ab}^\top)\text{vec}(\boldsymbol{ab}^\top)^\top = \boldsymbol{bb}^\top \otimes \boldsymbol{aa}^\top. \quad (33)$$

Also,

$$\mathrm{vec}(\boldsymbol{A})^\top \mathrm{vec}(\boldsymbol{B}) = \mathrm{tr}(\boldsymbol{A}^\top \boldsymbol{B}). \tag{34}$$

Let $\Pi_{p \times d}$ denote the permutation matrix with all entries 0 or 1, and each row has exactly one 1 and most one 1 per column. Let $a, b, c, b', c', \beta, \kappa \in \mathbb{R}$ be constants. Also, let $\mathcal{S} \subset \{1, \ldots, d\}$ denote a subset of indices, with $|\mathcal{S}| = p$. Next, we define matrices

$$\boldsymbol{A} = c\mathbb{I}_p + b\boldsymbol{W}_p, \quad \boldsymbol{B} = a\mathbb{I}_p + \boldsymbol{A}^{-1} = a\mathbb{I}_p + (c\mathbb{I}_p + b\boldsymbol{W}_p)^{-1},$$
$$\boldsymbol{\Sigma} = \boldsymbol{A} \otimes \mathbb{I}_p = c\mathbb{I}_{p^2} + b\boldsymbol{W}_p \otimes \mathbb{I}_p,$$
$$\mathbf{M} = \boldsymbol{B} \otimes \mathbb{I}_p = a\mathbb{I}_{p^2} + \boldsymbol{\Sigma}^{-1},$$
$$\boldsymbol{\Sigma}(\boldsymbol{w}) = \left(c' \frac{\|\boldsymbol{w}\|^2}{d} + b'\right)\mathbb{I}_{p^2} + (\boldsymbol{w}[\mathcal{S}]\boldsymbol{w}[\mathcal{S}]^\top) \otimes \mathbb{I}_p,$$
$$\boldsymbol{\Sigma}(\boldsymbol{w}_j) = \left(c' \frac{\|\boldsymbol{w}_j\|^2}{d} + b'\right)\mathbb{I}_{p^2} + (\boldsymbol{w}_j[\mathcal{S}]\boldsymbol{w}_j[\mathcal{S}]^\top) \otimes \mathbb{I}_p,$$

where $\boldsymbol{w}_j = \boldsymbol{g}_j$, the j$^{\text{th}}$ row of $\boldsymbol{G}$. Next, define vectors

$$\boldsymbol{v} = \mathrm{vec}(\boldsymbol{W}_p), \qquad \boldsymbol{\theta} = \boldsymbol{\Sigma}^{-1}\left(\frac{1}{M}\sum_{i=1}^{M} \boldsymbol{g}_i \otimes \boldsymbol{g}_i\right) = \boldsymbol{\Sigma}^{-1}\boldsymbol{v},$$
$$\boldsymbol{u} = \beta\boldsymbol{g} - d^{1/2}\boldsymbol{\Sigma}^{-1/2}\boldsymbol{\theta},$$

and scalars

$$z_a = -\frac{(1 + ac)}{ab}, \quad z_c = -\frac{c}{b}.$$

We consider the asymptotic regime where $d, p, M \to \infty$ with ratios

$$\frac{p}{d} =: \rho, \qquad \frac{M}{d} =: \mu,$$

where $\rho, \mu \in (0, \infty)$. The following lemmas are in the asymptotic regime $d, p, M \to \infty$ with fixed ratios, we will use shorthand $\lim_{d \to \infty}$ to denote $\lim_{d,p,M \to \infty}$.

### E.3 ASYMPTOTIC LIMIT OF $S(\tau, \beta)$

**Lemma 2.** *The normalized trace of the $p^2 \times p^2$ random matrix $\mathbf{M}^{-1}$ converges almost surely to a deterministic limit:*

$$\lim_{d \to \infty} \frac{1}{d^2}\mathrm{tr}(\mathbf{M}^{-1}) = \frac{\rho^2}{a}\left(1 - \frac{1}{ab}m_\gamma(z_a)\right).$$

*Proof.* Using properties of the Kronecker product (Eqs. (29) and (30)), we have:

$$\mathrm{tr}(\mathbf{M}^{-1}) = \mathrm{tr}(\boldsymbol{B}^{-1} \otimes \mathbb{I}_p) = p\,\mathrm{tr}(\boldsymbol{B}^{-1}) = p\,\mathrm{tr}\left(\left(a\mathbb{I}_p + (c\mathbb{I}_p + b\boldsymbol{W}_p)^{-1}\right)^{-1}\right).$$

The normalized trace is:

$$\frac{1}{d^2}\mathrm{tr}(\boldsymbol{M}^{-1}) = \frac{p\rho^2}{p^2}\sum_{i=1}^{p}\left(a + \frac{1}{c + b\lambda_i}\right)^{-1} = \frac{\rho^2}{p}\sum_{i=1}^{p}\frac{b\lambda_i + c}{ab\lambda_i + ac + 1}.$$

As $p, M \to \infty$, and the empirical spectral distribution of $\boldsymbol{W}_p$ converges to the Marchenko-Pastur law $\mu_\gamma$. The sum converges to the integral:

$$\mathcal{I} = \frac{\rho^2}{a}\int \frac{\lambda + c/b}{\lambda + (ac + 1)/ab}\,\mathrm{d}\mu_\gamma(\lambda).$$

Using Lemma 10 ($\mathcal{I}_1$ with $z_1 = z_c, z_2 = z_a$) gives the final expression. $\qquad \square$

**Lemma 3.** *Let* $Q_p := \boldsymbol{\theta}^\top \boldsymbol{\Sigma}^{-1/2} \mathbf{M}^{-1} \boldsymbol{\Sigma}^{-1/2} \boldsymbol{\theta}$. *Almost surely,*

$$\lim_{d \to \infty} \frac{1}{d} Q_p = \rho\Big( \frac{z_c^2}{b^2} m'_\gamma(z_c) + \frac{a}{b} \Big( z_c(z_c - 2z_a)\, m_\gamma(z_c) + z_a^2\, m_\gamma(z_a) \Big) \Big),$$

$$\lim_{d \to \infty} \frac{1}{d} \boldsymbol{\theta}^\top \boldsymbol{\Sigma} \boldsymbol{\theta} = \frac{\rho}{b} \Big( 1 + z_c + z_c^2 m_\gamma(z_c) \Big).$$

*Proof.* First, we simplify $Q_d$:

$$\boldsymbol{\theta}^\top \boldsymbol{\Sigma}^{-1/2} \mathbf{M}^{-1} \boldsymbol{\Sigma}^{-1/2} \boldsymbol{\theta} = \boldsymbol{v}^\top \boldsymbol{\Sigma}^{-1} \boldsymbol{\Sigma}^{-1/2} \mathbf{M}^{-1} \boldsymbol{\Sigma}^{-1/2} \boldsymbol{\Sigma}^{-1} \boldsymbol{v}$$

$$= \boldsymbol{v}^\top \boldsymbol{\Sigma}^{-3/2} \mathbf{M}^{-1} \boldsymbol{\Sigma}^{-3/2} \boldsymbol{v}.$$

Since $\mathbf{M}$ shares eigenvectors with $\boldsymbol{\Sigma}$, it commutes with $\boldsymbol{\Sigma}$, and we have that

$$\boldsymbol{\Sigma}^{-3/2} \mathbf{M}^{-1} \boldsymbol{\Sigma}^{-3/2} = \boldsymbol{\Sigma}^{-3} \mathbf{M}^{-1} = (a\boldsymbol{\Sigma}^3 + \boldsymbol{\Sigma}^2)^{-1}.$$

Hence, $Q_p = \boldsymbol{v}^\top (a\boldsymbol{\Sigma}^3 + \boldsymbol{\Sigma}^2)^{-1} \boldsymbol{v}$. Using Eqs. (29) and (31), we have

$$(a\boldsymbol{\Sigma}^3 + \boldsymbol{\Sigma}^2)^{-1} \boldsymbol{v} = \mathrm{vec}\left( (a\boldsymbol{A}^3 + \boldsymbol{A}^2)^{-1} \boldsymbol{W}_d \right).$$

Using Eq. (34), we get

$$Q_p = \mathrm{tr}\left( \boldsymbol{W}_p (a\boldsymbol{A}^3 + \boldsymbol{A}^2)^{-1} \boldsymbol{W}_p \right) = \mathrm{tr}\left( \boldsymbol{W}_p^2 (a\boldsymbol{A}^3 + \boldsymbol{A}^2)^{-1} \right).$$

Since $\boldsymbol{W}_p$ and $\boldsymbol{A}$ share eigenvectors, with eigenvalues $\lambda_i$ and $c + b\lambda_i$, respectively,

$$Q_p = \sum_{i=1}^p \frac{\lambda_i^2}{(c + b\lambda_i)^2 \, (a(c + b\lambda_i) + 1)}.$$

As $p, M \to \infty$, the empirical spectral distribution of $\boldsymbol{W}_p$ converges to the Marchenko-Pastur law $\mu_\gamma$, yielding

$$\lim_{d \to \infty} \frac{1}{d} Q_p = \frac{\rho}{ab^3} \int \frac{\lambda^2}{(\lambda + c/b)^2 (\lambda + (ac+1)/ab)} \, \mathrm{d}\mu_\gamma(x).$$

Using Lemma 10 ($\mathcal{I}_3$ with $z_1 = z_c, z_2 = z_a$) gives the final expression for the first result.

For the second part, we have that

$$\boldsymbol{\theta}^\top \boldsymbol{\Sigma} \boldsymbol{\theta} = \boldsymbol{v}^\top \boldsymbol{\Sigma}^{-1} \boldsymbol{\Sigma} \boldsymbol{\Sigma}^{-1} \boldsymbol{v} = \boldsymbol{v}^\top \boldsymbol{\Sigma}^{-1} \boldsymbol{v}.$$

Using Eqs. (29) and (31),

$$\boldsymbol{\Sigma}^{-1} \boldsymbol{v} = \boldsymbol{\Sigma}^{-1} \mathrm{vec}(\boldsymbol{W}_d) = \mathrm{vec}(\boldsymbol{A}^{-1} \boldsymbol{W}_d).$$

Using Eq. (34),

$$\boldsymbol{\theta}^\top \boldsymbol{\Sigma} \boldsymbol{\theta} = \mathrm{vec}(\boldsymbol{W}_p)^\top \mathrm{vec}(\boldsymbol{A}^{-1} \boldsymbol{W}_p) = \mathrm{tr}\left( \boldsymbol{W}_d \boldsymbol{A}^{-1} \boldsymbol{W}_d \right) = \mathrm{tr}\left( \boldsymbol{W}_d^2 \boldsymbol{A}^{-1} \right).$$

Since $\boldsymbol{W}_p$ and $\boldsymbol{A}$ share eigenvectors, with eigenvalues $\lambda_i$ and $c + b\lambda_i$, respectively,

$$\boldsymbol{\theta}^\top \boldsymbol{\Sigma} \boldsymbol{\theta} = \sum_{i=1}^p \frac{\lambda_i^2}{c + b\lambda_i}.$$

As $p, M \to \infty$, the empirical spectral distribution of $\boldsymbol{W}_p$ converges to the Marchenko-Pastur law $\mu_\gamma$, yielding

$$\lim_{d \to \infty} \frac{1}{d} \boldsymbol{\theta}^\top \boldsymbol{\Sigma} \boldsymbol{\theta} = \frac{\rho}{b} \int \frac{\lambda^2}{\lambda + c/b} \, \mathrm{d}\mu_\gamma(x).$$

Using Lemma 10 ($\mathcal{I}_1$ with $z_1 = 0, z_2 = z_c$), we get

$$\int \frac{\lambda}{\lambda + c/b} \, \mathrm{d}\mu_\gamma(x) = 1 + z_c m_\gamma(z_c).$$

Then using Lemma 10 ((v), with $\alpha_0 = 1, \alpha_1 = z_1 = z_c, \alpha_2 = 0$), we get the final expression. $\qquad \square$

## E.5 FIRST ORDER OPTIMALITY CONDITIONS OF $D(\bar{u}, \bar{\beta})$

**Theorem 5.** *Consider $D(\bar{u}, \bar{\beta})$ as in Eq.* (25). *Let $a := \frac{\bar{\beta}\nu}{\bar{u}}$. Then, the solution of $\min_{\bar{u}} \max_{\bar{\beta}} D(\bar{u}, \bar{\beta})$ is given by*

$$\bar{u} = \frac{\bar{\beta}\nu}{a}, \quad a = \frac{\gamma\nu/\rho^2 - (c+1) + \sqrt{\left(\gamma\nu/\rho^2 - (c+1)\right)^2 + 4\gamma c\nu/\rho^2}}{2\gamma c(1 - \nu/\rho^2)}, \quad \gamma = \mu^{-1},$$

$$\bar{\beta}^2 = \frac{a^2\left(\sigma_n^2 + 1 - \rho + \rho c\right) + \rho(1 - a^2 c^2)m_\gamma(z_a) - \rho a z_a^2 m_\gamma'(z_a)}{\rho^2\left(1 - \frac{\nu}{\rho^2} - \frac{m_\gamma'(z_a)}{a^2}\right)}. \tag{35}$$

*Proof.* We derive the first order optimality conditions of $D(\bar{u}, \bar{\beta})$. First, we have

$$\frac{\partial D(\bar{u}, \bar{\beta})}{\partial \bar{u}} = 0$$

$$\implies \frac{\bar{\beta}}{2} - \frac{\bar{\beta}c_\infty}{2\bar{u}^2} - \frac{\bar{\beta}^2}{2}\frac{\partial s_\infty(\bar{u}, \bar{\beta})}{\partial \bar{u}} - \frac{1}{2}\frac{\partial v_\infty(\bar{u}, \bar{\beta})}{\partial \bar{u}} = 0 \tag{36}$$

Similarly, we have

$$\frac{\partial D(\bar{u}, \bar{\beta})}{\partial \bar{\beta}} = 0$$

$$\implies \frac{\bar{u}}{2} + \frac{c_\infty}{2\bar{u}} - \bar{\beta}s_\infty(\bar{u}, \bar{\beta}) - \frac{\bar{\beta}^2}{2}\frac{\partial s_\infty(\bar{u}, \bar{\beta})}{\partial \bar{\beta}} - \frac{1}{2}\frac{\partial v_\infty(\bar{u}, \bar{\beta})}{\partial \bar{\beta}} = 0 \tag{37}$$

We first simplify the limits $v_\infty, s_\infty$ before we get the partial derivatives. First recall that

$$\overline{\boldsymbol{\Sigma}}_{\boldsymbol{H}_{\mathcal{S}_p}} := d^3 \boldsymbol{\Sigma}_{\boldsymbol{H}_{\mathcal{S}_p}} = \frac{1}{\tau}\left(\sigma_n^2 + \frac{1}{M}\sum_{i=1}^M \frac{\|\boldsymbol{w}_i\|^2}{d}\right)\mathbb{I}_{p^2} + \frac{1}{M}\sum_{i=1}^M (\boldsymbol{w}_i[\mathcal{S}]\boldsymbol{w}_i[\mathcal{S}]^\top) \otimes \mathbb{I}_p,$$

$$\overline{\boldsymbol{M}} := \frac{\boldsymbol{M}}{d^3} = \frac{\bar{\beta}\nu}{\bar{u}}\mathbb{I} + \overline{\boldsymbol{\Sigma}}_{\boldsymbol{H}_{\mathcal{S}_p}}^{-1}$$

$$\overline{\boldsymbol{\theta}}_\star := \frac{\boldsymbol{\theta}_\star}{d} = \frac{1}{dn}\boldsymbol{\Sigma}_{\boldsymbol{H}_{\mathcal{S}_p}}^{-1}\boldsymbol{\sigma}_{\boldsymbol{H}_{\mathcal{S}_p}\boldsymbol{y}} = \overline{\boldsymbol{\Sigma}}_{\boldsymbol{H}_{\mathcal{S}_p}}^{-1}\left(\frac{1}{M}\sum_{i=1}^M \boldsymbol{w}_i[\mathcal{S}] \otimes \boldsymbol{w}_i[\mathcal{S}]\right).$$

Then, since $\lim_{d\to\infty}\frac{\|\boldsymbol{w}_i\|^2}{d} = 1$, using Lemma 2 with $\mathbf{M} = \overline{\boldsymbol{M}}, a = \frac{\bar{\beta}\nu}{\bar{u}}, b = 1, c = \frac{\sigma_n^2+1}{\tau}, \gamma = \mu^{-1}, z_a = -\frac{1}{a} - c$, we get

$$s_\infty(\bar{u}, \bar{\beta}) = \lim_{d\to\infty} d\operatorname{tr}(\boldsymbol{M}^{-1}) = \lim_{d\to\infty}\frac{1}{d^2}\operatorname{tr}(\overline{\boldsymbol{M}}^{-1}) = \rho^2\left(\frac{1}{a} - \frac{1}{a^2}m_\gamma(z_a)\right). \tag{38}$$

Using Lemma 3 with $\boldsymbol{\Sigma} = \overline{\boldsymbol{\Sigma}}_{\boldsymbol{H}_{\mathcal{S}_p}}, \boldsymbol{\theta} = \overline{\boldsymbol{\theta}}_\star$, we get

$$v_\infty(\bar{u}, \bar{\beta}) = \lim_{d\to\infty}\frac{1}{d}(\boldsymbol{\Sigma}_{\boldsymbol{H}_{\mathcal{S}_p}}^{-1/2}\overline{\boldsymbol{\theta}}_\star)^\top \boldsymbol{M}^{-1}(\boldsymbol{\Sigma}_{\boldsymbol{H}_{\mathcal{S}_p}}^{-1/2}\overline{\boldsymbol{\theta}}_\star) = \lim_{d\to\infty}\frac{1}{d}(\overline{\boldsymbol{\Sigma}}_{\boldsymbol{H}_{\mathcal{S}_p}}^{-1/2}\overline{\boldsymbol{\theta}}_\star)^\top \overline{\boldsymbol{M}}^{-1}(\overline{\boldsymbol{\Sigma}}_{\boldsymbol{H}_{\mathcal{S}_p}}^{-1/2}\overline{\boldsymbol{\theta}}_\star)$$

$$= \rho(c^2 m_\gamma'(z_c) + ac(c + 2z_a)m_\gamma(z_c) + az_a^2 m_\gamma(z_a)).$$

As both $s_\infty(\bar{u}, \bar{\beta})$ and $v_\infty(\bar{u}, \bar{\beta})$ depend on $\bar{u}, \bar{\beta}$ through $a$, we use the derivatives with respect to $a$ to find the partial derivatives wrt $\bar{u}, \bar{\beta}$.

$$\frac{\partial s_\infty(\bar{u}, \bar{\beta})}{\partial \bar{u}} = \frac{\partial s_\infty(\bar{u}, \bar{\beta})}{\partial a}\frac{\partial a}{\partial \bar{u}} = \frac{\partial s_\infty(\bar{u}, \bar{\beta})}{\partial a}\left(-\frac{\bar{\beta}\nu}{\bar{u}^2}\right).$$

$$\frac{\partial s_\infty(\bar{u}, \bar{\beta})}{\partial \bar{\beta}} = \frac{\partial s_\infty(\bar{u}, \bar{\beta})}{\partial a}\frac{\partial a}{\partial \bar{\beta}} = \frac{\partial s_\infty(\bar{u}, \bar{\beta})}{\partial a}\left(\frac{\nu}{\bar{u}}\right).$$

$$\frac{\partial v_\infty(\bar{u}, \bar{\beta})}{\partial \bar{u}} = \frac{\partial v_\infty(\bar{u}, \bar{\beta})}{\partial a} \frac{\partial a}{\partial \bar{u}} = \frac{\partial v_\infty(\bar{u}, \bar{\beta})}{\partial a} \left( -\frac{\bar{\beta}\nu}{\bar{u}^2} \right).$$

$$\frac{\partial v_\infty(\bar{u}, \bar{\beta})}{\partial \bar{\beta}} = \frac{\partial v_\infty(\bar{u}, \bar{\beta})}{\partial a} \frac{\partial a}{\partial \bar{\beta}} = \frac{\partial v_\infty(\bar{u}, \bar{\beta})}{\partial a} \left( \frac{\nu}{\bar{u}} \right).$$

Substituting back in Eq. (36)

$$\begin{aligned}
\frac{\partial D(\bar{u}, \bar{\beta})}{\partial \bar{u}} &= \frac{\bar{\beta}}{2} - \frac{\bar{\beta}c_\infty}{2\bar{u}^2} - \frac{\bar{\beta}^2}{2} \frac{\partial s_\infty(\bar{u}, \bar{\beta})}{\partial \bar{u}} - \frac{1}{2} \frac{\partial v_\infty(\bar{u}, \bar{\beta})}{\partial \bar{u}} \\
&= \frac{\bar{\beta}}{2} - \frac{\bar{\beta}c_\infty}{2\bar{u}^2} - \frac{\bar{\beta}^2}{2} \left( \frac{\partial s_\infty(\bar{u}, \bar{\beta})}{\partial a} \right) \left( -\frac{\bar{\beta}\nu}{\bar{u}^2} \right) - \frac{1}{2} \left( \frac{\partial v_\infty(\bar{u}, \bar{\beta})}{\partial a} \right) \left( -\frac{\bar{\beta}\nu}{\bar{u}^2} \right) \\
&= \frac{\bar{\beta}}{2} - \frac{\bar{\beta}c_\infty}{2\bar{u}^2} + \frac{\bar{\beta}^3\nu}{2\bar{u}^2} \frac{\partial s_\infty(\bar{u}, \bar{\beta})}{\partial a} + \frac{\bar{\beta}\nu}{2\bar{u}^2} \frac{\partial v_\infty(\bar{u}, \bar{\beta})}{\partial a} = 0.
\end{aligned}$$

Simplifying it further we get this is

$$\bar{u}^2 - c_\infty + \bar{\beta}^2\nu\frac{\partial s_\infty(\bar{u}, \bar{\beta})}{\partial a} + \nu\frac{\partial v_\infty(\bar{u}, \bar{\beta})}{\partial a} = 0$$

$$\iff \quad \bar{u}^2 = c_\infty - \nu\left( \bar{\beta}^2\frac{\partial s_\infty(\bar{u}, \bar{\beta})}{\partial a} + \frac{\partial v_\infty(\bar{u}, \bar{\beta})}{\partial a} \right). \tag{39}$$

Similarly, substituting in Eq. (37)

$$\begin{aligned}
\frac{\partial D(\bar{u}, \bar{\beta})}{\partial \bar{\beta}} &= \frac{\bar{u}}{2} + \frac{c_\infty}{2\bar{u}} - \bar{\beta}\, s_\infty(\bar{u}, \bar{\beta}) - \frac{\bar{\beta}^2}{2} \frac{\partial s_\infty(\bar{u}, \bar{\beta})}{\partial \bar{\beta}} - \frac{1}{2} \frac{\partial v_\infty(\bar{u}, \bar{\beta})}{\partial \bar{\beta}} \\
&= \frac{\bar{u}}{2} + \frac{c_\infty}{2\bar{u}} - \bar{\beta}\, s_\infty(\bar{u}, \bar{\beta}) - \frac{\bar{\beta}^2}{2} \left( \frac{\partial s_\infty(\bar{u}, \bar{\beta})}{\partial a} \right) \left( \frac{\nu}{\bar{u}} \right) - \frac{1}{2} \left( \frac{\partial v_\infty(\bar{u}, \bar{\beta})}{\partial a} \right) \left( \frac{\nu}{\bar{u}} \right) = 0.
\end{aligned}$$

Simplifying it further we have

$$\bar{u} + \frac{c_\infty}{\bar{u}} - 2\bar{\beta}\, s_\infty(\bar{u}, \bar{\beta}) - \frac{\nu}{\bar{u}} \left( \bar{\beta}^2\frac{\partial s_\infty(\bar{u}, \bar{\beta})}{\partial a} + \frac{\partial v_\infty(\bar{u}, \bar{\beta})}{\partial a} \right) = 0.$$

Using Eq. (39), we get

$$\begin{aligned}
0 &= \bar{u} + \frac{c_\infty}{\bar{u}} - 2\bar{\beta}\, s_\infty(\bar{u}, \bar{\beta}) - \frac{1}{\bar{u}} \left( c_\infty - \bar{u}^2 \right) \\
&= 2\bar{u} - 2\bar{\beta}\, s_\infty(\bar{u}, \bar{\beta}) \\
\iff \quad &\bar{u} = \bar{\beta}\, s_\infty(\bar{u}, \bar{\beta}). \tag{40}
\end{aligned}$$

Using Eq. (38) and simplifying Eq. (40), we get

$$\begin{aligned}
&\bar{u} = \frac{\bar{\beta}\rho^2}{a} \left( 1 - \frac{1}{a}m_\gamma\left(z_a\right) \right) \\
\iff \quad &\frac{\rho^2}{\nu} \left( 1 - \frac{1}{a}m_\gamma\left(z_a\right) \right) = 1, \\
\iff \quad &\left( 1 - \frac{\nu}{\rho^2} \right) a = m_\gamma\left(z_a\right). \tag{41}
\end{aligned}$$

Eq. (41) can be simplified as follows. MP quadratic for $m_\gamma(z)$:

$$z\gamma m_\gamma(z)^2 + (z + \gamma - 1)m_\gamma(z) + 1 = 0.$$

Substituting $z = z_a = -1/a - c, m_\gamma(z_a) = a(1 - \nu)$, we have

$$\begin{aligned}
&\left( -\frac{1}{a} - c \right) \gamma a^2(1 - \nu/\rho^2)^2 + \left( -\frac{1}{a} - c + \gamma - 1 \right) a(1 - \nu/\rho^2) + 1 = 0 \\
\iff \quad &-(1 + ac)\gamma a(1 - \nu/\rho^2)^2 - (1 + ac)(1 - \nu/\rho^2) + (\gamma - 1)a(1 - \nu/\rho^2) + 1 = 0 \\
\iff \quad &-\gamma c(1 - \nu/\rho^2)^2 a^2 + (1 - \nu/\rho^2)\left( \gamma\nu/\rho^2 - (c + 1) \right)a + \nu/\rho^2 = 0 \\
\iff \quad &c\gamma(1 - \nu/\rho^2)a^2 + \left( (c + 1) - \gamma\nu/\rho^2 \right)a - \frac{\nu/\rho^2}{1 - \nu/\rho^2} = 0.
\end{aligned}$$

Solving for $a$ using the quadratic formula finishes the proof for the first part.

Define

$$q_\infty(\bar{u}, \bar{\beta}) := -\left(\bar{\beta}^2 \frac{\partial s_\infty(\bar{u}, \bar{\beta})}{\partial a} + \frac{\partial v_\infty(\bar{u}, \bar{\beta})}{\partial a}\right).$$

Next to get a simplified form of this, we use $\frac{dz}{da} = \frac{1}{a^2}$ and get the partials

$$s_\infty(\bar{u}, \bar{\beta}) = \rho^2 \left(\frac{1}{a} - \frac{1}{a^2} m_\gamma(z_a)\right),$$

$$\frac{\partial s_\infty(\bar{u}, \bar{\beta})}{\partial a} = \rho^2 \left(-\frac{1}{a^2} + \frac{2}{a^3} m_\gamma(z_a) - \frac{1}{a^4} m'_\gamma(z_a)\right).$$

$$v_\infty(\bar{u}, \bar{\beta}) = \rho(c^2 m'_\gamma(-c) + ac(c + 2z_a) m_\gamma(-c) + a(z_a)^2 m_\gamma(z_a))$$

$$\frac{\partial v_\infty(\bar{u}, \bar{\beta})}{\partial a} = \rho(c m_\gamma(-c)\left(c + 2z_a + \frac{2}{a}\right) + z_a m_\gamma(z_a)\left(z_a + \frac{2}{a}\right) + \frac{z_a^2}{a} m'_\gamma(z_a))$$

$$= \rho(-c^2 m_\gamma(-c) + z_a m_\gamma(z_a)\left(\frac{1}{a} - c\right) + \frac{z_a^2}{a} m'_\gamma(z_a)).$$

$$q_\infty(\bar{u}, \bar{\beta}) = -\left[\bar{\beta}^2 \rho^2\left(-\frac{1}{a^2} + \frac{2}{a^3} m_\gamma(z_a) - \frac{1}{a^4} m'_\gamma(z_a)\right)\right.$$

$$+ \rho\left(-c^2 m_\gamma(-c) + z_a m_\gamma(z_a)\left(\frac{1}{a} - c\right) + \frac{z_a^2}{a} m'_\gamma(z_a)\right)\bigg]$$

$$= \frac{\bar{\beta}^2 \rho^2}{a^2}\left(1 - \frac{2}{a} m_\gamma(z_a) + \frac{1}{a^2} m'_\gamma(z_a)\right) + \rho\left(c^2 m_\gamma(z_c) + \left(\frac{1}{a^2} - c^2\right) m_\gamma(z_a) - \frac{z_a^2}{a} m'_\gamma(z_a)\right).$$

(42)

Next, we simplify $c_\infty$. Using Eq. (11) and Lemma 3 with $c = \frac{\sigma_n^2 + 1}{\tau}, b = 1$, we have

$$c_\infty == \lim_{d\to\infty} \frac{1}{d^2}\left(\sum_{i=1}^n d_i^2\right) = \lim_{d\to\infty} \frac{1}{d^2}\left(\sum_{i=1}^n \sigma_x^2 \|\boldsymbol{w}_i\|^2 + n\sigma_n^2 - n\boldsymbol{\theta}_\star^\top \boldsymbol{\Sigma}_{\boldsymbol{H}_{\mathcal{S}_p}} \boldsymbol{\theta}_\star\right)$$

$$= \lim_{d\to\infty} \frac{1}{d^2}\left(\sum_{i=1}^M \frac{n}{M} \sigma_x^2 \|\boldsymbol{w}_i\|^2\right) - \nu \lim_{d\to\infty} \boldsymbol{\theta}_\star^\top \boldsymbol{\Sigma}_{\boldsymbol{H}_{\mathcal{S}_p}} \boldsymbol{\theta}_\star + \nu\sigma_n^2$$

$$= \nu \lim_{d\to\infty} \frac{1}{Md}\left(\sum_{i=1}^M \|\boldsymbol{w}_i\|^2\right) - \nu \lim_{d\to\infty} \frac{1}{d} \overline{\boldsymbol{\theta}}_\star^\top \overline{\boldsymbol{\Sigma}}_{\boldsymbol{H}_{\mathcal{S}_p}} \overline{\boldsymbol{\theta}}_\star + \nu\sigma_n^2$$

$$= \nu(1 + \sigma_n^2) - \nu\rho(1 - c + c^2 m_\gamma(-c)) = \nu(\sigma_n^2 + 1 - \rho + \rho c - \rho c^2 m_\gamma(-c)). \quad (43)$$

We can write Eq. (39) as

$$\bar{u}^2 = c_\infty + \nu\, q_\infty(\bar{u}, \bar{\beta}),$$

Substituting $c_\infty$ and $q_\infty$ using Eqs. (42) and (43), and using $a = \frac{\bar{\beta}\nu}{\bar{u}}$, we get

$$\bar{\beta}_*^2 \nu = a^2(\sigma_n^2 + 1 - \rho + \rho c) + \bar{\beta}^2 \rho^2\left(1 - \frac{2}{a} m_\gamma(z_a) + \frac{1}{a^2} m'_\gamma(z_a)\right) + \rho\left(1 - a^2 c^2\right) m_\gamma(z_a) - \rho a z_a^2 m'_\gamma(z_a).$$

Solving for $\bar{\beta}^2$ gives Eq. (35).

$\square$

## F    PROOF FOR LS AO

The PO is
$$\min_{\boldsymbol{a}} \frac{1}{2}\|\boldsymbol{Ga} - \boldsymbol{D_w g'}\|^2 = \min_{\boldsymbol{a}} \max_{\boldsymbol{u}} \ \boldsymbol{u}^\top \boldsymbol{Ga} - \boldsymbol{u}^\top \boldsymbol{D_w g'} - \frac{1}{2}\|\boldsymbol{u}\|^2.$$

By direct application of the CGMT, the AO is

$$\min_{\boldsymbol{a}} \max_{\boldsymbol{u}} \ \|\boldsymbol{u}\|\boldsymbol{g}^\top \boldsymbol{a} + \|\boldsymbol{a}\|\mathbf{h}^\top \boldsymbol{u} - \boldsymbol{u}^\top \boldsymbol{D_w g'} - \frac{1}{2}\|\boldsymbol{u}\|^2.$$

Let $\alpha = \|\boldsymbol{a}\|, \beta = \|\boldsymbol{u}\|$, and $\hat{\boldsymbol{a}}$ and $\hat{\boldsymbol{u}}$ be the corresponding unit norm directions. Then the above can be written as

$$\min_{\alpha \geq 0, \hat{\boldsymbol{a}}} \max_{\beta \geq 0, \hat{\boldsymbol{u}}} \ \beta\alpha\boldsymbol{g}^\top \hat{\boldsymbol{a}} + \beta\hat{\boldsymbol{u}}^\top(\alpha\mathbf{h} - \boldsymbol{D_w g'}) - \frac{1}{2}\beta^2.$$

Solving for $\hat{\boldsymbol{a}}$ and $\hat{\boldsymbol{u}}$, the optimal is at

$$\hat{\boldsymbol{a}} = -\frac{\boldsymbol{g}}{\|\boldsymbol{g}\|}, \quad \hat{\boldsymbol{u}} = \frac{\alpha\mathbf{h} - \boldsymbol{D_w g'}}{\|\alpha\mathbf{h} - \boldsymbol{D_w g'}\|}.$$

Plugging these back, we have

$$\min_{\alpha \geq 0} \max_{\beta \geq 0} \ \beta\|\alpha\mathbf{h} - \boldsymbol{D_w g'}\| - \beta\alpha\|\boldsymbol{g}\| - \frac{1}{2}\beta^2$$

$$= \min_{\alpha \geq 0} \max_{\beta \geq 0} \ \beta\left(\sqrt{n\alpha^2 + \sum_{i\in[n]} d_i^2} - \alpha\|\boldsymbol{g}\|\right) - \frac{1}{2}\beta^2$$

$$= \frac{1}{2}\left(\min_{\alpha \geq 0}\sqrt{n\alpha^2 + \sum_{i=1}^n d_i^2} - \alpha\|\boldsymbol{g}\|\right)_+^2$$

where $(\cdot)_+ = \max\{x, 0\}$. Thus, using Eq. (1),

$$\alpha_*^2 = \frac{\sum_{i=1}^n d_i^2}{n\left(\frac{n}{\|\boldsymbol{g}\|^2} - 1\right)} = \frac{\sum_{i=1}^n d_i^2}{\nu d^2\left(\frac{\nu p^2}{\rho^2\|\boldsymbol{g}\|^2} - 1\right)}.$$

Then, we get $\boldsymbol{a}_* = -\alpha_*\hat{\boldsymbol{g}}$. Using Eq. (24),

$$\kappa_\infty := \lim_{d\to\infty} \alpha_* = \sqrt{\frac{c_\infty}{\nu\left(\frac{\nu}{\rho^2} - 1\right)}}.$$

In the limit, $\boldsymbol{a}_* = \kappa_\infty \frac{\boldsymbol{g}}{\|\boldsymbol{g}\|}$.

## G    ASYMPTOTIC RISK

### G.1    ASYMPTOTIC RISK OF MINIMUM-NORM LGP

**Lemma 4** (Term-1, OOD Loss). *Using $\boldsymbol{M}, \boldsymbol{\Sigma}, \boldsymbol{\Sigma}(\boldsymbol{w}), \boldsymbol{u}, \boldsymbol{\theta}$ of the same form as Appendix E.2, let*

$$T_1(\boldsymbol{w}) := \frac{1}{d^2}\boldsymbol{u}^\top \mathbf{M}^{-1}\boldsymbol{\Sigma}^{-1/2}\boldsymbol{\Sigma}(\boldsymbol{w})\boldsymbol{\Sigma}^{-1/2}\mathbf{M}^{-1}\boldsymbol{u},$$

$$T_2(\boldsymbol{w}) := \frac{1}{d}\boldsymbol{\theta}^\top \boldsymbol{\Sigma}(\boldsymbol{w})\boldsymbol{\theta}, \qquad T_3(\boldsymbol{w}) := \frac{1}{d^{3/2}}\boldsymbol{u}^\top \mathbf{M}^{-1}\boldsymbol{\Sigma}^{-1/2}\boldsymbol{\Sigma}(\boldsymbol{w})\boldsymbol{\theta}.$$

*Then,*

$$\lim_{d\to\infty} \mathbb{E}_{\boldsymbol{w}}[T_1(\boldsymbol{w})] = (1 + b' + c')J_1,$$

$$\lim_{d\to\infty} \mathbb{E}_{\boldsymbol{w}}[T_2(\boldsymbol{w})] = \frac{(1 + b' + c')\rho}{b^2}\left(1 + 2z_c m_\gamma(z_c) + z_c^2 m_\gamma'(z_c)\right), \tag{44}$$

$$\lim_{d\to\infty} \mathbb{E}_{\boldsymbol{w}}[T_3(\boldsymbol{w})] = -(1 + b' + c')\rho\left(-\frac{c(ac+2)}{b^3}m_\gamma(z_c) + \frac{a}{b}z_a^2 m_\gamma(z_a) + \frac{1}{b^2}z_c^2 m_\gamma'(z_c)\right), \tag{45}$$

*where*

$$J_1 = \frac{\rho}{b^2}\left[\frac{\beta^2\rho}{a^2}\left(bm_\gamma(z_a) - \frac{1}{a}m'_\gamma(z_a)\right) - 2\,abz_cz_a\big(m_\gamma(z_c) - m_\gamma(z_a)\big) + z_c^2 m'_\gamma(z_c) + z_a^2 m'_\gamma(z_a)\right].$$

(46)

*Proof.* We first work with $T_1(\boldsymbol{w})$. Using Lemma 11, since $\boldsymbol{w}$ is independent of $\boldsymbol{g}, \boldsymbol{W}_p$, we get

$$\mathbb{E}_{\boldsymbol{w}}[T_1(\boldsymbol{w})] = \frac{c' + b' + 1}{d^2}\boldsymbol{u}^\top \mathbf{M}^{-1}\boldsymbol{\Sigma}^{-1}\mathbf{M}^{-1}\boldsymbol{u}.$$

Since $\mathbf{M}$ and $\boldsymbol{\Sigma}$ share eigenvectors, they commute, and we have that $\mathbf{M}^{-1}\boldsymbol{\Sigma}^{-1}\mathbf{M}^{-1} = \boldsymbol{\Sigma}^{-1}\mathbf{M}^{-2}$. Further, we have that

$$\begin{aligned}
\boldsymbol{u}^\top\boldsymbol{\Sigma}^{-1}\mathbf{M}^{-2}\boldsymbol{u} &= \beta^2\boldsymbol{g}^\top(\boldsymbol{\Sigma}^{-1}\mathbf{M}^{-2})\boldsymbol{g} + d\boldsymbol{\theta}^\top\boldsymbol{\Sigma}^{-1/2}\boldsymbol{\Sigma}^{-1}\mathbf{M}^{-2}\boldsymbol{\Sigma}^{-1/2}\boldsymbol{\theta} \\
&\quad - \beta d^{1/2}\boldsymbol{\theta}^\top\boldsymbol{\Sigma}^{-1/2}\boldsymbol{\Sigma}^{-1}\mathbf{M}^{-2}\boldsymbol{g} - \beta d^{1/2}\boldsymbol{g}^\top\boldsymbol{\Sigma}^{-1}\mathbf{M}^{-2}\boldsymbol{\Sigma}^{-1/2}\boldsymbol{\theta} \\
&= \beta^2\boldsymbol{g}^\top(\boldsymbol{\Sigma}^{-1}\mathbf{M}^{-2})\boldsymbol{g} + d\boldsymbol{\theta}^\top\boldsymbol{\Sigma}^{-2}\mathbf{M}^{-2}\boldsymbol{\theta} - 2\beta d^{1/2}\boldsymbol{g}^\top\boldsymbol{\Sigma}^{-3/2}\mathbf{M}^{-2}\boldsymbol{\theta}.
\end{aligned}$$

$$\begin{aligned}
J_1 &= \lim_{d\to\infty}\frac{1}{d^2}\boldsymbol{u}^\top\boldsymbol{\Sigma}^{-1}\mathbf{M}^{-2}\boldsymbol{u} \\
&= \lim_{p\to\infty}\frac{\rho^2}{p^2}\beta^2\boldsymbol{g}^\top(\boldsymbol{\Sigma}^{-1}\mathbf{M}^{-2})\boldsymbol{g} + \lim_{p\to\infty}\frac{\rho}{p}\boldsymbol{\theta}^\top\boldsymbol{\Sigma}^{-2}\mathbf{M}^{-2}\boldsymbol{\theta} - \lim_{p\to\infty}\frac{\rho^{3/2}}{p^{3/2}}2\beta\boldsymbol{g}^\top\boldsymbol{\Sigma}^{-3/2}\mathbf{M}^{-2}\boldsymbol{\theta} \\
&= \lim_{p\to\infty}\beta^2\mathcal{I}'_1(p) + \lim_{p\to\infty}\mathcal{I}'_2(p),
\end{aligned}$$

(47)

where

$$\mathcal{I}'_1(p) := \frac{1}{d^2}\operatorname{tr}(\boldsymbol{\Sigma}^{-1}\mathbf{M}^{-2}), \quad \mathcal{I}'_2(p) := \lim_{d\to\infty}\frac{1}{d}\boldsymbol{\theta}^\top\boldsymbol{\Sigma}^{-2}\mathbf{M}^{-2}\boldsymbol{\theta},$$

For the first term as $\boldsymbol{g} \in \mathbb{R}^{p^2}$ we apply Lemma 14 with $d_1 = p^2$ and get

$$\lim_{p\to\infty}\frac{1}{p^2}\boldsymbol{g}^\top(\boldsymbol{\Sigma}^{-1}\mathbf{M}^{-2})\boldsymbol{g} = \lim_{p\to\infty}\frac{1}{p^2}\operatorname{tr}(\boldsymbol{\Sigma}^{-1}\mathbf{M}^{-2}).$$

Also, for the third term, from Lemma 14 it follows that

$$\lim_{p\to\infty}\frac{1}{p^{3/2}}\boldsymbol{g}^\top\boldsymbol{\Sigma}^{-3/2}\mathbf{M}^{-2}\boldsymbol{\theta} = 0,$$

(48)

as $\|\boldsymbol{\Sigma}^{-3/2}\mathbf{M}^{-2}\boldsymbol{\theta}\| = O(\|\boldsymbol{\theta}\|) = O(\texttt{vec}(\boldsymbol{W}_p)) = O(\sqrt{p})$.

For $\mathcal{I}'_1(p)$, using Eqs. (29), (30) and (32) we have that

$$\begin{aligned}
\mathcal{I}'_1(p) &= \frac{\rho^2}{p^2}\operatorname{tr}((\boldsymbol{A}^{-1}\boldsymbol{B}^{-2}) \otimes \mathbb{I}_p) \\
&= \frac{\rho^2}{p}\operatorname{tr}((c\mathbb{I}_p + b\boldsymbol{W}_p)^{-1}(a\mathbb{I}_p + (c\mathbb{I}_p + b\boldsymbol{W}_p)^{-1})^{-2}) \\
&= \frac{\rho^2}{p}\sum_{i=1}^{p}\frac{1}{c + b\lambda_i}\frac{1}{\left(a + \frac{1}{c+b\lambda_i}\right)^2} = \frac{\rho^2}{p}\sum_{i=1}^{p}\frac{c + b\lambda_i}{\left(1 + a(c + b\lambda_i)\right)^2}.
\end{aligned}$$

As $p, M \to \infty$, the empirical spectral distribution of $\boldsymbol{W}_p$ converges to the Marchenko-Pastur law $\mu_\gamma$, yielding

$$\lim_{p\to\infty}\mathcal{I}'_1(p) = \frac{\rho^2}{a^2b}\int\frac{\lambda + c/b}{(\lambda + (1 + ac)/ab)^2}\,d\mu_{\mathrm{MP},\gamma}(\lambda).$$

Using Lemma 10 ($\mathcal{I}_2$ with $z_1 = z_c, z_2 = z_a$) gives

$$\lim_{p\to\infty}\mathcal{I}'_1(p) = \frac{\rho^2}{a^2b}\left(m_\gamma(z_a) - \frac{1}{ab}m'_\gamma(z_a)\right).$$

(49)

For $\mathcal{I}_2'(p)$, since $\mathbf{M}$ and $\boldsymbol{\Sigma}$ share eigenvectors, and using Eqs. (29), (31), (32) and (34), we have

$$\boldsymbol{\theta}^\top \boldsymbol{\Sigma}^{-2}\mathbf{M}^{-2}\boldsymbol{\theta} = \boldsymbol{v}^\top \boldsymbol{\Sigma}^{-1}\boldsymbol{\Sigma}^{-2}\mathbf{M}^{-2}\boldsymbol{\Sigma}^{-1}\boldsymbol{v} = \boldsymbol{v}^\top \boldsymbol{\Sigma}^{-4}\mathbf{M}^{-2}\boldsymbol{v}$$
$$= \boldsymbol{v}^\top (\boldsymbol{A}^{-4}\boldsymbol{B}^{-2}\otimes \mathbb{I})\mathrm{vec}(\boldsymbol{W}_p) = \mathrm{vec}(\boldsymbol{W}_p)^\top \mathrm{vec}(\boldsymbol{A}^{-4}\boldsymbol{B}^{-2}\boldsymbol{W}_p)$$
$$= \mathrm{tr}(\boldsymbol{W}_p^2 \boldsymbol{A}^{-4}\boldsymbol{B}^{-2}).$$

Using this, we get

$$\mathcal{I}_2'(p) = \frac{\rho}{p}\sum_{i=1}^p \frac{\lambda_i^2}{(c+b\lambda_i)^4}\frac{1}{(a+(c+b\lambda_i)^{-1})^2} = \frac{\rho}{p}\sum_{i=1}^p \frac{\lambda_i^2}{(c+b\lambda_i)^2(a(c+b\lambda_i)+1)^2}.$$

As $p, M \to \infty$, the empirical spectral distribution of $\boldsymbol{W}_p$ converges to the Marchenko-Pastur law $\mu_\gamma$, yielding

$$\lim_{p\to\infty}\mathcal{I}_2'(p) = \frac{\rho}{a^2 b^4}\int \frac{\lambda^2}{(\lambda+c/b)^2(\lambda+(1+ac)/ab)^2}\mathrm{d}\mu_{\mathrm{MP},\gamma}(\lambda)$$

Using Lemma 10 ($\mathcal{I}_7$ with $z_1 = z_a, z_2 = z_c$), we get

$$\lim_{d\to\infty}\mathcal{I}_2'(p) = \frac{\rho}{a^2 b^4}\left[\frac{2z_a z_c}{(z_a - z_c)^3}\big(m_\gamma(z_c) - m_\gamma(z_a)\big) + \frac{z_c^2}{(z_a - z_c)^2}m_\gamma'(z_c) + \frac{z_a^2}{(z_a - z_c)^2}m_\gamma'(z_a)\right].$$
$$(50)$$

Using $z_a - z_c = -1/(ab)$ to simplify Eq. (50), and substituting it with Eq. (49) in Eq. (47) gives the final expression in Eq. (46).

Next, working with $T_2(\boldsymbol{w})$, using Lemma 11, we have

$$\mathbb{E}_{\boldsymbol{w}}\left[\frac{1}{d}\boldsymbol{\theta}^\top \boldsymbol{\Sigma}(\boldsymbol{w})\boldsymbol{\theta}\right] = \frac{(1+b'+c')}{d}\boldsymbol{\theta}^\top \boldsymbol{\theta} = \frac{(1+b'+c')}{d}\boldsymbol{v}^\top \boldsymbol{\Sigma}^{-2}\boldsymbol{v}.$$

Using Eqs. (29), (31) and (34), we have

$$\boldsymbol{v}^\top \boldsymbol{\Sigma}^{-2}\boldsymbol{v} = \mathrm{vec}(\boldsymbol{W}_p)^\top \mathrm{vec}(\boldsymbol{A}^{-2}\boldsymbol{W}_p) = \mathrm{tr}(\boldsymbol{W}_p^2 (c\mathbb{I}_p + b\boldsymbol{W}_p)^{-2}).\qquad (51)$$

Therefore,

$$\mathbb{E}_{\boldsymbol{w}}\left[\frac{1}{d}\boldsymbol{\theta}^\top \boldsymbol{\Sigma}(\boldsymbol{w})\boldsymbol{\theta}\right] = \frac{(1+b'+c')\rho}{p}\sum_{i=1}^p \frac{\lambda_i^2}{(c+b\lambda_i)^2}.$$

As $p, M \to \infty$, the empirical spectral distribution of $\boldsymbol{W}_p$ converges to the Marchenko-Pastur law $\mu_\gamma$, yielding

$$\lim_{d\to\infty}\mathbb{E}_{\boldsymbol{w}}\left[\frac{1}{d}\boldsymbol{\theta}^\top \boldsymbol{\Sigma}(\boldsymbol{w})\boldsymbol{\theta}\right] = \frac{(1+b'+c')\rho}{b^2}\int \frac{\lambda^2}{(\lambda+c/b)^2}\mathrm{d}\mu_\gamma(\lambda).\qquad (52)$$

Using Lemma 10 ($\mathcal{I}_6$ with $z_1 = z_c$) then gives the expression in Eq. (44).

Next, working with $T_3(\boldsymbol{w})$, using Lemma 11, we have that

$$\mathbb{E}_{\boldsymbol{w}}[T_3(\boldsymbol{w})] = \frac{1+b'+c'}{d^{3/2}}\boldsymbol{u}^\top \mathbf{M}^{-1}\boldsymbol{\Sigma}^{-1/2}\boldsymbol{\theta}$$
$$= \frac{1+b'+c'}{d^{3/2}}(\beta\boldsymbol{g} - d^{1/2}\boldsymbol{\Sigma}^{-1/2}\boldsymbol{\theta})^\top \mathbf{M}^{-1}\boldsymbol{\Sigma}^{-1/2}\boldsymbol{\theta}$$
$$= \frac{1+b'+c'}{d}\left(\beta d^{-1/2}\boldsymbol{g}^\top \mathbf{M}^{-1}\boldsymbol{\Sigma}^{-1/2}\boldsymbol{\theta} - \boldsymbol{\theta}^\top \boldsymbol{\Sigma}^{-1/2}\mathbf{M}^{-1}\boldsymbol{\Sigma}^{-1/2}\boldsymbol{\theta}\right).$$

Using Lemma 14, we have that

$$\lim_{d\to\infty}\frac{1}{d^{3/2}}\boldsymbol{g}^\top \mathbf{M}^{-1}\boldsymbol{\Sigma}^{-1/2}\boldsymbol{\theta} = \lim_{p\to\infty}\frac{\rho^{3/2}}{p^{3/2}}\boldsymbol{g}^\top \mathbf{M}^{-1}\boldsymbol{\Sigma}^{-1/2}\boldsymbol{\theta} = 0,$$

as $\|\mathbf{M}^{-1}\mathbf{\Sigma}^{-1/2}\boldsymbol{\theta}\| = O(\|\boldsymbol{\theta}\|) = O(\|\mathbf{\Sigma}^{-1}\mathrm{vec}(\boldsymbol{W}_p)\|) = O(\sqrt{p})$.

Using this we have

$$\lim_{d\to\infty} \mathbb{E}_{\boldsymbol{w}}[T_3(\boldsymbol{w})] = -(1 + b' + c') \lim_{d\to\infty} \mathcal{I}'_3(p), \quad \mathcal{I}'_3(p) = \frac{1}{d}\boldsymbol{\theta}^\top\mathbf{\Sigma}^{-1/2}\mathbf{M}^{-1}\mathbf{\Sigma}^{-1/2}\boldsymbol{\theta}. \tag{53}$$

Since $\mathbf{\Sigma}$ and $\mathbf{M}$ share eigenvectors, and using Eqs. (29), (31), (32) and (34), we have

$$\begin{aligned}
\mathcal{I}'_3(p) &= \frac{1}{d}\boldsymbol{v}^\top\mathbf{\Sigma}^{-1}\mathbf{\Sigma}^{-1/2}\mathbf{M}^{-1}\mathbf{\Sigma}^{-1/2}\mathbf{\Sigma}^{-1}\boldsymbol{v} = \frac{1}{d}\boldsymbol{v}^\top\mathbf{\Sigma}^{-3}\mathbf{M}^{-1}\boldsymbol{v} \\
&= \frac{1}{d}\mathrm{vec}(\boldsymbol{W}_p)^\top\mathrm{vec}(\boldsymbol{A}^{-3}\boldsymbol{B}^{-1}\boldsymbol{W}_p) = \frac{1}{d}\mathrm{tr}(\boldsymbol{W}_p^2\boldsymbol{A}^{-3}\boldsymbol{B}^{-1}) \\
&= \frac{1}{d}\mathrm{tr}(\boldsymbol{W}_p^2(c\mathbb{I} + b\boldsymbol{W}_p)^{-3}(a\mathbb{I} + (c\mathbb{I} + b\boldsymbol{W}_p)^{-1})^{-1}) \\
&= \frac{\rho}{p}\sum_{i=1}^{p} \frac{\lambda_i^2}{(c + b\lambda_i)^2(a(c + b\lambda_i) + 1)}.
\end{aligned} \tag{54}$$

As $p, M \to \infty$, the empirical spectral distribution of $\boldsymbol{W}_p$ converges to the Marchenko-Pastur law $\mu_\gamma$, yielding,

$$\lim_{d\to\infty} \mathcal{I}'_3(p) = \frac{\rho}{ab^2} \int \frac{\lambda^2}{(\lambda + c/b)^2(\lambda + (1 + ac)/ab)}\, d\mu_\gamma(\lambda). \tag{55}$$

Using Lemma 10 ($\mathcal{I}_3$ with $z_1 = z_c, z_2 = z_a$) and simplifying gives the final expression in Eq. (45).
$\square$

**Lemma 5** (Term-2, OOD Loss). *Using $\boldsymbol{M}, \mathbf{\Sigma}, \boldsymbol{u}, \boldsymbol{\theta}$ of the same form as Appendix E.2, consider the scalar random variable*

$$U_p(\boldsymbol{w}) := \frac{1}{d^{3/2}}\boldsymbol{u}^\top\mathbf{M}^{-1}\mathbf{\Sigma}^{-1/2}(\boldsymbol{w} \otimes \boldsymbol{w}) + \frac{1}{d}\boldsymbol{\theta}^\top(\boldsymbol{w} \otimes \boldsymbol{w}). \tag{56}$$

*Then, almost surely,*

$$\lim_{d\to\infty} \mathbb{E}_{\boldsymbol{w}}[U_p(\boldsymbol{w})] = \frac{\rho}{b}\Big(1 + z_a m_\gamma(z_a)\Big).$$

*Proof.* Using $\mathbb{E}_{\boldsymbol{w}}[\boldsymbol{w} \otimes \boldsymbol{w}] = \mathrm{vec}(\mathbb{I}_p)$ and since $\boldsymbol{w}$ is independent of $\boldsymbol{g}$ and $\boldsymbol{W}_p$, we have that

$$\mathbb{E}_{\boldsymbol{w}}[U_p(\boldsymbol{w})] = \frac{1}{d^{3/2}}\boldsymbol{u}^\top\mathbf{M}^{-1}\mathbf{\Sigma}^{-1/2}\mathrm{vec}(\mathbb{I}_p) + \frac{1}{d}\boldsymbol{\theta}^\top\mathrm{vec}(\mathbb{I}_p).$$

Considering the first term, we have

$$\frac{1}{d^{3/2}}\boldsymbol{u}^\top\mathbf{M}^{-1}\mathbf{\Sigma}^{-1/2}\mathrm{vec}(\mathbb{I}_p) = \frac{\beta}{d^{3/2}}\boldsymbol{g}^\top\mathbf{M}^{-1}\mathbf{\Sigma}^{-1/2}\mathrm{vec}(\mathbb{I}_p) - \frac{1}{d}\boldsymbol{\theta}^\top\mathbf{\Sigma}^{-1/2}\mathbf{M}^{-1}\mathbf{\Sigma}^{-1/2}\mathrm{vec}(\mathbb{I}_p).$$

Using Lemma 14, we have that

$$\lim_{d\to\infty} \frac{\beta}{d^{3/2}}\boldsymbol{g}^\top\mathbf{M}^{-1}\mathbf{\Sigma}^{-1/2}\mathrm{vec}(\mathbb{I}_p) = \lim_{p\to\infty} \frac{\beta\rho^{3/2}}{p^{3/2}}\boldsymbol{g}^\top\mathbf{M}^{-1}\mathbf{\Sigma}^{-1/2}\mathrm{vec}(\mathbb{I}_p) = 0,$$

as $\|\mathbf{M}^{-1}\mathbf{\Sigma}^{-1}\mathrm{vec}(\mathbb{I}_p)\| = O(\sqrt{p})$. Since $\mathbf{M}$ and $\mathbf{\Sigma}$ share eigenvectors, they commute and we have that $\mathbf{\Sigma}^{-1/2}\mathbf{M}^{-1}\mathbf{\Sigma}^{-1/2} = \mathbf{\Sigma}^{-1}\mathbf{M}^{-1}$. Therefore

$$\lim_{d\to\infty} \mathbb{E}_{\boldsymbol{w}}[U_p(\boldsymbol{w})] = -\lim_{d\to\infty} \mathcal{I}'_1(p) + \lim_{d\to\infty} \mathcal{I}'_2(p) \tag{57}$$

$$\mathcal{I}'_1(p) = \frac{1}{d}\boldsymbol{\theta}^\top\mathbf{\Sigma}^{-1}\mathbf{M}^{-1}\mathrm{vec}(\mathbb{I}_p), \quad \mathcal{I}'_2(p) = \frac{1}{d}\boldsymbol{\theta}^\top\mathrm{vec}(\mathbb{I}_p). \tag{58}$$

For $\mathcal{I}'_1(p)$, using Eqs. (29), (31), (32) and (34) we have that

$$\begin{aligned}
\mathcal{I}'_1(p) &= \frac{1}{d}\boldsymbol{v}^\top\mathbf{\Sigma}^{-2}\mathbf{M}^{-1}\mathrm{vec}(\mathbb{I}_p) = \frac{1}{d}\boldsymbol{v}^\top((\boldsymbol{A}^{-2}\boldsymbol{B}^{-1}) \otimes \mathbb{I}_p)\mathrm{vec}(\mathbb{I}_p) = \frac{1}{d}\mathrm{vec}(\boldsymbol{W}_p)\mathrm{vec}(\boldsymbol{A}^{-2}\boldsymbol{B}^{-1}) \\
&= \frac{1}{d}\mathrm{tr}(\boldsymbol{W}_p\boldsymbol{A}^{-2}\boldsymbol{B}^{-1}) = \frac{\rho}{p}\sum_{i=1}^{p} \frac{\lambda_i}{(c + b\lambda_i)(a(c + b\lambda_i) + 1)}
\end{aligned}$$

As $p, M \to \infty$, the empirical spectral distribution of $\boldsymbol{W}_p$ converges to the Marchenko-Pastur law $\mu_\gamma$, yielding

$$\lim_{d \to \infty} \mathcal{I}_1'(p) = \frac{\rho}{ab^2} \int \frac{\lambda}{(\lambda + c/b)(\lambda + (1 + ac)/ab)} \mathrm{d}\mu_\gamma(\lambda).$$

Using Lemma 10 ($\mathcal{I}_4$ with $z_1 = z_c, z_2 = z_a$), and $z_c - z_a = 1/ab$ we get

$$\lim_{d \to \infty} \mathcal{I}_1'(p) = \frac{\rho}{b}(z_c m_\gamma(z_c) - z_a m_\gamma(z_a)). \tag{59}$$

Next, for $\mathcal{I}_2'(p)$, using Eqs. (29), (31), (32) and (34), we have that

$$\mathcal{I}_2'(p) = \frac{1}{d} \boldsymbol{v}^\top \boldsymbol{\Sigma}^{-1} \mathrm{vec}(\mathbb{I}_p) = \frac{1}{d} \mathrm{tr}(\boldsymbol{W}_p \boldsymbol{A}^{-1}) = \frac{\rho}{p} \sum_{i=1}^{d} \frac{\lambda_i}{c + b\lambda_i}.$$

As $p, M \to \infty$, the empirical spectral distribution of $\boldsymbol{W}_p$ converges to the Marchenko-Pastur law $\mu_\gamma$, yielding

$$\lim_{d \to \infty} \mathcal{I}_2'(p) = \frac{\rho}{b} \int \frac{\lambda}{(\lambda + c/b)} \mathrm{d}\mu_\gamma(\lambda).$$

Using Lemma 10 ($\mathcal{I}_1$ with $z_1 = z_c$), we get

$$\lim_{d \to \infty} \mathcal{I}_2'(p) = \frac{\rho}{b}(1 + z_c m_\gamma(z_c)). \tag{60}$$

Substituting Eq. (59) and Eq. (60) in Eq. (57) finishes the proof. $\qquad \square$

**Lemma 6** (Term-1, ID Loss). *Using $\boldsymbol{M}, \boldsymbol{\Sigma}, \boldsymbol{\Sigma}(\boldsymbol{w}_j), \boldsymbol{u}, \boldsymbol{\theta}$ of the same form as Appendix E.2, let*

$$T_1(\boldsymbol{w}_j) := \frac{1}{d^2} \boldsymbol{u}^\top \boldsymbol{M}^{-1} \boldsymbol{\Sigma}^{-1/2} \boldsymbol{\Sigma}(\boldsymbol{w}_j) \boldsymbol{\Sigma}^{-1/2} \boldsymbol{M}^{-1} \boldsymbol{u},$$

$$T_2(\boldsymbol{w}_j) := \frac{1}{d} \boldsymbol{\theta}^\top \boldsymbol{\Sigma}(\boldsymbol{w}_j) \boldsymbol{\theta}, \qquad T_3(\boldsymbol{w}_j) := \frac{1}{d^{3/2}} \boldsymbol{u}^\top \boldsymbol{M}^{-1} \boldsymbol{\Sigma}^{-1/2} \boldsymbol{\Sigma}(\boldsymbol{w}_j) \theta.$$

*Then, almost surely*

$$\lim_{d \to \infty} \frac{1}{M} \sum_{j=1}^{M} T_1(w_j) = (b' + c') T_{1,a} + T_{1,b},$$

$$\lim_{d \to \infty} \frac{1}{M} \sum_{j=1}^{M} T_2(w_j) = (b' + c') T_{2,a} + T_{2,b}$$

$$\lim_{d \to \infty} \frac{1}{M} \sum_{j=1}^{M} T_3(w_j) = (b' + c') T_{3,a} + T_{3,b}$$

*where $T_{1,a} = J_1$ from Eq. (46),*

$$T_{1,b} = \frac{\rho}{b^2} \left[ -2abz_a z_c(z_c m_\gamma(z_c) - z_a m_\gamma(z_a)) + z_c^2(m_\gamma(z_c) + z_c m_\gamma'(z_c)) + z_a^2(m_\gamma(z_a) + z_a m_\gamma'(z_a)) \right]$$

$$+ \frac{\rho^2 \beta^2}{a^2 b} \left( 1 + \left( z_a - \frac{1}{ab} \right) m_\gamma(z_a) - \frac{1}{ab} z_a m_\gamma'(z_a) \right),$$

$$T_{2,a} = \frac{\rho}{b^2} \left( 1 + 2z_c m_\gamma(z_c) + z_c^2 m_\gamma'(z_c) \right),$$

$$T_{2,b} = \frac{\rho}{b^2} \left( 1 + 2z_c + 3z_c^2 m_\gamma(z_c) + z_c^3 m_\gamma'(z_c) \right),$$

$$T_{3,a} = \rho \left( -\frac{c(ac + 2)}{b^3} m_\gamma(z_c) + \frac{a}{b} z_a^2 m_\gamma(z_a) + \frac{1}{b^2} z_c^2 m_\gamma'(z_c) \right),$$

$$T_{3,b} = \rho \left( -\frac{c(ac + 2)}{b^3}(1 + z_c m_\gamma(z_c)) + \frac{a}{b} z_a^2(1 + z_a m_\gamma(z_a)) + \frac{1}{b^2} z_c^2(m_\gamma(z_c) + z_c m_\gamma'(z_c)) \right).$$

*Proof.* Recall
$$\boldsymbol{\Sigma}(\boldsymbol{w}_j) = \Big(c'\frac{\|\boldsymbol{w}_j\|^2}{d} + b'\Big)\mathbb{I}_{p^2} + (\boldsymbol{w}_j[\mathcal{S}]\boldsymbol{w}_j[\mathcal{S}]^\top) \otimes \mathbb{I}_p.$$

We have

$$T_1(w_j) = \frac{1}{d^2}\boldsymbol{u}^\top\mathbf{M}^{-1}\boldsymbol{\Sigma}^{-1/2}\boldsymbol{\Sigma}(\boldsymbol{w}_j)\boldsymbol{\Sigma}^{-1/2}\mathbf{M}^{-1}\boldsymbol{u}$$

$$= \underbrace{\frac{1}{d^2}\boldsymbol{u}^\top\mathbf{M}^{-1}\boldsymbol{\Sigma}^{-1/2}\Big(c'\frac{\|\boldsymbol{w}_j\|^2}{d} + b'\Big)\boldsymbol{\Sigma}^{-1/2}\mathbf{M}^{-1}\boldsymbol{u}}_{T_{1,a}(\boldsymbol{w}_j)} + \underbrace{\frac{1}{d^2}\boldsymbol{u}^\top\mathbf{M}^{-1}\boldsymbol{\Sigma}^{-1/2}((\boldsymbol{w}_j[\mathcal{S}]\boldsymbol{w}_j[\mathcal{S}]^\top) \otimes \mathbb{I}_p)\boldsymbol{\Sigma}^{-1/2}\mathbf{M}^{-1}\boldsymbol{u}}_{T_{1,b}(\boldsymbol{w}_j)}.$$

Let us first consider the limit of $T_{1,a}(\boldsymbol{w}_j)$. We have

$$\lim_{d\to\infty}\frac{1}{d^2M}\sum_{j=1}^M T_{1,a}(\boldsymbol{w}_j) = \lim_{d\to\infty}\frac{1}{d^2M}\sum_{i=1}^M \boldsymbol{u}^\top\mathbf{M}^{-1}\boldsymbol{\Sigma}^{-1/2}\Big(c'\frac{\|\boldsymbol{w}_j\|^2}{d} + b'\Big)\boldsymbol{\Sigma}^{-1/2}\mathbf{M}^{-1}\boldsymbol{u}$$

$$= (b' + c')\lim_{d\to\infty}\frac{1}{d^2}\boldsymbol{u}^\top\mathbf{M}^{-1}\boldsymbol{\Sigma}^{-1}\mathbf{M}^{-1}\boldsymbol{u}, \tag{61}$$

where we use the fact that $\lim_{d\to\infty}\|\boldsymbol{w}\|^2/d = 1$ for $\boldsymbol{w} \sim \mathcal{N}(\mathbf{0}, \mathbb{I}_d)$. The above limit is same as Eq. (46) derived in OOD loss.

Next, we solve the limit of $T_{1,b}(\boldsymbol{w}_j)$. We have

$$\lim_{d\to\infty}\frac{1}{d^2M}\sum_{j=1}^M T_{1,b}(\boldsymbol{w}_j) = \lim_{d\to\infty}\frac{1}{d^2M}\sum_{i=1}^M \boldsymbol{u}^\top\mathbf{M}^{-1}\boldsymbol{\Sigma}^{-1/2}((\boldsymbol{w}_j[\mathcal{S}]\boldsymbol{w}_j[\mathcal{S}]^\top) \otimes \mathbb{I}_p)\boldsymbol{\Sigma}^{-1/2}\mathbf{M}^{-1}\boldsymbol{u}$$

$$= \lim_{d\to\infty}\frac{1}{d^2}\boldsymbol{u}^\top\mathbf{M}^{-1}\boldsymbol{\Sigma}^{-1/2}\Big(\frac{1}{M}\sum_{j=1}^M(\boldsymbol{w}_j[\mathcal{S}]\boldsymbol{w}_j[\mathcal{S}]^\top) \otimes \mathbb{I}_p\Big)\boldsymbol{\Sigma}^{-1/2}\mathbf{M}^{-1}\boldsymbol{u}$$

$$= \lim_{d\to\infty}\frac{1}{d^2}\boldsymbol{u}^\top\mathbf{M}^{-1}\boldsymbol{\Sigma}^{-1/2}(\boldsymbol{W}_p \otimes \mathbb{I}_p)\boldsymbol{\Sigma}^{-1/2}\mathbf{M}^{-1}\boldsymbol{u}.$$

Using the definitions of $\boldsymbol{\Sigma}, \mathbf{M}$ we have

$$\boldsymbol{u}^\top\mathbf{M}^{-1}\boldsymbol{\Sigma}^{-1/2}(\boldsymbol{W}_p \otimes \mathbb{I}_p)\boldsymbol{\Sigma}^{-1/2}\mathbf{M}^{-1}\boldsymbol{u} = \boldsymbol{u}^\top\boldsymbol{\Sigma}^{-1}\boldsymbol{M}^{-2}\boldsymbol{W}_p \otimes \mathbb{I}_p\boldsymbol{u},$$

where we use the fact that $\boldsymbol{\Sigma}, \boldsymbol{M}, (\boldsymbol{W}_p \otimes \mathbb{I}_p)$ share Eigenvectors and hence commute. Plugging the definition of $\boldsymbol{u}$, we have

$$\boldsymbol{u}^\top\boldsymbol{M}^{-2}\boldsymbol{\Sigma}^{-1}\boldsymbol{W}_p \otimes \mathbb{I}_p\boldsymbol{u} = \beta^2\boldsymbol{g}^\top\boldsymbol{\Sigma}^{-1}\boldsymbol{M}^{-2}\boldsymbol{W}_p \otimes \mathbb{I}_p\boldsymbol{g} + d\boldsymbol{\theta}^\top\boldsymbol{\Sigma}^{-2}\boldsymbol{M}^{-2}\boldsymbol{W}_p \otimes \mathbb{I}_p\boldsymbol{\theta}$$
$$- 2\beta d^{1/2}\boldsymbol{g}^\top\boldsymbol{\Sigma}^{-3/2}\boldsymbol{M}^{-2}\boldsymbol{W}_p \otimes \mathbb{I}_p\boldsymbol{\theta}.$$

Following similar steps to Eq. (48),

$$\lim_{d\to\infty}\frac{1}{d^{3/2}}\boldsymbol{g}^\top\boldsymbol{\Sigma}^{-3/2}\boldsymbol{M}^{-2}\boldsymbol{W}_p \otimes \mathbb{I}_p\boldsymbol{\theta} = \lim_{p\to\infty}\frac{\rho^{3/2}}{p^{3/2}}\boldsymbol{g}^\top\boldsymbol{\Sigma}^{-3/2}\boldsymbol{M}^{-2}\boldsymbol{W}_p \otimes \mathbb{I}_p\boldsymbol{\theta} = 0,$$

Using Lemma 14 again, we get

$$\frac{1}{d^2}\lim_{d\to\infty}\boldsymbol{g}^\top(\boldsymbol{\Sigma}^{-1}\boldsymbol{M}^{-2}\boldsymbol{W}_p \otimes \mathbb{I}_p)\boldsymbol{g} = \lim_{p\to\infty}\frac{\rho^2}{p^2}\operatorname{tr}(\boldsymbol{\Sigma}^{-1}\boldsymbol{M}^{-2}\boldsymbol{W}_p \otimes \mathbb{I}_p).$$

Following similar steps to the calculation of Eq. (49) in OOD loss, we get the following integral (with an additional $\lambda$) factor in the limit $p, M \to \infty$

$$\lim_{p\to\infty}\frac{\rho^2}{p^2}\operatorname{tr}(\boldsymbol{\Sigma}^{-1}\boldsymbol{M}^{-2}\boldsymbol{W}_p \otimes \mathbb{I}_p) = \frac{\rho^2}{a^2b}\int\lambda\frac{\lambda + c/b}{(\lambda + (1+ac)/ab)^2}\,d\mu_{\mathrm{MP},\gamma}(\lambda).$$

Simplifying this using result in Eq. (49) and Lemma 10 (v) we get

$$\frac{\rho^2}{a^2b}\int\lambda\frac{\lambda + c/b}{(\lambda + (1+ac)/ab)^2}\,d\mu_{\mathrm{MP},\gamma}(\lambda) = \frac{\rho^2}{a^2b}\left(1 + \left(z_a - \frac{1}{ab}\right)m_\gamma(z_a) - \frac{1}{ab}z_a m'_\gamma(z_a)\right).$$

Again, using similar steps to result derived in Eq. (50), we get

$$\lim_{d\to\infty} \frac{1}{d}\boldsymbol{\theta}^\top \boldsymbol{\Sigma}^{-2} \boldsymbol{M}^{-2} \boldsymbol{W}_p \otimes \mathbb{I}_p \boldsymbol{\theta} = \frac{\rho}{a^2 b^4} \int \lambda \frac{\lambda^2}{(\lambda + c/b)^2(\lambda + (1+ac)/ab)^2} \mathrm{d}\mu_{\mathrm{MP},\gamma}(\lambda)$$

Using Lemma 10 (v) with Eq. (50) gives the following expression for the above integral

$$\frac{\rho}{a^2 b^4} \left[ \frac{2z_a z_c}{(z_a - z_c)^3}(z_c m_\gamma(z_c) - z_a m_\gamma(z_a)) + \frac{z_c^2}{(z_a - z_c)^2}(m_\gamma(z_c) + z_c m_\gamma'(z_c)) \right. $$
$$\left. + \frac{z_a^2}{(z_a - z_c)^2}(m_\gamma(z_a) + z_a m_\gamma'(z_a)) \right] \qquad (62)$$

Combining Eq. (61) and Eq. (62), gives the result. Next, we consider $T_2(\boldsymbol{w}_j)$. It is easy to see that it decomposes to the following similar to $T_1(\boldsymbol{w}_j)$

$$T_2(\boldsymbol{w}_j) = \frac{1}{d}\boldsymbol{\theta}^\top \boldsymbol{\Sigma}(\boldsymbol{w}_j)\boldsymbol{\theta}$$
$$= \underbrace{\frac{1}{d}\boldsymbol{\theta}\left(c'\frac{\|\boldsymbol{w}_j\|^2}{d} + b'\right)\boldsymbol{\theta}}_{T_{2,a}(\boldsymbol{w}_j)} + \underbrace{\frac{1}{d}\boldsymbol{\theta}^\top((\boldsymbol{w}_j[\mathcal{S}]\boldsymbol{w}_j[\mathcal{S}]^\top) \otimes \mathbb{I}_p)\boldsymbol{\theta}}_{T_{2,b}(\boldsymbol{w}_j)}.$$

The limiting average $\lim_{d\to\infty} \frac{1}{d^2 M}\sum_{j=1}^M T_2(\boldsymbol{w}_j)$, similar to the average for $T_1(\boldsymbol{w}_j)$, it is easy to see that $\lim_{d\to\infty} \frac{1}{d^2 M}\sum_{j=1}^M T_{2,a}(\boldsymbol{w}_j)$ uses the same integral as derived in Eq. (44) for the OOD loss. The term $T_{2,b}(\boldsymbol{w}_j)$ average in the limit

$$\lim_{d\to\infty} \frac{1}{dM}\sum_{j=1}^M T_{2,b}(\boldsymbol{w}_j) = \lim_{d\to\infty} \frac{1}{d}\boldsymbol{\theta}^\top \left(\frac{1}{M}\sum_{j=1}^M (\boldsymbol{w}_j[\mathcal{S}]\boldsymbol{w}_j[\mathcal{S}]^\top) \otimes \mathbb{I}_p\right)\boldsymbol{\theta}$$
$$= \lim_{d\to\infty} \frac{1}{d}\boldsymbol{\theta}^\top(\boldsymbol{W}_p \otimes \mathbb{I}_p)\boldsymbol{\theta}.$$

Following similar steps to the calculation in Eq. (51), we get the following trace expression (with an additional $\boldsymbol{W}_p$ factor)

$$\lim_{d\to\infty} \frac{1}{d}\boldsymbol{\theta}^\top(\boldsymbol{W}_p \otimes \mathbb{I}_p)\boldsymbol{\theta} = \lim_{d\to\infty} \frac{1}{d}\mathrm{tr}(\boldsymbol{W}_p^3(c\mathbb{I}_p + b\boldsymbol{W}_p)^{-2})$$
$$= \rho \int \lambda \frac{\lambda^2}{(\lambda + c/b)^2}\mathrm{d}\mu_\gamma(\lambda),$$
$$= \frac{\rho}{b^2}\left(1 + 2z_c m_\gamma(z_c) + 3z_c^2 m_\gamma(z_c) + z_c^3 m_\gamma'(z_c)\right).$$

where the integral follows from calculation in Eq. (52). The final equality follows by combining Lemma 10 (v) with integral solved in Eq. (52).

Next, for $T_3(\boldsymbol{w}_j)$, we get a similar decomposition to $T_1(\boldsymbol{w}_j)$ and $T_2(\boldsymbol{w}_j)$

$$T_3(\boldsymbol{w}_j) = \underbrace{\boldsymbol{u}^\top \mathbf{M}^{-1}\left(c'\frac{\|\boldsymbol{w}_j\|^2}{d} + b'\right)\boldsymbol{\theta}}_{T_{3,a}(\boldsymbol{w}_j)} + \underbrace{\boldsymbol{u}^\top \mathbf{M}^{-1}((\boldsymbol{w}_j[\mathcal{S}]\boldsymbol{w}_j[\mathcal{S}]^\top) \otimes \mathbb{I}_p)\boldsymbol{\theta}}_{T_{3,b}(\boldsymbol{w}_j)}.$$

The limiting average $\lim_{d\to\infty} \frac{1}{d^2 M}\sum_{j=1}^M T_3(\boldsymbol{w}_j)$, similar to the limiting averages computed for $T_1(\boldsymbol{w}_j)$ and $T_2(\boldsymbol{w}_j)$, it is easy to see that $\lim_{d\to\infty} \frac{1}{d^2 M}\sum_{j=1}^M T_{3,a}(\boldsymbol{w}_j)$ uses the integral Eq. (55).

For $\lim_{d\to\infty} \frac{1}{d^2 M} \sum_{j=1}^M T_{3,b}(\boldsymbol{w}_j)$, following similar calculations to Eq. (54), we have the expression below (with an additional $\boldsymbol{W}_p$ factor)

$$\lim_{d\to\infty} \frac{1}{d^2 M} \sum_{j=1}^M T_{3,b}(\boldsymbol{w}_j) = \frac{1}{d} \operatorname{tr}(\boldsymbol{W}_p^3 \boldsymbol{A}^{-3} \boldsymbol{B}^{-1})$$

$$= \frac{\rho}{ab^2} \int \lambda \frac{\lambda^2}{(\lambda + c/b)^2 (\lambda + (1+ac)/ab)} \, d\mu_\gamma(\lambda)$$

$$= \rho \left( -\frac{c(ac+2)}{b^3}(1 + z_c m_\gamma(z_c)) + \frac{a}{b} z_a^2 (1 + z_a m_\gamma(z_a)) + \frac{1}{b^2} z_c^2 (m_\gamma(z_c) + z_c m'_\gamma(z_c)) \right),$$

where the second equality follows similar steps from Eq. (54) to Eq. (55), and the last inequality follows by combining the result in Eq. (55) with Lemma 10 (v). $\qquad\square$

**Lemma 7** (Term-2, ID Loss). *Using $\boldsymbol{M}, \boldsymbol{\Sigma}, \boldsymbol{\Sigma}(\boldsymbol{w}), \boldsymbol{u}, \boldsymbol{\theta}$ of the same form as Appendix E.2, let*

$$T_2(\boldsymbol{w}_j) := \frac{1}{d^{3/2}} \boldsymbol{u}^\top \mathbf{M}^{-1} \boldsymbol{\Sigma}^{-1/2}(\boldsymbol{w}_j[\mathcal{S}] \otimes \boldsymbol{w}_j[\mathcal{S}]) + \frac{1}{d} \boldsymbol{\theta}^\top (\boldsymbol{w}_j[\mathcal{S}] \otimes \boldsymbol{w}_j[\mathcal{S}]).$$

*Then, almost surely*

$$\lim_{d\to\infty} \frac{1}{M} \sum_{j=1}^M T_2(\boldsymbol{w}_j) = -\frac{\rho}{b}(z_c(1 + z_c m_\gamma(z_c)) - z_a(1 + z_a m_\gamma(z_a))) + \frac{\rho}{b}(1 + z_c(1 + z_c m_\gamma(z_c)))$$

*Proof.* Consider the average of the first term. Using $\frac{1}{M} \sum_{j=1}^M \boldsymbol{w}_j[\mathcal{S}] \otimes \boldsymbol{w}_j[\mathcal{S}] = \operatorname{vec}(\boldsymbol{W}_p)$, we have

$$\frac{1}{M d^{3/2}} \sum_{j=1}^M \boldsymbol{u}^\top \mathbf{M}^{-1} \boldsymbol{\Sigma}^{-1/2}(\boldsymbol{w}_j[\mathcal{S}] \otimes \boldsymbol{w}_j[\mathcal{S}]) = \frac{\beta}{d^{3/2}} \boldsymbol{g}^\top \mathbf{M}^{-1} \boldsymbol{\Sigma}^{-1/2}\operatorname{vec}(\boldsymbol{W}_p) - \frac{1}{d} \boldsymbol{\theta}^\top \boldsymbol{\Sigma}^{-1/2} \mathbf{M}^{-1} \boldsymbol{\Sigma}^{-1/2}\operatorname{vec}(\boldsymbol{W}_p).$$

Term 1 above is 0 in the limit $d \to \infty$, using Lemma 14. Term 2 simplifies to

$$\frac{1}{d} \boldsymbol{\theta}^\top \boldsymbol{\Sigma}^{-1/2} \mathbf{M}^{-1} \boldsymbol{\Sigma}^{-1/2}\operatorname{vec}(\boldsymbol{W}_p) = \frac{1}{d}\operatorname{vec}(\boldsymbol{W}_p)^\top \boldsymbol{\Sigma}^{-2} \mathbf{M}^{-1}\operatorname{vec}(\boldsymbol{W}_p) = \frac{1}{d}\operatorname{vec}(\boldsymbol{W}_p)^\top \boldsymbol{A}^{-2} \boldsymbol{B}^{-1} \otimes \mathbb{I}_p \operatorname{vec}(\boldsymbol{W}_p)$$

$$= \frac{1}{d}\operatorname{vec}(\boldsymbol{W}_p)^\top \operatorname{vec}(\boldsymbol{A}^{-2}\boldsymbol{B}^{-1}\boldsymbol{W}_p)$$

$$= \frac{1}{d}\operatorname{tr}(\boldsymbol{W}_p^2 \boldsymbol{A}^{-2}\boldsymbol{B}^{-1}).$$

Here, second equality follows from the definition of $\boldsymbol{\theta}$ and the fact that $\mathbf{M}$ and $\boldsymbol{\Sigma}$ commute. Third and fourth equality use the definition of $\boldsymbol{\Sigma}, \mathbf{M}$ and Eq. (31) and Eq. (32). In the limit

$$\lim_{d\to\infty} \frac{1}{d}\operatorname{tr}(\boldsymbol{W}_p^2 \boldsymbol{A}^{-2}\boldsymbol{B}^{-1}) = \frac{\rho}{ab^2} \int \frac{\lambda}{(\lambda + c/b)(\lambda + (1+ac)/ab)} d\mu_\gamma(\lambda)$$

$$= \frac{\rho}{b}(z_c(1 + z_c m_\gamma(z_c)) - z_a(1 + z_a m_\gamma(z_a))),$$

where second equality follows by Lemma 10 with Eq. (59). Next, for term 2

$$\lim_{d\to\infty} \frac{1}{d}\boldsymbol{\theta}^\top \operatorname{vec}(\boldsymbol{W}_p) = \lim_{d\to\infty} \frac{1}{d}\operatorname{tr}(\boldsymbol{W}_p^2 \boldsymbol{A}^{-1})$$

$$= \frac{\rho}{b} \int \lambda \frac{\lambda}{(\lambda + c/b)} d\mu_\gamma(\lambda)$$

$$= \frac{\rho}{b}(1 + z_c(1 + z_c m_\gamma(z_c))).$$

The first equality follows by using definition of $\boldsymbol{\theta}, \boldsymbol{\Sigma}$ and Eq. (31) and Eq. (32). Second equality follows by the fact the empirical distribution of $\boldsymbol{W}_p$ converges to Marchenko-Pastur law $\mu_\gamma$ in the limit, and the final equality uses Lemma 10 with Eq. (60). $\qquad\square$

We restate Theorem 3 below followed by the proof.

**Theorem 6** (Asymptotic Risk of Minimum-Norm LGP). *Consider the minimum norm estimator $\hat{\boldsymbol{\theta}}_{\mathcal{S}_p}$ for the linear Gaussian equivalent problem in Definition 3 under the overparameterized regime ($\nu < \rho^2$). Under the asymptotic scaling defined in Eq. (1), as $d \to \infty$, the ID and OOD risks converge in probability to the following deterministic limits:*

$$\mathcal{L}_{OOD}(\hat{\boldsymbol{\theta}}_{\mathcal{S}_p}) \xrightarrow{P} \left(1 - \tfrac{\nu}{\rho^2}\right)\rho(1+ac)(1-c(1+a+ac)) + (1+a+ac)(\sigma_n^2 + \rho c + 1 - \rho), \quad (63)$$

$$\mathcal{L}_{ID}(\hat{\boldsymbol{\theta}}_{\mathcal{S}_p}) \xrightarrow{P} \frac{\nu}{a^2}\bar{\beta}^2. \qquad (64)$$

*Proof.* From Eq. (18) and using Eqs. (19) to (22), we have

$$\boldsymbol{a} = \boldsymbol{M}^{-1}(\beta \boldsymbol{g} - \boldsymbol{\Sigma}_{\boldsymbol{H}_{\mathcal{S}_p}}^{-1/2}\boldsymbol{\theta}_\star) = \frac{1}{d}\overline{\boldsymbol{M}}^{-1}(\bar{\beta}\boldsymbol{g} - d^{1/2}\overline{\boldsymbol{\Sigma}}_{\boldsymbol{H}_{\mathcal{S}_p}}^{-1/2}\overline{\boldsymbol{\theta}}_\star).$$

Substituting this in Lemma 1, we get Term-1(a) as

$$\frac{1}{d^2}(\bar{\beta}_*\boldsymbol{g} - d^{1/2}\overline{\boldsymbol{\Sigma}}_{\boldsymbol{H}_{\mathcal{S}_p}}^{-1/2}\overline{\boldsymbol{\theta}}_\star)^\top \overline{\boldsymbol{M}}^{-1}\overline{\boldsymbol{\Sigma}}_{\boldsymbol{H}_{\mathcal{S}_p}}^{-1/2}\overline{\boldsymbol{\Sigma}}(\boldsymbol{w})\overline{\boldsymbol{\Sigma}}_{\boldsymbol{H}_{\mathcal{S}_p}}^{-1/2}\overline{\boldsymbol{M}}^{-1}(\bar{\beta}_*\boldsymbol{g} - d^{1/2}\overline{\boldsymbol{\Sigma}}_{\boldsymbol{H}_{\mathcal{S}_p}}^{-1/2}\overline{\boldsymbol{\theta}}_\star).$$

Similarly, Term-1(c) is

$$\boldsymbol{a}^\top \boldsymbol{\Sigma}_{\boldsymbol{H}_{\mathcal{S}_p}}^{-1/2}\boldsymbol{\Sigma}(\boldsymbol{w})\boldsymbol{\theta}_\star = \frac{1}{d^{3/2}}(\bar{\beta}_*\boldsymbol{g} - d^{1/2}\overline{\boldsymbol{\Sigma}}_{\boldsymbol{H}_{\mathcal{S}_p}}^{-1/2}\overline{\boldsymbol{\theta}}_\star)^\top \overline{\boldsymbol{M}}^{-1}\overline{\boldsymbol{\Sigma}}_{\boldsymbol{H}_{\mathcal{S}_p}}^{-1/2}\overline{\boldsymbol{\Sigma}}(\boldsymbol{w})\overline{\boldsymbol{\theta}}_\star$$

Similarly, Term-2 is

$$\hat{\boldsymbol{\theta}}_{\mathcal{S}_p}^\top \mathbb{E}_{X,y}[\mathbf{h}_{\mathcal{S}_p}(X)y_q] = \frac{1}{d^{3/2}}(\bar{\beta}_*\boldsymbol{g} - d^{1/2}\overline{\boldsymbol{\Sigma}}_{\boldsymbol{H}_{\mathcal{S}_p}}^{-1/2}\overline{\boldsymbol{\theta}}_\star)^\top \overline{\boldsymbol{M}}^{-1}\overline{\boldsymbol{\Sigma}}_{\boldsymbol{H}_{\mathcal{S}_p}}^{-1/2}(\boldsymbol{w} \otimes \boldsymbol{w}) + \frac{1}{d}\overline{\boldsymbol{\theta}}_\star^\top \boldsymbol{w} \otimes \boldsymbol{w}.$$

**OOD Loss.** Using these, and Lemma 4 and Lemma 5 with $\boldsymbol{\Sigma} = \overline{\boldsymbol{\Sigma}}_{\boldsymbol{H}_{\mathcal{S}_p}}$, $\boldsymbol{\theta} = \overline{\boldsymbol{\theta}}_\star$, $\mathbf{M} = \overline{\boldsymbol{M}}$, $b = 1$, $b' + c' = c =$, $\gamma = \mu^{-1}$, $z_a = -\frac{1}{a} - c$, $z_c = -c$, and defining $m_a := m_\gamma(z_a)$, $m_a' := m_\gamma'(z_a)$, $m_c := m_\gamma(-c)$, $m_c' := m_\gamma'(-c)$, we have

$$\mathcal{L}_{\text{OOD}}(\hat{\boldsymbol{\theta}}_{\mathcal{S}_p}) = \underbrace{(1+c)\left(\bar{\beta}^2\rho^2\left(\frac{1}{a^2}m_a - \frac{1}{a^3}m_a'\right) + \rho(c^2 m_c' + 2z_a ac(m_c - m_a) + z_a^2 m_a')\right)}_{\text{Term-1(a)}}$$

$$+ \underbrace{(1+c)\rho\left(1 - 2cm_c + c^2\,m_c'\right)}_{\text{Term-1(b)}} \underbrace{-2(1+c)\rho\left[-c(ac+2)m_c + az_a^2 m_a + c^2 m_c'\right]}_{\text{Term-1(c)}}$$

$$\underbrace{-2\rho(1 + z_a m_a)}_{\text{Term-2}} + \ 1 + \sigma_n^2.$$

Using the expression for $\bar{\beta}$ from Eq. (35), Term-1(a) can be simplified as:

$$(1+c)\left(a(\sigma_n^2 + 1 - \rho + \rho c) + \rho z_a^2 a m_a + \rho c^2 m_c' - 2\rho(1+ac)cm_c\right)$$

Combining all terms, we get

$$\mathcal{L}_{\text{OOD}}(\hat{\boldsymbol{\theta}}_{\mathcal{S}_p}) = m_a(-z_a)\rho(-az_a(1+c) + 2az_a(1+c) + 2) + m_c(1+c)\rho(2z_a ac - 2c + 2c(ac+2))$$
$$+ m_c'(1+c)\rho c^2(1+1-2) + (1+c)a(\sigma_n^2 + 1 - \rho + \rho c) + \rho(1+c-2) + 1 + \sigma_n^2.$$

Simplifying this and using $m_a = a(1 - \nu/\rho^2)$, we get Eq. (63).

**ID Loss.** Using the expressions for Term-1(a), Term-1(c) and Term-2 computed above, Lemmas 1, 6 and 7, we have

$$\mathcal{L}_{\text{ID}}(\hat{\boldsymbol{\theta}}_{\mathcal{S}_p}) = \underbrace{c\Big( \bar{\beta}^2 \rho^2 \Big( \frac{1}{a^2} m_a - \frac{1}{a^3} m_a' \Big) + \rho(c^2 m_c' + 2z_a ac(m_c - m_a) + z_a^2 m_a') \Big)}_{\text{Term-1(a)-old part}}$$

$$+ \underbrace{\bar{\beta}^2 \rho^2 \Big( \frac{1}{a^2}(1 + z_a m_a) - \frac{1}{a^3}(m_a + z_a m_a') \Big) + \rho(c^2(m_c + z_c m_c') + 2z_a ac(z_c m_c - z_a m_a) + z_a^2(m_a + z_a m_a'))}_{\text{Term-1(a)-new part}}$$

$$+ \underbrace{c\rho\left(1 - 2cm_c + c^2 m_c'\right)}_{\text{Term-1(b)-old}} + \underbrace{\rho(1 - 2c + 3c^2 m_c - c^3 m_c')}_{\text{Term-1(b)-new}}$$

$$- \underbrace{2c\rho\left[-c(ac+2)m_c + az_a^2 m_a + c^2 m_c'\right]}_{\text{Term-1(c)-old}} - \underbrace{2\rho\Big[a^{-1} + az_a^3 m_a + (3c^2 + ac^3)m_c - c^3 m_c'\Big]}_{\text{Term-1(c)-new}}$$

$$\underbrace{-2\rho(1 - c + c^2 m_c - a^{-1} - z_c^2 m_c + z_a^2 m_a)}_{\text{Term-2-new}} + 1 + \sigma_n^2.$$

Combining terms, we have

$$\mathcal{L}_{\text{ID}}(\hat{\boldsymbol{\theta}}_{\mathcal{S}_p}) = \bar{\beta}^2 \rho^2 \frac{1}{a^2} \left( (z_a + c)\left( m_a - \frac{1}{a} m_a' \right) + 1 - \frac{1}{a} m_a \right) + m_a \rho z_a \left( -2ac^2 - 2z_a ac + z_a - 2acz_a - 2az_a^2 - 2z_a \right)$$

$$+ \rho m_a' \left( cz_a^2 + z_a^3 \right) + \rho m_c \left( 2ac^2 z_a + c^2 - 2z_a ac^2 - 2c^2 + 3c^2 + 2c^2(ac+2) - 2(3c^2 + ac^3) \right)$$

$$+ \rho m_c' \left( c^3 - c^3 + c^3 - c^3 - 2c^3 + 2c^3 \right) + 1 + \sigma_n^2 + \rho\left( -2(1-c) + 1 - 2c + c \right)$$

$$= \bar{\beta}^2 \rho^2 \frac{1}{a^2} \left( -\frac{1}{a}\left( m_a - \frac{1}{a} m_a' \right) + \frac{\nu}{\rho^2} \right) + \frac{\rho(1 - a^2 c^2)}{a^2} m_a - \frac{\rho z_a^2}{a} m_a' + \sigma_n^2 + 1 - \rho + \rho c.$$

Using the expression for $\bar{\beta}$ from Eq. (35), we simplify and get Eq. (64).

$\square$

## G.2 ASYMPTOTIC RISK OF LS LGP

**Lemma 8** (OOD Loss). *Using* $\boldsymbol{\Sigma}, \boldsymbol{\Sigma}(\boldsymbol{w}), \boldsymbol{\theta}$ *of the same form as Appendix E.2, let*

$$T_1(\boldsymbol{w}) := \frac{1}{\|\boldsymbol{g}\|^2} \boldsymbol{g}^\top \boldsymbol{\Sigma}^{-1/2} \boldsymbol{\Sigma}(\boldsymbol{w}) \boldsymbol{\Sigma}^{-1/2} \boldsymbol{g},$$

$$T_2(\boldsymbol{w}) := \frac{1}{\sqrt{d}\|\boldsymbol{g}\|} \boldsymbol{g}^\top \boldsymbol{\Sigma}^{-1/2} \boldsymbol{\Sigma}(\boldsymbol{w}) \boldsymbol{\theta}, \qquad T_3(\boldsymbol{w}) := \frac{1}{\sqrt{d}\|\boldsymbol{g}\|} \boldsymbol{g}^\top \boldsymbol{\Sigma}^{-1/2} (\boldsymbol{w} \otimes \boldsymbol{w}).$$

*then*

$$\lim_{d\to\infty} \mathbb{E}_{\boldsymbol{w}}[T_1(\boldsymbol{w})] = \frac{1 + b' + c'}{b} m_\gamma(z_c), \qquad \lim_{d\to\infty} \mathbb{E}_{\boldsymbol{w}}[T_2(\boldsymbol{w})] = 0, \qquad \lim_{d\to\infty} \mathbb{E}_{\boldsymbol{w}}[T_3(\boldsymbol{w})] = 0.$$

*Proof.* We first work with $T_1(\boldsymbol{w})$. Using Lemma 11, since $\boldsymbol{w}$ is independent of $\boldsymbol{g}, \boldsymbol{W}_p$, we get

$$\mathbb{E}_{\boldsymbol{w}}[T_1(\boldsymbol{w})] = \frac{(c' + b' + 1)}{\|\boldsymbol{g}\|^2} \boldsymbol{g}^\top \boldsymbol{\Sigma}^{-1} \boldsymbol{g}.$$

Using Lemma 14, and since $\lim_{d\to\infty} \frac{p}{\|\boldsymbol{g}\|} = 1$, we have that

$$\lim_{d\to\infty} \frac{1}{\|\boldsymbol{g}\|^2} \boldsymbol{g}^\top \boldsymbol{\Sigma}^{-1} \boldsymbol{g} = \lim_{d\to\infty} \frac{1}{p^2} \operatorname{tr}(\boldsymbol{\Sigma}^{-1}).$$

Using Eqs. (29) and (30), we have

$$\frac{1}{p^2} \operatorname{tr}(\boldsymbol{\Sigma}^{-1}) = \frac{1}{p} \operatorname{tr}(\boldsymbol{A}^{-1}) = \frac{1}{p} \sum_{i=1}^{p} \frac{1}{c + b\lambda_i}. \tag{65}$$

As $p \to \infty$, the empirical spectral distribution of $\boldsymbol{W}_p$ converges to the Marchenko-Pastur law $\mu_\gamma$, yielding

$$\lim_{d \to \infty} \mathbb{E}_{\boldsymbol{w}}[T_1(\boldsymbol{w})] = \frac{1}{b} \int \frac{1}{\lambda + c/b} \mathrm{d}\mu_\gamma(\lambda) = \frac{1}{b} m_\gamma(z_c).$$

Next, for $T_2(\boldsymbol{w})$, using Lemma 11, since $\boldsymbol{w}$ is independent of $\boldsymbol{g}, \boldsymbol{W}_p$, we have that

$$\mathbb{E}_{\boldsymbol{w}}[T_2(\boldsymbol{w})] = \frac{(c' + b' + 1)}{\sqrt{d} \|\boldsymbol{g}\|} \boldsymbol{g}^\top \boldsymbol{\Sigma}^{-1/2} \boldsymbol{\theta}.$$

Using Lemma 14, it follows that

$$\lim_{d \to \infty} \frac{1}{p^{3/2}} \boldsymbol{g}^\top \boldsymbol{\Sigma}^{-1/2} \boldsymbol{\theta} = 0,$$

which concludes the proof for the second part.

Finally, working with $T_3(\boldsymbol{w})$, Using $\mathbb{E}_{\boldsymbol{w}}[\boldsymbol{w} \otimes \boldsymbol{w}] = \mathtt{vec}(\mathbb{I}_p)$ and since $\boldsymbol{w}$ is independent of $\boldsymbol{g}$ and $\boldsymbol{W}_p$, we have that

$$\mathbb{E}_{\boldsymbol{w}}[T_3(\boldsymbol{w})] = \frac{1}{\sqrt{d} \|\boldsymbol{g}\|} \boldsymbol{g}^\top \boldsymbol{\Sigma}^{-1/2} \mathtt{vec}(\mathbb{I}_p).$$

Using Lemma 14, it follows that

$$\lim_{d \to \infty} \frac{1}{p^{3/2}} \boldsymbol{g}^\top \boldsymbol{\Sigma}^{-1/2} \mathtt{vec}(\mathbb{I}_p) = 0,$$

which concludes the proof for this part. $\qquad \square$

**Lemma 9** (ID Loss). *Using $\boldsymbol{\Sigma}, \boldsymbol{\Sigma}(\boldsymbol{w}_j), \boldsymbol{u}, \boldsymbol{\theta}$ of the same form as Appendix E.2, let*

$$T_1(\boldsymbol{w}_j) := \frac{1}{\|\boldsymbol{g}\|^2} \boldsymbol{g}^\top \boldsymbol{\Sigma}^{-1/2} \boldsymbol{\Sigma}(\boldsymbol{w}_j) \boldsymbol{\Sigma}^{-1/2} \boldsymbol{g},$$

$$T_2(\boldsymbol{w}_j) := \frac{1}{\sqrt{d} \|\boldsymbol{g}\|} \boldsymbol{g}^\top \boldsymbol{\Sigma}^{-1/2} \boldsymbol{\Sigma}(\boldsymbol{w}_j) \boldsymbol{\theta}, \qquad T_3(\boldsymbol{w}_j) := \frac{1}{\sqrt{d} \|\boldsymbol{g}\|} \boldsymbol{g}^\top \boldsymbol{\Sigma}^{-1/2} (\boldsymbol{w}_j \otimes \boldsymbol{w}_j).$$

*Then, almost surely*

$$\lim_{d \to \infty} \frac{1}{M} \sum_{j=1}^M T_1(\boldsymbol{w}_j) = \frac{1}{b}(1 + z_c m_\gamma(z_c)), \qquad \lim_{d \to \infty} \frac{1}{M} \sum_{j=1}^M T_2(\boldsymbol{w}_j) = 0, \qquad \lim_{d \to \infty} \frac{1}{M} \sum_{j=1}^M T_3(\boldsymbol{w}_j) = 0.$$

*Proof.* We first work with $T_1(\boldsymbol{w}_j)$. The average is

$$\frac{1}{M} \sum_{j=1}^M T_1(\boldsymbol{w}_j) = \frac{1}{M} \frac{1}{\|\boldsymbol{g}\|^2} \boldsymbol{g}^\top \boldsymbol{\Sigma}^{-1/2} \left( c' \frac{\|\boldsymbol{w}_j\|^2}{d} + b' \right) \boldsymbol{\Sigma}^{-1/2} \boldsymbol{g} + \frac{1}{\|\boldsymbol{g}\|^2} \boldsymbol{g}^\top \boldsymbol{\Sigma}^{-1/2} (\boldsymbol{W}_p \otimes \mathbb{I}_p) \boldsymbol{\Sigma}^{-1/2} \boldsymbol{g}.$$

This just follows using definition of $\boldsymbol{\Sigma}(\boldsymbol{w}_j)$. Next, in the limit $d \to \infty$, term 1 follows using Eq. (65). The limit of second quantity above

$$\begin{aligned}
\lim_{d \to \infty} \frac{1}{\|\boldsymbol{g}\|^2} \boldsymbol{g}^\top \boldsymbol{\Sigma}^{-1/2} (\boldsymbol{W}_p \otimes \mathbb{I}_p) \boldsymbol{\Sigma}^{-1/2} \boldsymbol{g} &= \lim_{d \to \infty} \frac{1}{p^2} \mathrm{tr}(\boldsymbol{\Sigma}^{-1/2} (\boldsymbol{W}_p \otimes \mathbb{I}_p) \boldsymbol{\Sigma}^{-1/2}) \\
&= \lim_{p \to \infty} \frac{1}{p^2} \mathrm{tr}(\boldsymbol{\Sigma}^{-1} \boldsymbol{W}_p \otimes \mathbb{I}_p)) \\
&= \lim_{p \to \infty} \frac{1}{p} \mathrm{tr}(\boldsymbol{A}^{-1} \boldsymbol{W}_p) \\
&= \frac{1}{b} \int \frac{\lambda}{\lambda + c/b} \mathrm{d}\mu_\gamma(\lambda) = \frac{1}{b}(1 + z_c m_\gamma(z_c)).
\end{aligned}$$

Here, we use commutation of $\boldsymbol{\Sigma}$ and $\boldsymbol{W}_p \otimes \mathbb{I}_p$ and properties Eq. (30), Eq. (32). The final equality follows as when $p, M \to \infty$, the empirical spectral distribution of $\boldsymbol{W}_p$ converges to the Marchenko-Pastur law $\mu_\gamma$.

Next, the limiting averages of $T_2(\boldsymbol{w}_j)$ and $T_3(\boldsymbol{w}_j)$, using Lemma 14, and similar steps used in Lemma 8 both evaluate to 0. $\qquad \square$

We restate Theorem 4 below followed by its proof.

**Theorem 7** (Asymptotic Risk of LS LGP). *Consider the least-squares estimator $\hat{\boldsymbol{\theta}}_{\mathcal{S}_p}$ for the linear Gaussian equivalent problem in the underparameterized regime ($\nu > \rho^2$). Under the asymptotic scaling in Eq. (1), as $d \to \infty$, the ID and OOD risks converge in probability to:*

$$\mathcal{L}_{OOD}(\hat{\boldsymbol{\theta}}_{\mathcal{S}_p}) \xrightarrow{P} m_\gamma(z_c)(\kappa_\infty^2(1+c) - 2c^2\rho) + (1+c)\rho c^2 m_\gamma'(z_c) + \sigma_n^2 + 1 - \rho + \rho c,$$

$$\mathcal{L}_{ID}(\hat{\boldsymbol{\theta}}_{\mathcal{S}_p}) \xrightarrow{P} \frac{\nu}{\rho^2}\kappa_\infty^2,$$

*where $\kappa_\infty$ is defined in Eq. (9).*

*Proof.* Substituting $\boldsymbol{a} = \frac{\kappa_\infty}{\|\boldsymbol{g}\|}\boldsymbol{g}$ and using Eq. (27) and Eq. (28) in Eq. (26), we have that

$$\mathcal{L}(\boldsymbol{a}; \boldsymbol{w}) = \frac{\kappa_\infty^2}{\|\boldsymbol{g}\|^2}\boldsymbol{g}^\top \overline{\boldsymbol{\Sigma}}_{\boldsymbol{H}_{\mathcal{S}_p}}^{-1/2}\overline{\boldsymbol{\Sigma}}(\boldsymbol{w})\overline{\boldsymbol{\Sigma}}_{\boldsymbol{H}_{\mathcal{S}_p}}^{-1/2}\boldsymbol{g} + \frac{1}{d}\overline{\boldsymbol{\theta}}_\star^\top \overline{\boldsymbol{\Sigma}}(\boldsymbol{w})\overline{\boldsymbol{\theta}}_\star + 2\frac{\kappa_\infty}{\sqrt{d}\|\boldsymbol{g}\|}\boldsymbol{g}^\top \overline{\boldsymbol{\Sigma}}_{\boldsymbol{H}_{\mathcal{S}_p}}^{-1/2}\overline{\boldsymbol{\Sigma}}(\boldsymbol{w})\overline{\boldsymbol{\theta}}_\star$$

$$- 2\left(\frac{\kappa_\infty}{d^{1/2}\|\boldsymbol{g}\|}\boldsymbol{g}^\top \overline{\boldsymbol{\Sigma}}_{\boldsymbol{H}_{\mathcal{S}_p}}^{-1/2}(\boldsymbol{w} \otimes \boldsymbol{w}) + \frac{1}{d}\overline{\boldsymbol{\theta}}_\star^\top(\boldsymbol{w} \otimes \boldsymbol{w})\right) + 1 + \sigma_n^2.$$

Using Lemma 8, Eq. (44), Eq. (60), and $b = 1, c' + b' = c$, we have that

$$\mathcal{L}_{\text{OOD}}(\hat{\boldsymbol{\theta}}_{\mathcal{S}_p}) := \lim_{d \to \infty} \mathbb{E}_{\boldsymbol{w}}[\mathcal{L}(\boldsymbol{a}; \boldsymbol{w})]$$

$$= \kappa_\infty^2(1+c)m_\gamma(z_c) + (1+c)\rho\left(1 + 2z_c m_\gamma(z_c) + z_c^2 m_\gamma'(z_c)\right) - 2\rho(1 + z_c m_\gamma(z_c)) + 1 + \sigma_n^2$$

$$= m_\gamma(z_c)(\kappa_\infty^2(1+c) - 2c^2\rho) + (1+c)\rho c^2 m_\gamma'(z_c) + \sigma_n^2 + 1 - \rho + \rho c,$$

where using Eq. (43),

$$\kappa_\infty^2 = \frac{c_\infty}{\nu\left(\frac{\nu}{\rho^2} - 1\right)} = \frac{\sigma_n^2 + 1 - \rho + \rho c - \rho c^2 m_\gamma(z_c)}{\frac{\nu}{\rho^2} - 1}.$$

Similarly, using Lemma 9, we have

$$\mathcal{L}_{\text{ID}}(\hat{\boldsymbol{\theta}}_{\mathcal{S}_p}) := \lim_{d \to \infty} \frac{1}{M}\sum_{i=1}^M \mathcal{L}(\boldsymbol{a}; \boldsymbol{w}_i)]$$

$$= \kappa_\infty^2\left(cm_\gamma(z_c) + 1 - cm_\gamma(z_c)\right) + c\rho\left(1 - 2cm_\gamma(z_c) + c^2 m_\gamma'(z_c)\right)$$

$$+ \rho(1 - 2c + 3c^2 m_\gamma(z_c) - c^3 m_\gamma'(z_c)) - 2\rho(1 - c + c^2 m_\gamma(z_c)) + 1 + \sigma_n^2$$

$$= \kappa_\infty^2 - \rho c^2 m_\gamma(z_c) + \sigma_n^2 + 1 - \rho + \rho c = \frac{\nu}{\rho^2}\kappa_\infty^2.$$

$\square$

# H HELPER LEMMAS

**Lemma 10.** *Let $z_1, z_2, z_3, \alpha_0, \alpha_1, \alpha_2 \in \mathbb{R}$ be constants and $f(\cdot)$ be a function. Using*

$$m_\gamma(z) = \int \frac{1}{\lambda - z}\,\mathrm{d}\mu_\gamma(\lambda), \qquad m_\gamma'(z) = \int \frac{1}{(\lambda - z)^2}\,\mathrm{d}\mu_\gamma(\lambda),$$

*(i)* $\mathcal{I}_1 := \int \frac{\lambda - z_1}{\lambda - z_2}\,\mathrm{d}\mu_\gamma(\lambda) = 1 + (z_2 - z_1)m_\gamma(z_2),$

*(ii)* $\mathcal{I}_2 := \int \frac{\lambda - z_1}{(\lambda - z_2)^2}\,\mathrm{d}\mu_\gamma(\lambda) = m_\gamma(z_2) + (z_2 - z_1)m_\gamma'(z_2),$

*(iii)* $\mathcal{I}_3 := \int \frac{\lambda^2}{(\lambda - z_1)^2(\lambda - z_2)}\,\mathrm{d}\mu_\gamma(\lambda) = \frac{1}{(z_1 - z_2)^2}(z_1(z_1 - 2z_2)m_\gamma(z_1) + z_2^2 m_\gamma(z_2) + z_1^2(z_1 - z_2)m_\gamma'(z_1)),$

(iv) $\mathcal{I}_4 := \int \frac{\lambda}{(\lambda - z_1)(\lambda - z_2)} \, \mathrm{d}\mu_\gamma(\lambda) = \frac{1}{z_1 - z_2}(z_1 m_\gamma(z_1) - z_2 m_\gamma(z_2))$,

(v) if $\int f(\lambda) \, \mathrm{d}\mu_\gamma(\lambda) = \alpha_0 + \alpha_1 m_\gamma(z_1) + \alpha_2 m_\gamma'(z_1)$, then

$$\int \lambda f(\lambda) \, \mathrm{d}\mu_\gamma(\lambda) = \alpha_0 + \alpha_1 + (\alpha_1 z_1 + \alpha_2) m_\gamma(z_1) + \alpha_2 z_1 m_\gamma'(z_1),$$

(vi) $\mathcal{I}_6 := \int \frac{\lambda^2}{(\lambda - z_1)^2} \, \mathrm{d}\mu_\gamma(\lambda) = 1 + 2z_1 m_\gamma(z_1) + z_1^2 m_\gamma'(z_1)$,

(vii) $\mathcal{I}_7 := \int \frac{\lambda^2}{(\lambda - z_1)^2 (\lambda - z_2)^2} \, \mathrm{d}\mu_\gamma(\lambda) = \frac{1}{(z_1 - z_2)^2} \left( z_1^2 m_\gamma'(z_1) + z_2^2 m_\gamma'(z_2) - \frac{2z_1 z_2}{z_1 - z_2}(m_\gamma(z_1) - m_\gamma(z_2)) \right)$.

*Proof.* The proof relies on obtaining partial fraction decompositions and using the definitions of $m_\gamma(z), m_\gamma'(z)$.

First, for $\mathcal{I}_1$, note that

$$\frac{\lambda - z_1}{\lambda - z_2} = 1 + \frac{z_2 - z_1}{\lambda - z_2}, \quad \mathcal{I}_1 = 1 + (z_2 - z_1) m_\gamma(z_2).$$

Next, for $\mathcal{I}_2$, we have

$$\frac{\lambda - z_1}{(\lambda - z_2)^2} = \frac{1}{\lambda - z_2} + \frac{z_2 - z_1}{(\lambda - z_2)^2} \implies \mathcal{I}_2 = m_\gamma(z_2) + (z_2 - z_1) m_\gamma'(z_2).$$

Next, for $\mathcal{I}_3$, we have

$$\frac{\lambda^2}{(\lambda - z_1)^2 (\lambda - z_2)} = \frac{z_1^2}{z_1 - z_2} \frac{1}{(\lambda - z_1)^2} + \frac{z_1(z_1 - 2z_2)}{(z_1 - z_2)^2} \frac{1}{\lambda - z_1} + \frac{z_2^2}{(z_1 - z_2)^2} \frac{1}{\lambda - z_2},$$

which gives $\mathcal{I}_3 = \frac{1}{(z_1 - z_2)^2}(z_1(z_1 - 2z_2) m_\gamma(z_1) + z_2^2 m_\gamma(z_2) + z_1^2(z_1 - z_2) m_\gamma'(z_1)$.

Next, for $\mathcal{I}_4$, we have

$$\frac{\lambda}{(\lambda - z_1)(\lambda - z_2)} = \frac{z_1}{z_1 - z_2} \frac{1}{\lambda - z_1} + \frac{z_2}{z_2 - z_1} \frac{1}{\lambda - z_2},$$

which gives $\mathcal{I}_4 = \frac{z_1}{z_1 - z_2} m_\gamma(z_1) + \frac{z_2}{z_2 - z_1} m_\gamma(z_2)$.

For the next part, we have

$$\alpha_1 m_\gamma(z_1) + \alpha_2 m_\gamma'(z_1) = \alpha_0 + \alpha_1 \int \frac{1}{\lambda - z_1} \, \mathrm{d}\mu_\gamma(\lambda) + \alpha_2 \int \frac{1}{(\lambda - z_1)^2} \, \mathrm{d}\mu_\gamma(\lambda)$$

This gives

$$\int \lambda f(\lambda) \, \mathrm{d}\mu_\gamma(\lambda) = \alpha_0 \int \lambda \, \mathrm{d}\mu_\gamma(\lambda) + \alpha_1 \int \frac{\lambda}{\lambda - z_1} \, \mathrm{d}\mu_\gamma(\lambda) + \alpha_2 \int \frac{\lambda}{(\lambda - z_1)^2} \, \mathrm{d}\mu_\gamma(\lambda)$$
$$= \alpha_0 + \alpha_1(1 + z_1 m_\gamma(z_1)) + \alpha_2(m_\gamma(z_1) + z_1 m_\gamma(z_1)),$$

where we used $\mathcal{I}_1$ for the first term and $\mathcal{I}_2$ for second term. Simplifying this finishes the proof for this part.

Next, for $\mathcal{I}_6$, we have

$$\frac{\lambda^2}{(\lambda - z_1)^2} = \left( 1 + \frac{z_1}{\lambda - z_1} \right)^2 = 1 + \frac{2z_1}{\lambda - z_1} + \frac{z_1^2}{(\lambda - z_1)^2},$$

which gives $\mathcal{I}_6 = 1 + 2z_1 m_\gamma(z_1) + z_1^2 m_\gamma'(z_1)$.

Next, for $\mathcal{I}_7$, we have

$$\frac{\lambda^2}{(\lambda - z_1)^2 (\lambda - z_2)^2} = \frac{1}{(z_1 - z_2)^2} \left( \frac{z_1}{\lambda - z_1} - \frac{z_2}{\lambda - z_2} \right)^2$$
$$= \frac{1}{(z_1 - z_2)^2} \left( \frac{z_1^2}{(\lambda - z_1)^2} + \frac{z_2^2}{(\lambda - z_2)^2} - \frac{2z_1 z_2}{z_1 - z_2} \left( \frac{1}{\lambda - z_1} - \frac{1}{\lambda - z_2} \right) \right),$$

which gives $\mathcal{I}_7 = \frac{1}{(z_1 - z_2)^2} \left( z_1^2 m_\gamma'(z_1) + z_2^2 m_\gamma'(z_2) - \frac{2z_1 z_2}{z_1 - z_2}(m_\gamma(z_1) - m_\gamma(z_2)) \right)$. $\qquad \square$

**Lemma 11.** $\mathbb{E}_{\boldsymbol{w}}[\Sigma(\boldsymbol{w})] = (c' + b' + 1)\mathbb{I}_{d^2}$.

*Proof.* Since $\boldsymbol{w} \sim \mathcal{N}(0, I_d)$, we have $\mathbb{E}[\boldsymbol{w}\boldsymbol{w}^\top] = \mathbb{I}_d$ and $\mathbb{E}[\|\boldsymbol{w}\|^2/d] = 1$, hence

$$\mathbb{E}_{\boldsymbol{w}}[\Sigma(\boldsymbol{w})] = \Big(c'\mathbb{E}[\tfrac{\|\boldsymbol{w}\|^2}{d}] + b'\Big)\mathbb{I}_{d^2} + \mathbb{E}[(\boldsymbol{w}\boldsymbol{w}^\top) \otimes \mathbb{I}_d] = (c' + b' + 1)\mathbb{I}_{d^2}.$$

$\square$

**Lemma 12** (Gaussian tail bound). *If $X \sim \mathcal{N}(0, \sigma^2)$, then for all $t > 0$,*

$$\Pr(|X| \geq t) \leq 2\exp\Big(-\frac{t^2}{2\sigma^2}\Big).$$

**Lemma 13** (Sub-exponential tail bound). *If $X_i$ are i.i.d., zero mean, sub-exponential random variables, then there exist universal constants $K, c_0 > 0$ such that $\|X_i\|_{\psi_1} \leq K$ and*

$$\Pr\Big(\Big|\sum_{i=1}^{d} a_i X_i\Big| \geq t\Big) \leq 2\exp\Big(-c_0\min\Big\{\frac{t^2}{K^2\|a\|_2^2}, \frac{t}{K\|a\|_\infty}\Big\}\Big) \quad \text{for all } t > 0.$$

**Lemma 14.** *Let $\boldsymbol{a}_1 \in \mathbb{R}^{d_1}$ and $\boldsymbol{A}_1 \in \mathbb{R}^{d_1 \times d_1}$, s.t. $\|\boldsymbol{a}_1\| = O(1/d_1^q)$, $q > 0$ and eigenvalues of $\boldsymbol{A}_1$ are $\Theta(1)$. Then,*

$$\lim_{d_1 \to \infty} \boldsymbol{g}^\top \boldsymbol{a}_1 = 0$$

$$\lim_{d_1 \to \infty} \frac{1}{d_1}\boldsymbol{g}^\top \boldsymbol{A}\boldsymbol{g} = \lim_{d_1 \to \infty} \frac{1}{d_1}\operatorname{tr}(\boldsymbol{A}_1).$$

*Proof.* For the first, we know that if $\boldsymbol{g} \sim \mathcal{N}(\boldsymbol{0}, \mathbb{I}_{d_1})$, then $\boldsymbol{g}^\top \boldsymbol{a}_1 \sim \mathcal{N}(0, \|\boldsymbol{a}_1\|^2)$. Using standard Gaussian tail bounds from Lemma 12 with $\|\boldsymbol{a}_1\| = O(1/d_1^q)$, we have $\lim_{d_1 \to \infty} \boldsymbol{g}^\top \boldsymbol{a}_1 = 0$.

Next, let $\boldsymbol{A}_1 = \boldsymbol{V}\Lambda\boldsymbol{V}^\top$ be the Eigenvalue decomposition of $\boldsymbol{A}_1$. We have

$$\boldsymbol{g}^\top \boldsymbol{A}_1 \boldsymbol{g} = (\boldsymbol{V}^\top \boldsymbol{g})^\top \Lambda(\boldsymbol{V}^\top \boldsymbol{g}) = \frac{1}{d_1}\sum_{i=1}^{d} \lambda_i \tilde{g_i}^2,$$

where $\boldsymbol{V}^\top \boldsymbol{g} =: \tilde{\boldsymbol{g}} \sim \mathcal{N}(\boldsymbol{0}, \mathbb{I}_{d_1})$. We know that $X_i := \tilde{g_i}^2 - 1$ are 0 mean i.i.d sub-exponential random variables for all $i \in [d_1]$. Applying sub-exponential tail bound from Lemma 13 to $\sum_{i=1}^{d} \lambda_i X_i$, we have

$$\Pr\Big(\Big|\frac{1}{d_1}\sum_{i=1}^{d_1} \lambda_i X_i\Big| \geq t\Big) = 2\exp\Big(-c_0 d_1\min\Big\{\frac{d_1 t^2}{\|\Lambda\|_F^2}, \frac{t}{\lambda_{\max}}\Big\}\Big) \quad \text{for all } t > 0.$$

$$\leq 2\exp\Big(-c_0 d_1\min\Big\{\frac{t^2}{C^2}, \frac{t}{C}\Big\}\Big),$$

where inequality uses $\lambda_i = \Theta(1)$. Therefore

$$\lim_{d_1 \to \infty} \frac{1}{d_1}\sum_{i=1}^{d_1} \lambda_i X_i = \lim_{d_1 \to \infty} \frac{1}{d_1}(\boldsymbol{g}^\top \boldsymbol{A}_1 \boldsymbol{g} - \operatorname{tr}(\boldsymbol{A}_1)) = 0,$$

where we use that $\sum_{i=1}^{d_1} \lambda_i = \operatorname{tr}(\Lambda) = \operatorname{tr}(\boldsymbol{A}_1)$. $\square$

**Lemma 15.** *Let $\boldsymbol{x}_t \in \mathbb{R}^d$ have i.i.d. entries $\boldsymbol{x}_t[i] \sim \mathcal{N}(0, \sigma_x^2)$, and let $\boldsymbol{w} \in \mathbb{R}^d$ be fixed. For a subset $\mathcal{S} \subseteq [d]$ with $|\mathcal{S}| = p$, denote by $\boldsymbol{x}_t[\mathcal{S}] \in \mathbb{R}^p$ and $\boldsymbol{w}[\mathcal{S}] \in \mathbb{R}^p$ the corresponding coordinate sub-vectors. Then*

$$\mathbb{E}\big[(\boldsymbol{w}^\top \boldsymbol{x}_t)^2 \boldsymbol{x}_t[\mathcal{S}]\boldsymbol{x}_t[\mathcal{S}]^\top\big] = \sigma_x^4\|\boldsymbol{w}\|^2\,\mathbb{I}_p + 2\sigma_x^4\,\boldsymbol{w}[\mathcal{S}]\boldsymbol{w}[\mathcal{S}]^\top.$$

*Proof.* Write $\boldsymbol{x}_t = \sigma_x \boldsymbol{g}$ where $\boldsymbol{g} \sim \mathcal{N}(0, I_d)$ has independent standard normal entries. Then

$$(\boldsymbol{w}^\top \boldsymbol{x}_t)^2 \, \boldsymbol{x}_t[i]\boldsymbol{x}_t[j] = \sigma_x^4 (\boldsymbol{w}^\top \boldsymbol{g})^2 g_i g_j = \sigma_x^4 \Big( \sum_{k=1}^d w_k g_k \Big)^2 g_i g_j.$$

Expanding the square and taking expectations,

$$\mathbb{E}\big[(\boldsymbol{w}^\top \boldsymbol{x}_t)^2 \, \boldsymbol{x}_t[i]\boldsymbol{x}_t[j]\big] = \sigma_x^4 \sum_{k,\ell=1}^d w_k w_\ell \, \mathbb{E}[g_k g_\ell g_i g_j].$$

By Isserlis' (Wick's) theorem for a zero-mean Gaussian vector,

$$\mathbb{E}[g_k g_\ell g_i g_j] = \delta_{k\ell}\delta_{ij} + \delta_{ki}\delta_{\ell j} + \delta_{kj}\delta_{\ell i},$$

where $\delta_{ab}$ is the Kronecker delta. Hence

$$\mathbb{E}\big[(\boldsymbol{w}^\top \boldsymbol{x}_t)^2 \, \boldsymbol{x}_t[i]\boldsymbol{x}_t[j]\big] = \sigma_x^4 \sum_{k,\ell=1}^d w_k w_\ell \big( \delta_{k\ell}\delta_{ij} + \delta_{ki}\delta_{\ell j} + \delta_{kj}\delta_{\ell i} \big)$$

$$= \sigma_x^4 \Big( \delta_{ij} \sum_{k=1}^d w_k^2 + w_i w_j + w_j w_i \Big)$$

$$= \sigma_x^4 \Big( \delta_{ij} \|\boldsymbol{w}\|^2 + 2 w_i w_j \Big).$$

Restricting to indices $i, j \in \mathcal{S}$ and collecting these entries into a $p \times p$ matrix yields

$$\mathbb{E}\big[(\boldsymbol{w}^\top \boldsymbol{x}_t)^2 \, \boldsymbol{x}_t[\mathcal{S}]\boldsymbol{x}_t[\mathcal{S}]^\top\big] = \sigma_x^4 \|\boldsymbol{w}\|^2 \mathbb{I}_p + 2\sigma_x^4 \boldsymbol{w}[\mathcal{S}]\boldsymbol{w}[\mathcal{S}]^\top,$$

as claimed. $\qquad\square$

**Lemma 16.** *Under the setup described in Section 2, where $\mathcal{S} \subseteq [d]$ be any subset of indices of size $p$, $\boldsymbol{w} \in \mathbb{R}^d$ be a task vector, $\boldsymbol{x}_1, \ldots, \boldsymbol{x}_T, \boldsymbol{x}_q \in \mathbb{R}^d$ be i.i.d. samples from $\mathcal{N}(\mathbf{0}, \sigma_x^2 \mathbb{I}_d)$, labels $y_t = \boldsymbol{w}^\top \boldsymbol{x}_t + \epsilon_t$, where $\epsilon_t \sim \mathcal{N}(0, \sigma_n^2)$, and*

$$\widehat{\boldsymbol{w}}_{\mathrm{avg}}[\mathcal{S}] := \frac{1}{T} \sum_{t=1}^T y_t \, \boldsymbol{x}_t[\mathcal{S}] \in \mathbb{R}^p, \quad \mathbf{h}_{\mathcal{S}_p}(X) := \mathrm{vec}\big( \boldsymbol{x}_q[\mathcal{S}] \, \widehat{\boldsymbol{w}}_{\mathrm{avg}}[\mathcal{S}]^\top \big).$$

*Then the following hold:*

$$\underbrace{\mathbb{E}_{X,\epsilon}[\mathbf{h}_{\mathcal{S}_p}(X)\mathbf{h}_{\mathcal{S}_p}(X)^\top]}_{\textit{Term-I}} = \frac{\sigma_x^4}{T} \left[ (\sigma_x^2 \|\boldsymbol{w}\|^2 + \sigma_n^2)\mathbb{I}_{p^2} + (T+1)\sigma_x^2 (\boldsymbol{w}[\mathcal{S}]\boldsymbol{w}[\mathcal{S}]^\top) \otimes \mathbb{I}_p \right],$$

$$\underbrace{\mathbb{E}_{X,y}[\mathbf{h}_{\mathcal{S}_p}(X)y_q]}_{\textit{Term-II}} = \sigma_x^4 \boldsymbol{w}[\mathcal{S}] \otimes \boldsymbol{w}[\mathcal{S}].$$

*Proof.* We first simplify Term-I. We have that

$$\mathbb{E}_{X,\epsilon}[\mathbf{h}_{\mathcal{S}_p}(X)\mathbf{h}_{\mathcal{S}_p}(X)^\top] = \mathbb{E}_{X,\epsilon} \left[ (\mathrm{vec}(\boldsymbol{x}_q[\mathcal{S}]\hat{\boldsymbol{w}}_{\mathrm{avg}}[\mathcal{S}]^\top))(\mathrm{vec}(\boldsymbol{x}_q[\mathcal{S}]\hat{\boldsymbol{w}}_{\mathrm{avg}}[\mathcal{S}]^\top))^\top \right].$$

Using Eq. (33), and since $\boldsymbol{x}_q$ is independent of $\hat{\boldsymbol{w}}_{\mathrm{avg}} = \frac{1}{T} \sum_t y_t \boldsymbol{x}_t$, separating the expectation, we get

$$\mathbb{E}_{X,\epsilon}[\mathbf{h}_{\mathcal{S}_p}(X)\mathbf{h}_{\mathcal{S}_p}(X)^\top] = \frac{1}{T^2} \underbrace{\mathbb{E}\left[ \left( \sum_{t=1}^T y_t \boldsymbol{x}_t[\mathcal{S}] \right) \left( \sum_{t'=1}^T y_{t'} \boldsymbol{x}_{t'}[\mathcal{S}] \right)^\top \right]}_{\textit{Term-I(i)}} \otimes \underbrace{\mathbb{E}[\boldsymbol{x}_q[\mathcal{S}]\boldsymbol{x}_q[\mathcal{S}]^\top]}_{\textit{Term-1(ii)}}$$

For Term-I(ii), since $\boldsymbol{x}_q[S] \sim \mathcal{N}(\mathbf{0}, \sigma_x^2 \mathbb{I}_p)$, we have $\mathbb{E}[\boldsymbol{x}_q[\mathcal{S}]\boldsymbol{x}_q[\mathcal{S}]^\top] = \sigma_x^2 \mathbb{I}_p$. Next, we simplify Term-I(i).

$$\mathbb{E}\left[ \left( \sum_{t=1}^T y_t \boldsymbol{x}_t[\mathcal{S}] \right) \left( \sum_{t'=1}^T y_{t'} \boldsymbol{x}_{t'}[\mathcal{S}] \right)^\top \right] = \sum_{t=1}^T \sum_{t'=1}^T \mathbb{E}[y_t y_{t'} \boldsymbol{x}_t[\mathcal{S}]\boldsymbol{x}_{t'}[\mathcal{S}]^\top]$$

$$= \sum_{t=1}^T \mathbb{E}[y_t^2 \boldsymbol{x}_t[\mathcal{S}] \, \boldsymbol{x}_t[\mathcal{S}]^\top] + \sum_{t \neq t'} \mathbb{E}[y_t y_{t'} \boldsymbol{x}_t[\mathcal{S}] \, \boldsymbol{x}_{t'}[\mathcal{S}]^\top]$$

First, looking at $t \neq t'$ terms, we have:

$$\mathbb{E}[y_t y_{t'} \boldsymbol{x}_t[\mathcal{S}] \, \boldsymbol{x}_{t'}[\mathcal{S}]^\top] = \mathbb{E}[(\boldsymbol{w}^\top \boldsymbol{x}_t + \epsilon_t) \, (\boldsymbol{w}^\top \boldsymbol{x}_{t'} + \epsilon_{t'}) \, \boldsymbol{x}_t[\mathcal{S}] \, \boldsymbol{x}_{t'}[\mathcal{S}]^\top]$$
$$= \mathbb{E}[(\boldsymbol{w}^\top \boldsymbol{x}_t) \, \boldsymbol{x}_t[\mathcal{S}]] \, \mathbb{E}[(\boldsymbol{w}^\top \boldsymbol{x}_{t'}) \, \boldsymbol{x}_{t'}[\mathcal{S}]^\top]$$
$$= \sigma_x^4 \Pi_{p \times d} \boldsymbol{w} \, \boldsymbol{w}^\top \Pi_{p \times d}^\top$$
$$= \sigma_x^4 \boldsymbol{w}[\mathcal{S}] \boldsymbol{w}[\mathcal{S}]^\top,$$

where the second equality follows by independence and $\mathbb{E}[\epsilon] = 0$, and for the third equality, we use the definition of $\Pi_{p \times d}$, and that $\boldsymbol{x} \sim \mathcal{N}(\boldsymbol{0}, \sigma_x^2 \mathbb{I})$.

Next, consider the terms with $t = t'$:

$$\mathbb{E}[y_t^2 \boldsymbol{x}_t \boldsymbol{x}_t^\top] = \mathbb{E}[(\boldsymbol{w}^\top \boldsymbol{x}_t + \epsilon_t)^2 \boldsymbol{x}_t[\mathcal{S}] \, \boldsymbol{x}_t[\mathcal{S}]^\top]$$
$$= \mathbb{E}[(\boldsymbol{w}^\top \boldsymbol{x}_t)^2 \, \boldsymbol{x}_t[\mathcal{S}] \, \boldsymbol{x}_t[\mathcal{S}]^\top] + \mathbb{E}[\epsilon_t^2 \, \boldsymbol{x}_t[\mathcal{S}] \, \boldsymbol{x}_t[\mathcal{S}]^\top]$$
$$= \sigma_x^4 \|\boldsymbol{w}\|^2 \, \mathbb{I}_p + 2\sigma_x^4 \boldsymbol{w}[\mathcal{S}] \boldsymbol{w}[\mathcal{S}]^\top + \sigma_n^2 \sigma_x^2 \mathbb{I}_p$$
$$= (\sigma_x^4 \|\boldsymbol{w}\|^2 + \sigma_x^2 \sigma_n^2)\mathbb{I}_p + 2\sigma_x^4 \boldsymbol{w}[\mathcal{S}] \boldsymbol{w}[\mathcal{S}]^\top,$$

where the third equality follows by using Lemma 15 and that $\boldsymbol{x} \sim \mathcal{N}(\boldsymbol{0}, \sigma_x^2 \mathbb{I}_d)$.

Combining the $T$ diagonal terms and $T(T-1)$ off-diagonal terms, we have

$$\mathbb{E}\left[\left(\sum_{t=1}^T y_t \boldsymbol{x}_t[\mathcal{S}]\right)\left(\sum_{t'=1}^T y_{t'} \boldsymbol{x}_{t'}[\mathcal{S}]\right)^\top\right] = T\left[(\sigma_x^4 \|\boldsymbol{w}\|^2 + \sigma_x^2 \sigma_n^2)\mathbb{I}_p + 2\sigma_x^4 \boldsymbol{w}[\mathcal{S}] \boldsymbol{w}[\mathcal{S}]^\top\right] + T(T-1)\sigma_x^4 \boldsymbol{w}[\mathcal{S}] \boldsymbol{w}[\mathcal{S}]^\top$$
$$= T(\sigma_x^4 \|\boldsymbol{w}\|^2 + \sigma_x^2 \sigma_n^2)\mathbb{I}_p + (2T + T^2 - T)\sigma_x^4 \boldsymbol{w}[\mathcal{S}] \boldsymbol{w}[\mathcal{S}]^\top$$
$$= T(\sigma_x^4 \|\boldsymbol{w}\|^2 + \sigma_x^2 \sigma_n^2)\mathbb{I}_p + (T^2 + T)\sigma_x^4 \boldsymbol{w}[\mathcal{S}] \boldsymbol{w}[\mathcal{S}]^\top.$$

Combining this resultant Term-1(i) with Term-1(ii), we get Term-1:

$$\mathbb{E}_{X,\epsilon}[\mathbf{h}_{\mathcal{S}_p}(X)\mathbf{h}_{\mathcal{S}_p}(X)^\top] = \frac{1}{T^2}\left[T(\sigma_x^4 \|\boldsymbol{w}\|^2 + \sigma_x^2 \sigma_n^2)\mathbb{I}_p + (T^2 + T)\sigma_x^4 \boldsymbol{w}[\mathcal{S}] \boldsymbol{w}[\mathcal{S}]^\top\right] \otimes \sigma_x^2 \mathbb{I}_p$$
$$= \frac{\sigma_x^4}{T}\left[(\sigma_x^2 \|\boldsymbol{w}\|^2 + \sigma_n^2)\mathbb{I}_{p^2} + (T+1)\sigma_x^2(\boldsymbol{w}[\mathcal{S}] \boldsymbol{w}[\mathcal{S}]^\top) \otimes \mathbb{I}_p\right].$$

Next, let us simplify Term-II:

$$\mathbb{E}_{X,y}[\mathbf{h}_{\mathcal{S}_p}(X)y_q] = \mathbb{E}_{X,\epsilon}\left[\text{vec}(\boldsymbol{x}_q[\mathcal{S}]\hat{\boldsymbol{w}}_{\text{avg}}[\mathcal{S}]^\top)(\boldsymbol{w}^\top \boldsymbol{x}_q + \epsilon)\right]$$
$$= \frac{1}{T}\mathbb{E}\underbrace{\left[\sum_t y_t \boldsymbol{x}_t[\mathcal{S}]\right]}_{\text{Term-II(i)}} \otimes \mathbb{E}\underbrace{\left[\boldsymbol{x}_q[\mathcal{S}]\boldsymbol{x}_q^\top\right]\boldsymbol{w}}_{\text{Term-II(ii)}}.$$

First, let us simplify Term-II(i). We have

$$\mathbb{E}\left[\sum_t y_t \boldsymbol{x}_t[\mathcal{S}]\right] = \mathbb{E}\left[\sum_t (\boldsymbol{w}^\top \boldsymbol{x}_t + \epsilon)\boldsymbol{x}_t[\mathcal{S}]\right]$$
$$= \sum_t \mathbb{E}\left[\boldsymbol{x}_t[\mathcal{S}]\boldsymbol{x}_t^\top\right]\boldsymbol{w} = T\sigma_x^2 \Pi_{p \times d}\boldsymbol{w} = T\sigma_x^2 \boldsymbol{w}[\mathcal{S}].$$

Similarly, Term-II(ii) is $\sigma_x^2 \boldsymbol{w}[\mathcal{S}]$. Combining these two, we get Term-II:

$$\mathbb{E}_{X,y}[\mathbf{h}_{\mathcal{S}_p}(X)y_q] = \sigma_x^4 \boldsymbol{w}[\mathcal{S}] \otimes \boldsymbol{w}[\mathcal{S}].$$

□

# I    ADDITIONAL RESULTS AND DETAILS OF EXPERIMENTAL SETTINGS

**Comparision with Lu et al. (2025).** Next, in Fig. 3, we isolate the setting studied by Lu et al. (2025) by fixing $\rho = 1$ (i.e., using all features) and vary the overparameterization ratio via $1/\nu$. We visualize the theoretical and empirical ID (left) and OOD (right) loss values. Crucially, in this setting, we do not observe in-context benign overfitting. While the OOD loss decreases as expected, the ID loss monotonic increases with overparameterization. This divergence stems from the nature of the scaling: varying $1/\nu$ requires altering the ambient dimension $d$ relative to $N$, which fundamentally changes the underlying data distribution. In contrast, our feature-selection model (varying $\rho < 1$) allows us to scale the model capacity $p$ while keeping the underlying data distribution fixed—a proxy that we argue more faithfully captures the effect of scaling model size. Consistent with our perspective, this parallels observations in supervised learning: standard linear regression without feature selection does not exhibit benign overfitting (Hastie et al., 2019).

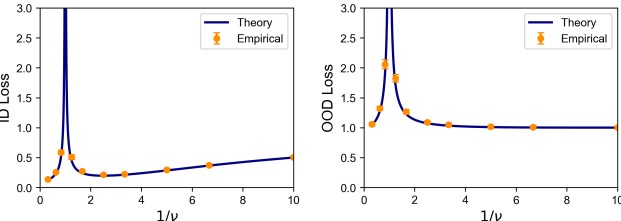

Figure 3: **Absence of In-Context Benign Overfitting.** Theoretical (lines) vs. empirical (points) ID (left) and OOD (right) risk as a function of overparameterization $1/\nu$ with fixed $\rho = 1$ (recovering the setting of Lu et al. (2025)). In contrast to Fig. 2, this setting exhibits a retrieval-learning tradeoff: as the model becomes more overparameterized, OOD generalization improves, but ID performance degrades (higher loss). This underscores that varying ambient dimension $d$ is distinct from scaling model capacity $p$ against a fixed data distribution.

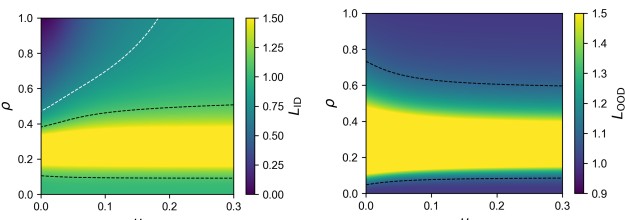

Figure 4: **Task Diversity vs. Model Scale**. Theoretical heatmaps for ID Loss (left) and OOD Loss (right) as a function of task diversity $\mu$ and model scale $\rho$. Lighter colors indicate higher loss. Vertical cuts (fixed $\mu$) demonstrate in-context benign overfitting: larger models ($\rho \approx 1$) minimize ID loss without sacrificing OOD performance. Horizontal cuts (fixed $\rho$) reveal the task diversity threshold: increasing diversity forces a transition from memorization (low ID, high OOD) loss to learning-based ICL (high ID, low OOD risk).

**Impact of Task Diversity and Scale**. Next, we investigate the role of varying the number of training tasks $M$, which we call task diversity $\mu$ and its interplay with model scale $\rho$. Figure 4 visualizes our theoretically derived ID (left) and OOD (right) risk landscapes as a joint function of model scale ($\rho$) and task diversity ($\mu$). Our analysis recovers and clarifies the task-diversity thresholds identified in prior work (Raventos et al., 2023; Park et al., 2025):

- **Fixed Task Diversity** ($\mu$): Consistent with Fig. 2, for any fixed number of pre-training tasks, increasing model scale ($\rho$) induces double descent in both ID and OOD risks. Consequently, sufficiently large models ($\rho \approx 1$) achieve lower ID loss than their underparameterized counterparts, and similar OOD loss confirming that in-context benign overfitting persists across different diversity regimes.
- **Fixed scale** ($\rho$): In contrast, for a fixed overparameterized model ($\rho > \sqrt{\nu}$), increasing task diversity ($\mu$) presents a trade-off. Higher diversity improves OOD generalization but degrades ID performance, as memorizing more tasks becomes increasingly difficult for a fixed-capacity model.

The critical point where ID performance begins to degrade while OOD performance improves corresponds to the task diversity threshold established in previous literature (Raventos et al., 2023; Park et al., 2025).

**Power Law Covariance.** Thus far, we have assumed isotropic covariance matrices for $\Sigma_x$ and $\Sigma_w$, which suffices to establish in-context benign overfitting. For completeness, in Fig. 5, we replace the isotropic covariance matrices $\Sigma_x$ and $\Sigma_w$ with power-law decay (diagonal entries decaying with exponent $\alpha = 1$; see Appendix I for details). In this setting, we observe a stronger form of in-context benign overfitting: as scale increases, both ID and OOD losses converge to values strictly lower than their respective minima in the underparameterized regime.

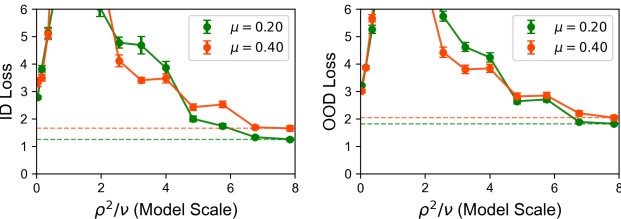

Figure 5: **Power Law Covariance (Random Selection)**. Experimental (Left) ID and (Right) OOD loss curves where covariance eigenvalues ($\Sigma_w, \Sigma_x$) follow a power-law decay ($\alpha = 1$). Using random feature selection, the overparameterized regime achieves strictly lower risk than the underparameterized minimum for both ID and OOD tasks.

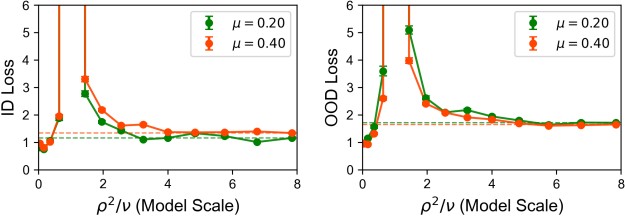

Figure 6: **Power Law Covariance (Top-$p$ Features)**. Experimental (Left) ID and (Right) OOD loss curves using top-$p$ feature selection. In contrast to random features, we do not observe in-context benign overfitting here; the overparameterized performance is strictly worse than the underparameterized minimum for both ID and OOD risks.

In Fig. 6, instead of random $p$ features, we consider top $p$ features. In this case, consistent with observations in the supervised learning setting (Belkin et al., 2020), we do not observe in-context benign overfitting: As the model scale is increased, both ID and OOD loss values converge to a higher value than in the underparameterized regime.

**Settings.** For Fig. 2, we set $d = 80, T = 120, M = 10, n = 400$, and sweep over $p = 2, 6, \ldots, 62$. For Fig. 3, we set $d = p = 80, T = 120, M = 10$, and sweep over $n = 640, 960, 1280, \ldots, 20480$. For the experiments in Figs. 5 and 6 and those in this section, we set $d = 30, T = 40, M = 6, 12, n = 100$, and sweep over $p = 2, 4, \ldots, 28$.

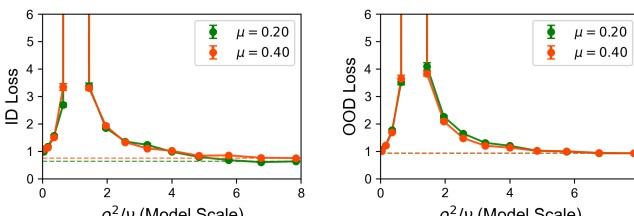

Figure 7: **Power Law Covariance (Only $\Sigma_w$)**. Experimental (Left) ID and (Right) OOD loss curves where task vector covariance eigenvalues follow a power-law decay ($\alpha = 1$). Using random feature selection, the overparameterized regime achieves strictly lower risk than the underparameterized minimum for ID tasks but not for OOD tasks.

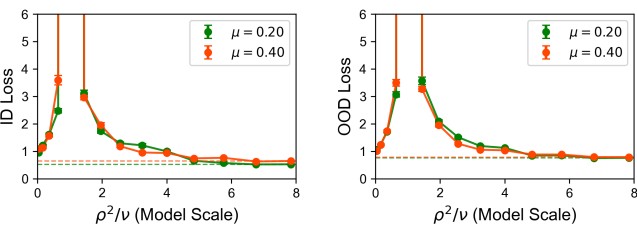

Figure 8: **Power Law Covariance (Only $\Sigma_w$)**. Experimental (Left) ID and (Right) OOD loss curves where task vector covariance eigenvalues follow a power-law decay ($\alpha = 1.5$). Using random feature selection, the overparameterized regime achieves strictly lower risk than the underparameterized minimum for both ID and OOD tasks.

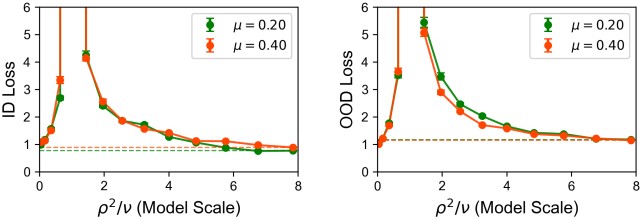

Figure 9: **Power Law Covariance (Only $\Sigma_x$)**. Experimental (Left) ID and (Right) OOD loss curves where feature covariance eigenvalues follow a power-law decay ($\alpha = 1$). Using random feature selection, the overparameterized regime achieves strictly lower risk than the underparameterized minimum for ID tasks but not for OOD tasks.

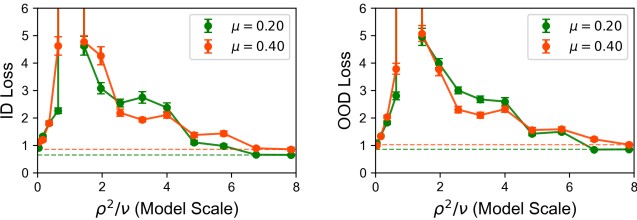

Figure 10: **Power Law Covariance (Only $\Sigma_x$)**. Experimental (Left) ID and (Right) OOD loss curves where feature covariance eigenvalues follow a power-law decay ($\alpha = 2.5$). Using random feature selection, the overparameterized regime achieves strictly lower risk than the underparameterized minimum for ID tasks and for lower $\mu$ for OOD tasks.

