# OpenReview forum: "In-Context Benign Overfitting: A Feature-Selection Model in In-Context Linear Regression"
_ICLR.cc/2026/Workshop/Sci4DL — Sci4DL 2026_

### Official Review · Reviewer_jWo5 · 2026-02-22

**Fit:** 3
**Significance:** 2
**Confidence:** 3

**Summary:**

The authors investigate in-context benign overfitting, a phenomenon similar to benign overfitting in supervised learning. The in-context version of benign overfitting shows how models can overfit to tasks in the training task distribution, while still generalizing to novel tasks not seen during training. This observation expands the analysis from earlier work, such as [1], showing how at least in certain settings achieving essentially perfect loss on the training tasks does not imply poor generalization on held-out tasks.

[1] Asymptotic theory of in-context learning by linear attention. Yue M. Lu, Mary I. Letey, Jacob A. Zavatone-Veth, Anindita Maiti, Cengiz Pehlevan, https://arxiv.org/abs/2405.11751

**Strengths:**

The authors address an important question that has far-reaching implications, and usefully extend the analysis from [1] to show how in different regimes, overfitting to the training task distribution may or may not be harmful to downstream performance. I appreciated the authors' clear description of the methods used, and the explicit connections and extensions to prior work.

Overall, I believe the paper is in a good spot for a workshop, and would encourage the authors to continue their work along this direction.

[1] Asymptotic theory of in-context learning by linear attention. Yue M. Lu, Mary I. Letey, Jacob A. Zavatone-Veth, Anindita Maiti, Cengiz Pehlevan, https://arxiv.org/abs/2405.11751

**Suggestions:**

I have a small issue with the authors' description of OOD generalization. Although the train and test task distributions used technically differ, the train task distribution is just a sample (delta distribution) from the test task distribution, and so there is no real notion of a distribution shift here, in contrast to earlier work investigating such distribution shifts, including [1,2,3]. In particular, there is no reason for the authors' results to hold in the presence of such a distribution shift, and so I would encourage the authors to clarify this point, and perhaps rethink the language used to refer to the difference between the train/test task distribution.

In fact, the work [3] shows how to write down a theoretical model for ICL under such distribution shifts. While not necessary for acceptance to the workshop, I'd be interested to see if the authors are able to extend their results to incorporate the effect of distribution shift.

Another suggestion I have would be to reproduce the "empirical" data points in Figure 2 by training full, nonlinear transformers on the regression task. I would hope that the authors' model achieves at least qualitative agreement with such experiments, and these experiments would also let the authors investigate the effect of increasing parameter count via increasing depth vs increasing width.

Minor points: typo on line 2108 "Comparision"

[1] Can Transformer Models Generalize Via In-Context Learning Beyond Pretraining Data? Steve Yadlowsky, Lyric Doshi, Nilesh Tripuraneni, https://openreview.net/forum?id=oW7oQHas3m
[2] When can in-context learning generalize out of task distribution? Chase Goddard, Lindsay M. Smith, Vudtiwat Ngampruetikorn, David J. Schwab, https://openreview.net/forum?id=YKyza9lrv4
[3] Pretrain-Test Task Alignment Governs Generalization in In-Context Learning. Mary I. Letey, Jacob A. Zavatone-Veth, Yue M. Lu, Cengiz Pehlevan, https://arxiv.org/abs/2509.26551

---

### Official Review · Reviewer_VZES · 2026-02-27

**Fit:** 3
**Significance:** 2
**Confidence:** 2

**Summary:**

The authors reframe in-context learning as another form of classical benign overfitting. In-context learning can be broken down into in-distribution generalization (the model has seen a task before in pretraining, but not the specific prompt that is tested at inference time) and out-of-distribution generalization (the model has never seen the task or prompt before in pretraining). They explore the linear regression ICL task both theoretically and experimentally (with linear attention models) using two classical predictors, and they find that large (overparameterized) models perform well at both ID and OOD ICL.

**Strengths:**

1.	Well written and clear definitions of the problem set-up.
2.	Many experiments, including those that replicate key earlier results. I found these to be the most convincing aspects of the paper.

**Suggestions:**

1.	Figure 2’s labels and legend could be larger.
2.	The theory and experimental results only apply to a very toy setting: non-softmax attention Transformers (linear attention). The conclusion and significance of this work seem vastly overstated as a result. The authors write “scaling up need not force a tradeoff between solving known tasks and generalizing to new ones, even though prior theoretical models of ICL might suggest otherwise (Lu et al., 2025; Raventos et al., 2023; Park et al., 2025),” yet Raventos et al., (2023) and Park et al., (2025) study GPT-2 style Transformers. The authors do not show that their discovery of models that can both perform well at ID and OOD ICL also applies to standard Transformer models.

---

### Official Review · Reviewer_JPqQ · 2026-02-28

**Fit:** 2
**Significance:** 2
**Confidence:** 3

**Summary:**

They study in-context linear regression in a linear model of a fixed subset of the $x_q$ to be predicted and the average of the label weighted in-context $x_t$'s. Specifically, they are interested in in-distribution (ID) task versus out-of-distribution (OOD) task performance, where tasks refer to in-context examples generated from a fixed $w \sim \mathcal{D}_\text{task}$.

First, they theoretically describe the performance in the high-dimensional limit. Theorems 1 and 2 provide sharp high-dimensional limits for the ID and OOD risks under proportional scaling. In the underparameterized regime ($\nu > \rho^2$), the least-squares estimator exhibits classical bias–variance behavior: both ID and OOD risks decrease with increasing model scale until the interpolation threshold $\rho^2 = \nu$, where a double-descent peak occurs. In the overparameterized regime ($\nu < \rho^2$), the minimum-norm interpolator achieves low ID risk by effectively memorizing pretraining tasks, while the OOD risk converges to a finite constant comparable to the best underparameterized performance. Thus, increasing model capacity beyond interpolation does not degrade OOD generalization, but instead enables simultaneous task memorization and novel-task generalization. They call this phenomenon in-context benign overfitting, and it demonstrates that scaling can eliminate the retrieval–learning tradeoff predicted by earlier fixed-capacity models.

Additionally, they have thorough experimental results on their toy model showing great alignment of the theoretical loss predictions as a function of model scale ($\rho^2 / \nu$) and experimental results on a range of different task distributions and data distributions.

**Strengths:**

The paper is technically careful: the asymptotic analysis is detailed, the presentation is clean, and the empirical curves match the theoretical predictions closely. Conceptually, the work provides an existence proof that a benign-overfitting-like phenomenon can occur at the task level in an ICL-style regression setting, and it highlights the importance of choosing an appropriate scaling axis (here, varying effective parameter dimension via $\rho$ rather than implicitly tying scale to the ambient dimension). Relative to prior asymptotic analyses of linear-attention ICL (particularly Asymptotic theory of in-context learning by linear attention by Lu et al. (2024)), the feature-selection mechanism is an explicit modeling device that breaks the apparent retrieval--learning tradeoff with respect to model scale in this toy environment.

**Suggestions:**

My suggestions mainly concern external validity and the interpretability of the model scale knob. The key mechanism that enables the claimed benign regime is the feature-selection parameter $\rho<1$, which hard-restricts the learner to a random subset of coordinates. While this is mathematically natural for studying interpolation and double descent (since the parameter dimension scales like $p^2$), it is not obvious what this corresponds to in trained transformers (or other models), where scaling typically changes width/depth and induces representation learning rather than literal coordinate masking. Strengthening the paper would require a clearer empirical bridge: e.g., an operational definition of an effective dimension (or effective degrees of freedom) in trained models that plays the role of $p$, and evidence that plotting performance against an analogue of $N/p_{\mathrm{eff}}^2$ meaningfully organizes real ICL behavior.

Relatedly, the paper downplays the task diversity lever that is central in prior synthetic ICL work: in settings like Lu et al. (2024), the transition between ID and OOD behavior is strongly controlled by the diversity/size of the pretraining task family, and tradeoffs appear as this diversity changes. In contrast, the present theory is organized around capacity scaling at fixed proportional limits, and does not provide a joint characterization of how capacity and task diversity interact. A concrete improvement would be to include (even in a simplified trained-transformer regression testbed) a two-dimensional phase diagram over (i) task diversity and (ii) model size, reporting both ID-style and OOD-style errors. This would help clarify when the proposed benign phenomenon persists versus when diversity-driven tradeoffs dominate, and would more directly motivate why the paper's scaling limit should be preferred for understanding realistic ICL regimes.

---

### Meta-Review · Area_Chair_fJYk · 2026-02-28

**Recommendation:** Accept

**Metareview:**

This paper studies overfitting and generalization in in-context learning of linear regression. All reviewers noted the novelty and quality of the contributions of the paper. I recommend acceptance.

---

### Decision · Program_Chairs · 2026-03-02

Accept